

# The oldest *Archaeopteryx* (Theropoda: Avialiae): a new specimen from the Kimmeridgian/Tithonian boundary of Schamhaupten, Bavaria

Oliver W.M. Rauhut[1,2,3], Christian Foth[4,5] and Helmut Tischlinger[6]

[1] SNSB—Bayerische Staatssammlung für Paläontologie und Geologie, Munich, Germany
[2] Department of Earth and Environmental Sciences, Ludwig-Maximilians University, Munich, Germany
[3] GeoBioCenter, Ludwig-Maximilians University, Munich, Germany
[4] Department of Geosciences, Université de Fribourg, Fribourg, Switzerland
[5] Staatliches Museum für Naturkunde, Stuttgart, Germany
[6] Stammham, Germany

Corresponding author
Oliver W.M. Rauhut,
o.rauhut@lrz.uni-muenchen.de

## ABSTRACT

The iconic primeval bird *Archaeopteryx* was so far mainly known from the Altmühltal Formation (early Tithonian) of Bavaria, southern Germany, with one specimen having been found in the overlying Mörnsheim Formation. A new specimen (the 12th skeletal specimen) from the earliest Tithonian Painten Formation of Schamhaupten (Bavaria) represents the so far oldest representative of the genus. The new specimen shows several interesting anatomical details, including the presence of a postorbital in contact with the jugal, the presence of a separate prefrontal and coronoid, and opisthocoelous mid-cervical vertebrae. Based on observations on the new specimen, we discuss several problematic issues concerning *Archaeopteryx*, including the monophyly and diagnosis of the genus, the absence/presence of the sternum, the position of the gastralia, and variation in morphometrics and dental morphology in that genus. Based on a new diagnosis for the genus *Archaeopteryx*, the Berlin, Eichstätt, Solnhofen, Munich, Daiting, Thermopolis, 11th, and 12th specimens can be referred to this genus with high certainty. The Maxberg specimen is very probably also an *Archaeopteryx*, based on overall similarity, although none of the diagnostic characters can be evaluated with certainty. The ninth specimen ('chicken wing') might be *Archaeopteryx*, but cannot be referred to the genus with any certainty. In comparison with other paravians, the presence of distally thickened anterior pectoral ribs indicates that a rather large cartilagenous sternum was present in this taxon. In contrast to non-opisthopubic theropods, opisthopubic taxa, such as *Archaeopteryx* and many other paravians, have the posterior end of the gastral basket preserved at about half-length of the pubis, which might reflect the post-mortem collapse of enlarged abdominal air sacs in these taxa. Specimens that can be referred to *Archaeopteryx* show a high amount of variation, both in the morphometrics of the limb bones as well as in the dentition. In respect to the latter aspect, variation is found in tooth number, spacing, orientation, and morphology, with no two specimens showing the exact same pattern. The significance of this variation is unclear, and possible explanations reach from high intraspecific (and possibly ontogenetic and/or sexual dimorphic)

variation to the possibility that the known specimens represent a 'species flock' of *Archaeopteryx*, possibly due to island speciation after the initial dispersal of the genus into the Solnhofen Archipelago.

## INTRODUCTION

When the first skeleton of the 'Urvogel' *Archaeopteryx* was discovered in 1861, it represented the first skeletal evidence for a pre-Tertiary bird (*Wellnhofer, 2008*, *2009*). Furthermore, the discovery came just two years after the publication of Darwin's *Origin of species* (*Darwin, 1859*), and the intermediate morphology of *Archaeopteryx* rapidly became an important argument in favour of his theory (*Huxley, 1868*). It is thus not surprising that *Archaeopteryx* became a famous and important fossil, an 'icon of evolution' (*Wellnhofer, 2009*), as it long represented the only good evidence for the transition from reptiles to birds. One other aspect that certainly added to this fame was the elusive nature and rarity of *Archaeopteryx* discoveries: after the second discovery of an *Archaeopteryx*, probably in 1875 (*Tischlinger, 2005*), it was not until 1959 that a new specimen was announced (*Heller, 1959*). However, since the 1970s, an increasing number of new (or newly identified) specimens have been described (*Ostrom, 1970*, *1972*; *Mayr, 1973*; *Wellnhofer, 1974*, *1988a*, *1993*; *Mäuser, 1997*; *Mayr, Pohl & Peters, 2005*; *Wellnhofer & Röper, 2005*; *Tischlinger, 2009*; *Foth, Tischlinger & Rauhut, 2014*), making *Archaeopteryx* a rather well-known taxon today (see *Wellnhofer, 2008*, *2009*; *Rauhut & Tischlinger, 2015*).

Of the 11 *Archaeopteryx* specimens recovered so far, the vast majority comes from the Altmühltal Formation ('Solnhofen limestones' sensu stricto; *Niebuhr & Pürner, 2014*). Only the eighth (Daiting) specimen was found in the overlying Mörnsheim Formation and is thus slightly younger than the other specimens. The so far oldest specimen is the Berlin specimen, which comes from the Lower Eichstätt Member of the Altmühltal Formation (*Tischlinger, 2005*; *Wellnhofer, 2008*; *Niebuhr & Pürner, 2014*), corresponding to the higher part of the *Hybonotum riedense* subzone of the early Tithonian (*Schweigert, 2007*, *2015*). In summer 2010, a private collector found a new specimen of *Archaeopteryx* in the visitor quarry at Schamhaupten, in sediments at the base of the Painten Formation, which are probably slightly older than the rocks that have hitherto yielded remains of the Urvogel.

## HISTORY OF FIND AND GEOLOGICAL CONTEXT

The new specimen comes from the village of Schamhaupten, east-central Bavaria (Fig. 1). Two localities are found in this area. One is the Stark Quarry, west of the village, which was excavated by the Jura-Museum Eichstätt in the 1990s and has yielded numerous important vertebrate specimens (*Renesto & Viohl, 1997*; *Göhlich & Chiappe, 2006*; *López-Arbarello & Sferco, 2011*). These fossils were found in silicified limestones that were referred to the Painten Formation by *Viohl & Zapp (2007)* and *Viohl (2015a)*,

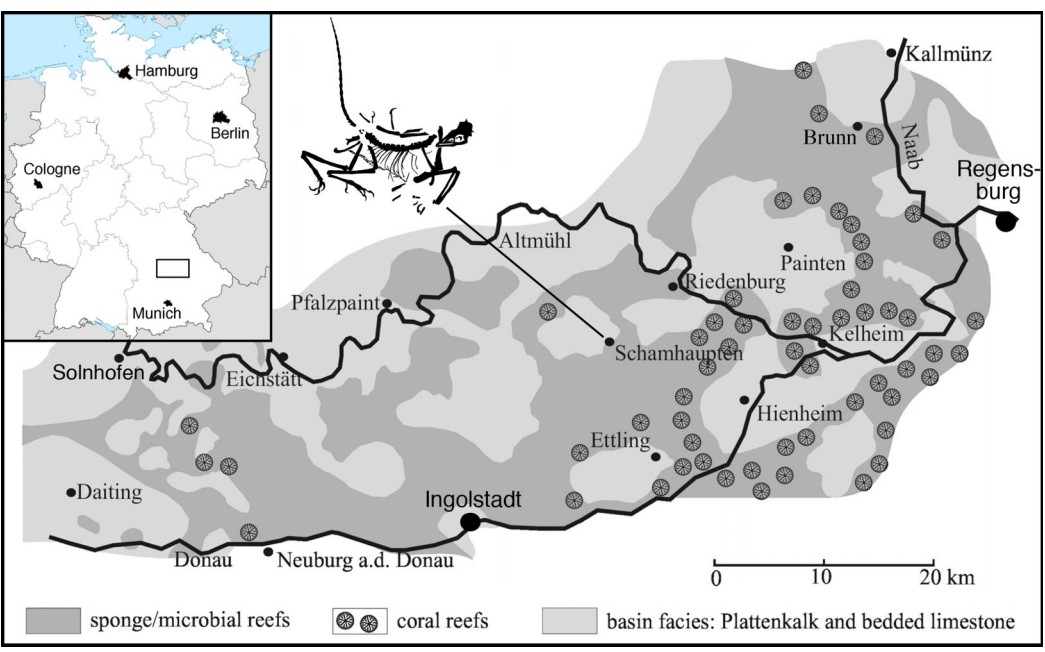

**Figure 1 Geographic position of the locality of the 12th specimen of *Archaeopteryx*, the village of Schamhaupten, within the palaeo-archipelago of Solnhofen.** Modified from *Rauhut et al. (2017)*.

but considered to be equivalent to the uppermost part of the Torleite Formation by *Niebuhr & Pürner (2014)*. Regardless of the exact geological unit, these sediments have been dated in the *rebouletianum* horizon of the *beckeri* ammonite zone of the uppermost Kimmeridgian by *Schweigert (2007, 2015)*.

The second locality is the Gerstner Quarry, northwest of the village of Schamhaupten, which is a touristic quarry provided by the rural district of Eichstätt for tourists and interested laymen to collect fossils. The quarry represents strata of the Öchselberg member within the Painten Formation, which correspond to the Kimmeridgian/Tithonian boundary (*Schweigert, 2007, 2015*). The new specimen was discovered and excavated with great care by a private collector. After initial preparation indicated that the specimen (originally believed to be a pterosaur) indeed represented *Archaeopteryx*, the finder graciously brought it to the attention of one of us (OR) at the Bayerische Staatssammlung für Paläontologie und Geologie, where a first evaluation of the specimen was carried out. In the course of this evaluation, one of the authors (HT) visited the locality together with the finder to take additional data on the geological context of the specimen. Furthermore, the specimen was registered as German national cultural heritage, which guarantees its permanent availability, even though it remains in private hands (Datenbank National Wertvollen Kulturgutes number DNWK 02924). Thus, thanks to the prudence and generosity of the finder, the specimen could not only be secured for scientific research, but its geological context could be evaluated in greater detail than it is the case for many other specimens of *Archaeopteryx*. The difficult preparation of the specimen was carried out under the supervision of Raimund Albersdörfer, and upon termination of the preparation, the specimen was given on loan to the authors at the Bayerische

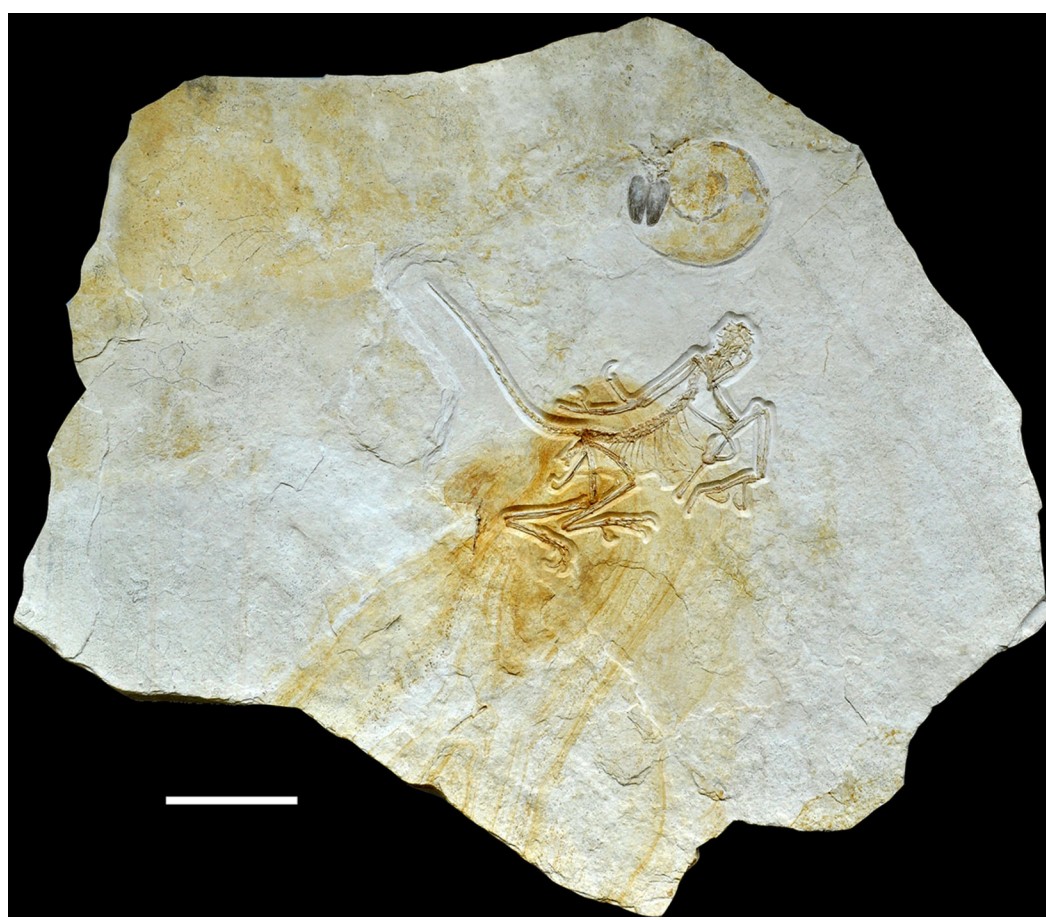

**Figure 2 Complete slab of the 12th *Archaeopteryx*, with ammonite, probably *Neochetoceras bous*, preserved on the same slab.** Scale bar is 10 cm.

Staatssammlung für Paläontologie und Geologie for scientific study. It is currently housed at the Dinosaurier Freiluftmuseum Altmühltal in Denkendorf, Bayern, where it is on public display and available for further scientific study.

The new specimen of *Archaeopteryx* was found in lithographic, slightly silicified limestones (Fig. 2) near the lowermost section of the quarry. The exact biostratigraphic position of the horizon is difficult to establish due to the rarity of clearly discernable index ammonites. A poorly preserved, large ammonite near the skull of the new *Archaeopteryx* (Fig. 2) probably represents *Neochetoceras bous* (Oppel) (G. Schweigert, 2016, personal communication to HT), which is typical of the lowermost Tithonian biohorizon and points to an age older than the *riedlingensis* horizon of the Lower Solnhofen member of the Altmühtal Formation. Another ammonite, found in beds near the place of discovery of the new *Archaeopteryx,* belongs to *Lithacoceras eigeltingense* Ohmert & Zeiss (G. Schweigert, 2016, personal communication to HT) and is characteristic of the lowermost Tithonian biohorizon, the *eigeltingense* horizon (*Schweigert, 2007*, *2015*). Given the proximity of the latest Kimmeridgian beds in this area, the new *Archaeopteryx* certainly comes from close to the Kimmeridgian–Tithonian boundary and is thus older than previous finds (Fig. 3), but the time difference to the Berlin specimen might be minimal.

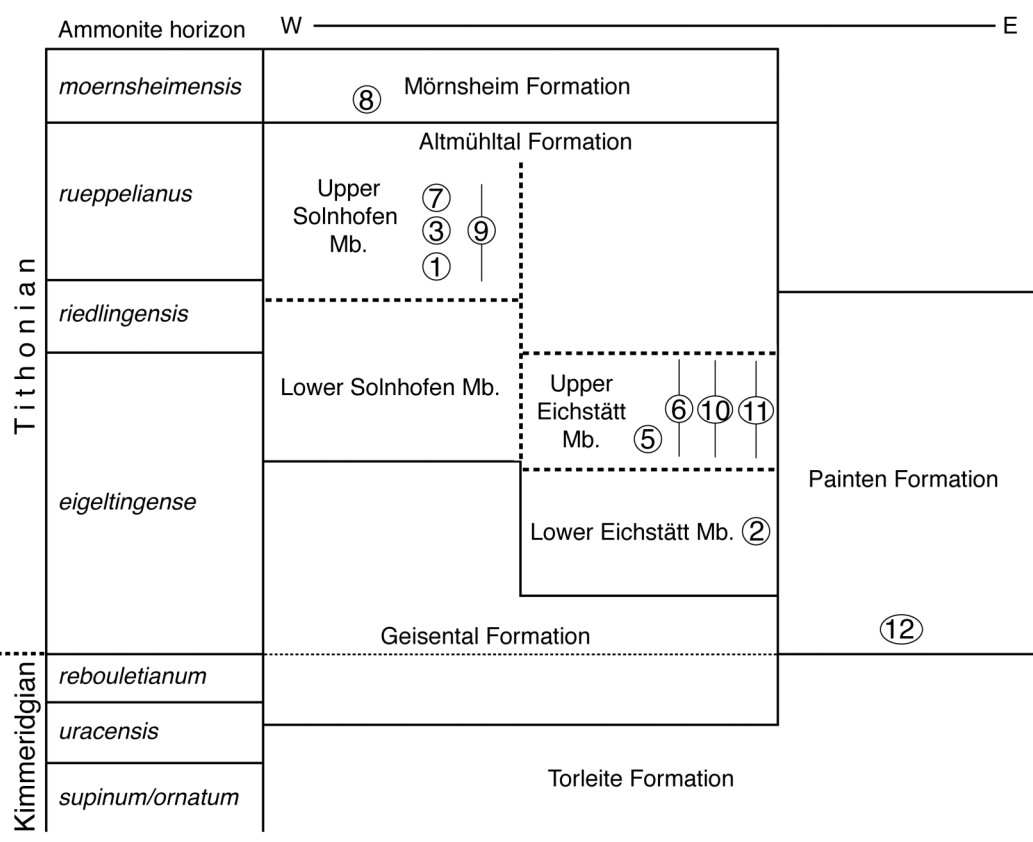

**Figure 3 Stratigraphic position of the new find of *Archaeopteryx*, in comparison with previous *Archaeopteryx* specimens.** Specimens: 1, London specimen; 2, Berlin specimen; 3, Maxberg specimen; 5, Eichstätt specimen; 6, Solnhofen specimen; 7, Munich specimen; 8, Daiting specimen; 9, 'Chicken Wing;' 10, Thermopolis specimen; 11, 11th specimen; 12, 12th specimen described here. Specimens are figured in their relative position within their members; specimens for which no exact stratigraphic position is known are indicated by vertical bars. E, east; Mb, member; W, west. Note that vertical thickness of ammonite horizons is for practical reasons and does not reflect absolute duration of horizons. Stratigraphic scheme modified from *Niebuhr & Pürner (2014)*, with biostratigraphic data from G. Schweigert (2007, 2015, 2017, personal communication to OR) and occurrences of *Archaeopteryx* specimens from *Wellhofer (2008)*.               

# UV DOCUMENTATION

Generally, skeletal remains and slightly mineralised soft parts from the Upper Jurassic plattenkalks of southern Germany and from many other deposits are fluorescent under ultraviolet (UV) light. During the past two decades the use of UV in fossil tetrapod research has proven to be important in revealing new information (*Tischlinger, 2002*; *Tischlinger & Arratia, 2013*). Observations under UV light allow a more precise investigation of morphological details of skeletal remains as well as soft parts. Frequently, delicate skeletal elements and relics of soft parts are hardly or not identifiable in visible light but light up conspicuously under filtered UV. The technique can be used to distinguish bone sutures from cracks, to establish outlines of compressed skeletal elements more clearly, and to separate bones or soft parts from the underlying matrix.

Sometimes only by pictorial documentation under UV light essential details of bones and soft parts can be demonstrated, due to the fact that the researcher will not be able to

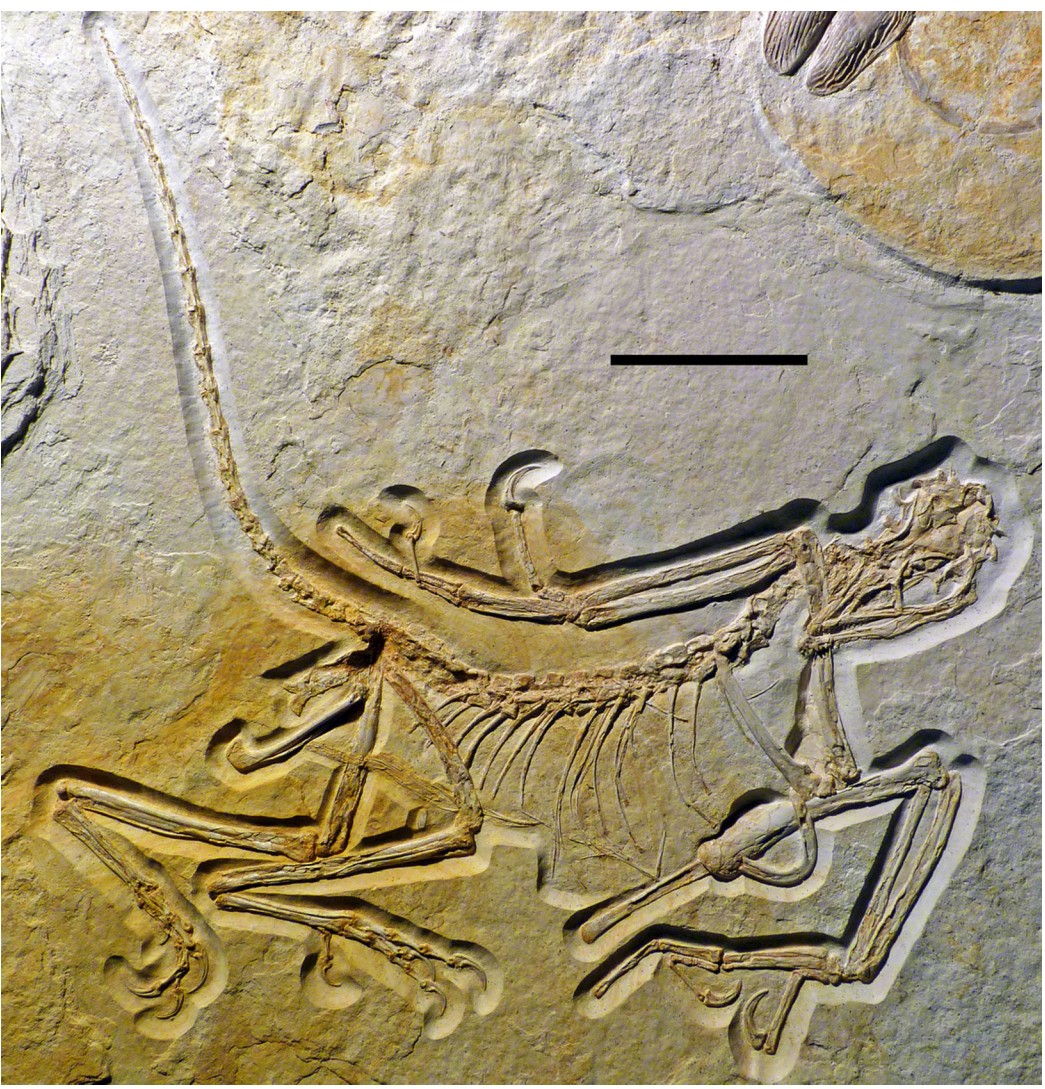

**Figure 4 Overview of the skeleton of the new *Archaeopteryx* specimen under normal light.** Scale bar is 50 mm.

differentiate tiny structures and differences in colour and composition under UV light with the naked eye or with the microscope. The visibility of details is enhanced considerably by an established filtering technique, crucial for the photographic documentation. The application of different filters allows a selective visualisation of peculiar fine structures. Colour compensation filters (yellow, cyan and magenta of different types and densities) are adjusted in front of the camera lens or under the microscope objective lens (if pictures are taken through the microscope). In most cases a selection of different colour compensation filters is necessary. The predominant colour of luminescence is of minor importance. Rather, the essential decision on the amount of filtering is the perfect visibility of details and their differentiation from surrounding structures and the matrix (*Tischlinger & Arratia, 2013*).

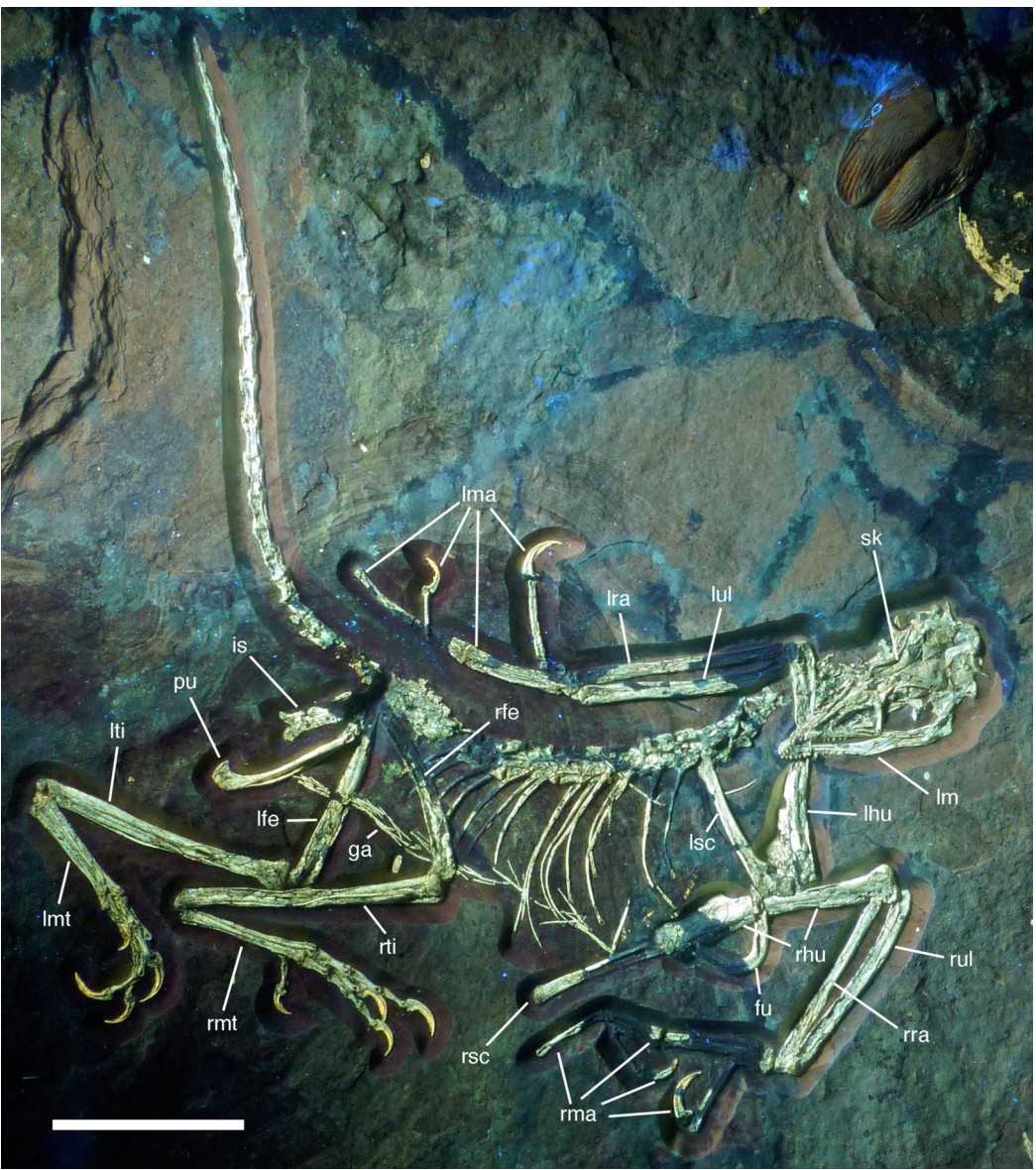

**Figure 5 Overview of the skeleton of the new *Archaeopteryx* specimen under UV light.** Areas of the skeleton that remain dark have been reconstructed during preparation. fu, furcula; ga, gastralia; is, ischium; lfe, left femur; lhu, left humerus; lma, remains of left manus; lmt, left metatarsus; lra, left radius; lsc, left scapula; lti, left tibia; lul, left ulna; pu, pubis; rfe, right femur; rhu, right humerus; rma, remains of right manus; rmt, right metatarsus; rra, right radius; rsc, right scapula; rti, right tibia; rul, right ulna; sk, skull. Scale bar is 50 mm.

During the UV investigation of the 12th specimen of *Archaeopteryx* best results were obtained with a wavelength of 365–366 nm (long-wave radiation, UV-A). For the pictorial documentation of the 12th specimen of *Archaeopteryx* the following UV lamps were used: three Benda UV lamps: type N, 16 W, UV-A, 366 nm (size of filter 200 mm × 50 mm); one Labino UV lamp: UV-Spotlight S135, 35 W, UV-A, peak at 365 nm: spotlight (>50,000 µW per cm$^2$ at 30 cm distance) plus midlight reflector replacement (>8,000 µW per cm$^2$ at 30 cm distance).

| Table 1 Skeletal measurements of the new specimen of *Archaeotperyx*. | |
|---|---|
| **Element** | **Length (mm)** |
| Skull | 56.0 |
| Mandible | 45.5 |
| Scapula sin | 43.0 |
| Furcula height | 18.6 |
| Furcula width | 20 (est) |
| Humerus sin | 61.0 |
| Radius sin | 54.4 |
| Ulna sin | 55.0 |
| Mc I sin | 6.8 |
| Mc II sin | 28.2 |
| Mc III sin | 27.2 |
| P I-1 sin | 20.7 |
| Ungual I sin | 9.4 |
| P II-1 sin | 16.0 |
| P II-2 | 19.3 |
| P III-1 | 7.1 |
| P III-2 | 4.6 |
| P III-3 | 12.0 |
| Ungual III sin | 7.5 |
| Femur sin | 53 (est) |
| Tibiotarsus sin | 67.4 |
| Tibia sin | 66.0 |
| Mt I sin | 8.2 |
| Mt II sin | 31.6 |
| Mt III dex | 34.0 |
| Mt IV sin | 33.1 |
| P I-1 sin | 7.0 |
| Ungual I sin | 6.3 |
| P II-1 sin | 8.4 |
| P II-2 sin | 8.3 |
| Ungual II sin | 9.8 |
| P III-1 dex | 10.5 |
| P III-2 dex | 9.2 |
| P III-3 dex | 8.0 |
| Ungual III dex | 8.1 |
| P IV-1 dex | 7.3 |
| P IV-2 dex | 6.5 |
| P IV-3 dex | 5.8 |
| P IV-4 dex | 5.9 |
| Ungual IV dex | 6.2 |

(Continued)

**Vertebrae**

| Position | Centrum length (mm) |
|---|---|
| Cervical | >7 |
| C8 | 8.1 |
| C9 | 6.1 |
| D1 | 5.3 |
| D7 | 6.2 |
| D8 | 6.2 |
| D9 | 6 |
| D10 | 6 |
| Ca 4 | c. 5.8 |
| Ca 5 | c. 6.1 |
| Ca 6 | 6.2 |
| Ca 7 | 7.5 |
| Ca 8 | 9 |
| Ca 9 | 9.7 |
| Ca 10 | 9.9 |
| Ca 11 | 10.1 |
| Ca 12 | 10.1 |
| Ca 13 | 9.9 |
| Ca 14 | 9.9 |
| Ca 15 | 9.7 |
| Ca 16 | 9.3 |
| Ca 17 | 9 |
| Ca 18 | 8.7 |
| Ca 19 | 6.8 |
| Ca 20 | 5.3 |
| Ca 21 | 3.2 |
| Ca 22 | 2.4 |

**Notes:**
All measurements in mm.
C, cervical vertebra; ca, caudal vertebra; D, dorsal vertebra; dex, dextra (right); est, estimated; Mc, metacarpal; Mt, metatarsal; P, phalanx; sin, sinistra (left).

## DESCRIPTION

### Preservation

The new specimen of *Archaeopteryx* is preserved as a largely articulated skeleton, lying on its left side (Figs. 4 and 5). Only the shoulder girdles and arms, as well as the skull have been slightly dislocated from their original positions, but the forelimbs remain in articulation. The skull has been detached from the vertebral column and rotated as to face backwards. The specimen was collected in four bigger and many small pieces, and breaks between the different slabs affected especially the presacral vertebral column, sacrum and ilium, as well as parts of the shoulder girdle and the right forelimb (Fig. 5). The specimen is furthermore strongly flattened, and as a consequence bone preservation

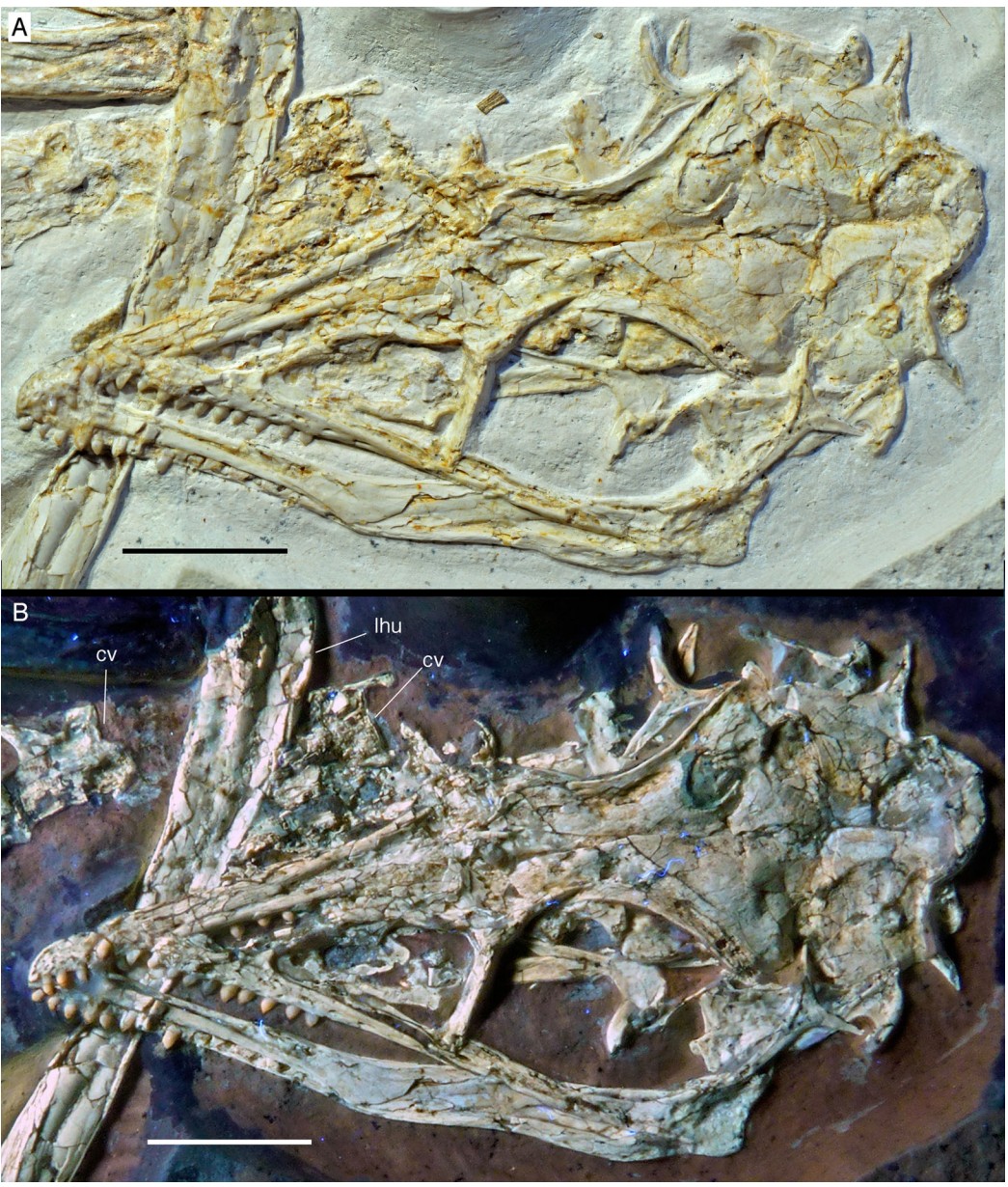

**Figure 6 The skull and mandibles of the 12th specimen of *Archaeopteryx*.** Skull and mandibles of the 12th specimen of *Archaeopteryx*, under normal light (A) and UV light (B). cv, cervical vertebra; lhu, left humerus. Scale bars are 10 mm.

is rather poor, with most long bones being collapsed and fractured. Likewise, the skull has suffered from compression and is somewhat incomplete. No feathers or impressions thereof are preserved, but some of the pedal unguals show remains of their horny sheaths.

For skeletal measurements of the specimen see Table 1.

## Skull

As noted above, the skull has been rotated from its former attachment to the vertebral column and is mainly exposed in dorsal and dorsolateral view (Figs. 6 and 7). Most

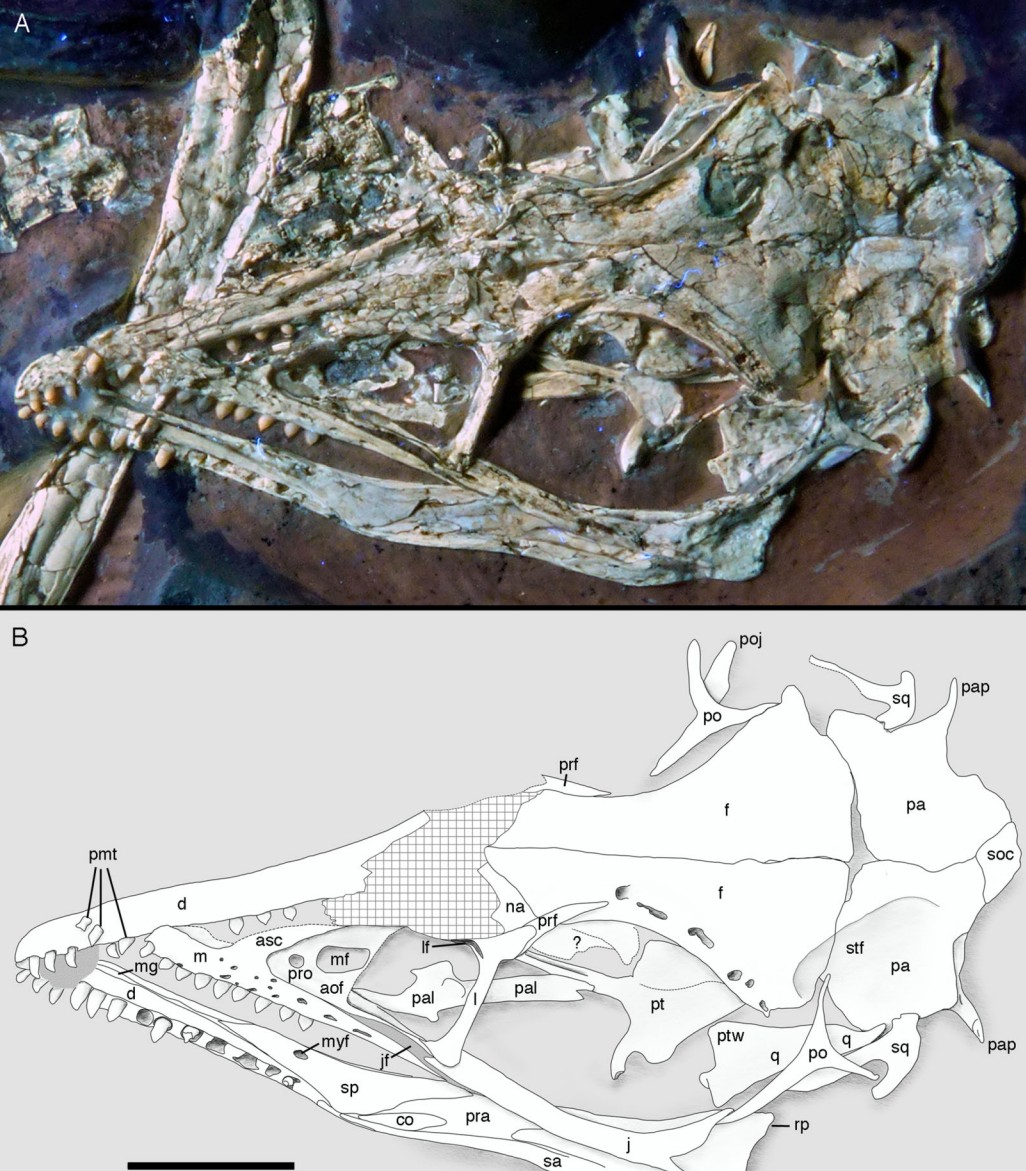

**Figure 7 Morphology of the skull and mandibles of the 12th specimen of *Archaeopteryx.*** (A) Photograph under UV light. (B) Explanatory drawing. aof, antorbital fossa; asc, ascending process of the maxilla; co, coronoid; d, dentary; f, frontal; j, jugal; jf, jugal facet on the maxilla; l, lacrimal; lf, lacrimal fenestra; m, maxilla; mf, maxillary fenestra; mg, meckelian groove; myf, myliohyoid foramen; na, nasal; pa, parietal; pal, palatine; pap, paroccipital process; pmt, premaxillary teeth; po, postorbital; poj, postorbital process of right jugal; pra, prearticular; prf, prefrontal; pro, promaxillary foramen; pt, pterygoid; ptw, pterygoid wing of the quadrate; q, quadrate; rp, retroarticular process of the mandible; sa, surangular; soc, supraoccipital; sp, splenial; sq, squamosal; stf, supratempora fossa. Question mark denotes unidentified element; hatched area represents broken fragments, mainly of the nasals. Drawing by A. López-Arbarello. Scale bar is 10 mm.   

cranial bones are preserved in articulation, but have been affected by compression and breakage. The left side of the skull roof has largely been flattened into the bedding plane, and the dorsal side is exposed posteriorly. The mandibles are preserved in articulation, but

have been flattened, so that the left mandible is exposed in medial view and the right mandible in lateral view (Figs. 6 and 7). Whereas the left mandible lies below the skull and is only overlapped by skull bones in minor parts posteriorly, only the anterior end of the right dentary is exposed and was obviously originally also overlapped by skull elements, such as the premaxilla and nasals.

The skull is approximately 56 mm long, with an error margin of 1–2 mm to account for compression. As in other specimens of *Archaeopteryx*, the skull is triangular in lateral outline. The orbit is the largest cranial opening, being approximately 16 mm long anteroposteriorly (the height cannot be established, as the skull is dorsolaterally compressed). The antorbital fenestra was approximately 6 mm long (the anterior margin is broken) and thus occupies half of the total length of the antorbital fossa (12 mm; Figs. 6–8). The lateral temporal fenestra is collapsed, but was obviously very narrow anteroposteriorly, as in other specimens of *Archaeopteryx*.

Only fragments and the impression of the anterior tip of the left **premaxilla** and remains of three premaxillary teeth are preserved (Fig. 8). As in other specimens of *Archaeopteryx*, the premaxilla was obviously elongate, with a straight, posterodorsally sloping anterior margin that is set at approximately 45° towards the alveolar border. The rounded anterior border of the external narial opening is placed above or just behind the fourth premaxillary tooth, as in other specimens (*Wellnhofer, 1974*; *Mayr et al., 2007*). At least parts of three tooth crowns are preserved in their original position, although the bone containing them is largely gone. The anteriormost two of these teeth lie on the lateral side of the anterior part of the right dentary and the third touches the dorsal margin of this bone, but is collapsed into a small cavity in between the two dentaries just posterior to the dentary symphysis. Between the anterior two and the last tooth there is a gap that indicates the presence of a fourth premaxillary tooth, as in other specimens of *Archaeopteryx* (Berlin specimen: *Dames, 1884*; Eichstätt specimen: *Wellnhofer, 1974*; Solnhofen specimen: *Wellnhofer, 1988b, 1992*; Thermopolis specimen: *Mayr et al., 2007*; 11th specimen: *Foth, Tischlinger & Rauhut, 2014*). A tooth is indeed present within this gap, but it is in the plane of the underlying dentary and perpendicular to the alveolar border of the latter and thus rather represents a dentary tooth. The first premaxillary tooth is notably offset from the anterior end of the bone impression so that there would be space for another tooth; however, as no known specimen of *Archaeopteryx* has more than four premaxillary teeth, and no tooth or tooth fragment is preserved anywhere near to this point, it seems more likely that the tip of the premaxilla was edentulous than that a further tooth position was present.

The left **maxilla** is largely complete and exposed in lateral and, partially, dorsolateral view (Figs. 6–8). The maxilla is approximately 20 mm long and c. 7 mm high and bears nine tooth positions, as in the Thermopolis specimen of *Archaeopteryx* (*Mayr, Pohl & Peters, 2005*; *Mayr et al., 2007*). The last maxillary tooth is placed at about the mid-length of the subantorbital ramus of the maxilla, below the posterior margin of the maxillary antorbital fossa on the ascending process, some 8.5 mm anterior to the orbit. Teeth are more widely spaced than in most other theropods, but spacing between individual

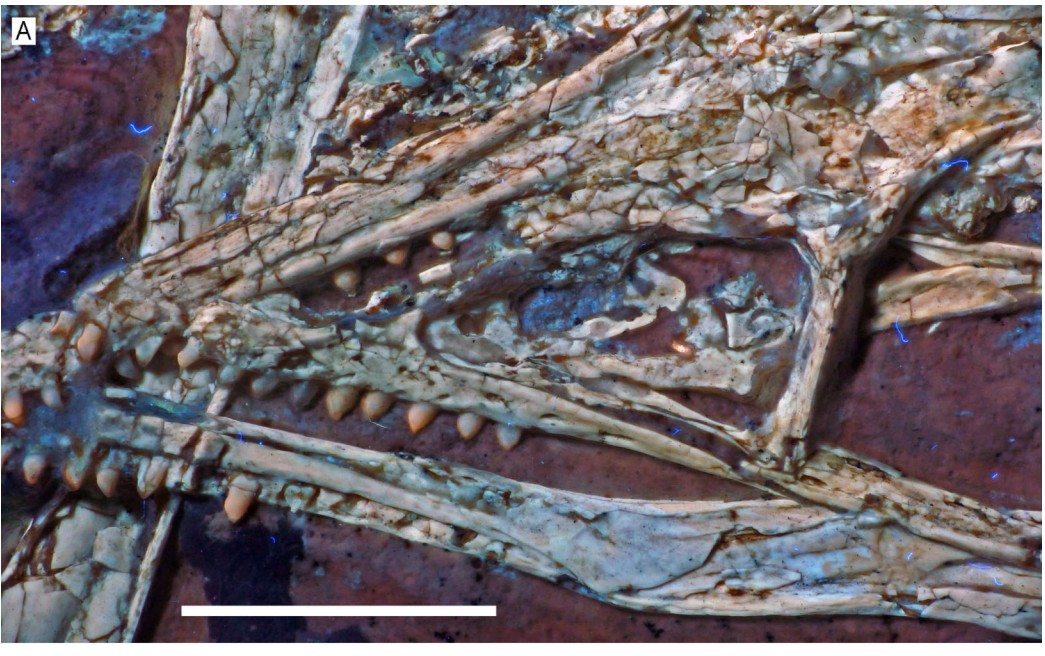

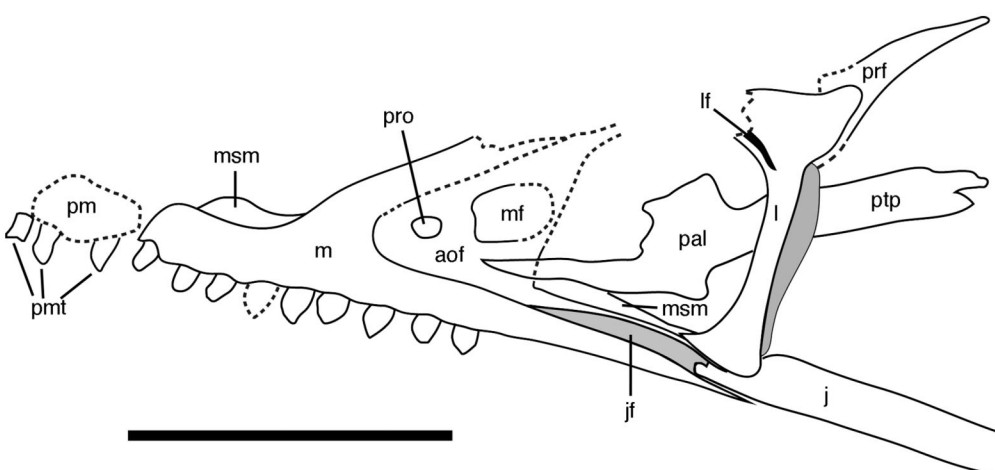

**Figure 8 Anterior part of the skull of the 12th specimen of *Archaeopteryx*.** (A) Photograph under UV light. (B) Interpretative drawing of anterior skull elements, with non-skull elements, mandibles and fragmentary elements omitted for clarity. aof, antorbital fossa; j, jugal; jf, jugal facet on maxilla; l, lacrimal; lf, lacrimal fenestra; m, maxilla; mf, maxillary fenestra; msm, medial shelf of the maxilla; pal, palatine; pm, remains of premaxilla; pmt, premaxillary teeth; prf, prefrontal; pro, promaxillary foramen; ptp, pterygoid wing of palatine. Scale bars are 10 mm.

elements varies. However, there seems to be a general tendency towards slightly more widely spaced teeth in the posterior part of the maxilla in the new specimen.

The maxilla has a long anterior process anterior to the ascending process (Fig. 8), which is considerably longer (c. 3.5 mm) than high (c. 1.8 mm). This seems to be in contrast to several articulated skulls (*Wellnhofer, 1974*, *1992*, *2008*; *Mayr et al., 2007*) and most skull reconstructions, which show a gradually anteriorly sloping anterior margin of the maxilla

(*Elzanowski, 2001a*, *2002*; *Wellnhofer, 2008*; *Rauhut, 2014*), but several lines of evidence indicate that this is the normal condition in this taxon. First, close inspection of the Eichstätt and Thermopolis specimens indicates that the anterior process of the maxilla is actually rather long, but partially overlapped by the subnarial process of the premaxilla. The latter observation is confirmed by detailed observation of the maxilla of the 12th specimen: Although the anterior process is somewhat widened transversely, its dorsal surface is convex labiolingually and shows a laterodorsal, slightly depressed facet for the subnarial process of the premaxilla. Thus, in articulation, the latter process would largely cover the dorsal part of the anterior process of the maxilla, resulting in the apparently gradually sloping anterior margin of this bone. Interestingly, a narrow ridge is present laterodorsally in continuation of this facet at the base of the ascending process of the maxilla, but becomes rapidly lower dorsally and fades into the transversely flat anterior margin of the ascending process at about the level of the anteriormost point of the antorbital fossa.

As in other specimens of *Archaeopteryx* (Eichstätt specimen: *Wellnhofer, 1974*; Thermopolis specimen: *Mayr et al., 2007*; *Rauhut, 2014*), a deeply depressed antorbital fossa is present and occupies most of the lateral surface of the ascending process (Fig. 8), but has little lateral exposure on the maxillary ramus ventral to the antorbital fenestra. Parts of the vertical and smooth medial wall of the fossa are broken away, including most of the margins of the apparently large maxillary fenestra. However, the anterior part of this wall is preserved and shows that the promaxillary foramen in this specimen is placed further dorsally and posteriorly than in the Thermopolis (*Mayr et al., 2007*; *Rauhut, 2014*) and, apparently, the Eichstätt specimen (*Wellnhofer, 1974*). In the latter two specimens, the foramen is dorsoventrally expanded and placed directly at the anteriormost end of the antorbital fossa and ventrally flush with the ventral border of the latter. In contrast, in the new specimen the foramen is offset from both borders by c. 1 mm each (Fig. 8). Furthermore, the foramen seems to be relatively smaller than in the other specimens and slightly wider anteroposteriorly (c. 1 mm) than dorsoventrally (c. 0.7 mm), although some uncertainty remains, as the dorsal margin is poorly preserved. Not much of the margin of the maxillary fenestra is preserved, but this opening seems to have been rather large, although its ventral margin also seems to have been offset from the ventral margin of the antorbital fossa, unlike the situation in the Eichstätt (*Wellnhofer, 1974*) and Thermopolis specimens (*Mayr et al., 2007*; *Rauhut, 2014*), but similar to *Anchiornis* (BMNHC PH804; *Pei et al., 2017*) and dromaeosaurids (*Xu & Wu, 2001*; *Burnham, 2004*; *Norell et al., 2006*; *Pei et al., 2014*; *Lü & Brusatte, 2015*). The dorsal part of the ascending process of the maxilla is poorly preserved, and nothing can be said about the contact with the nasal or the lacrimal.

The subantorbital ramus of the maxilla is slender and becomes gradually lower posteriorly, as in *Sinornithosaurus* (*Xu & Wu, 2001*) and *Zhenyuanlong* (*Lü & Brusatte, 2015*), but unlike the apparently more robust ramus in *Sapeornis* (*Wang et al., 2017a*). It reaches posteriorly to the level of the anteriormost part of the orbit. The posterior end of the lateral surface of the maxilla is twisted to face somewhat ventrolaterally, although this might be exaggerated by compression. At least six large lateral foramina are present in

the lateral surface of the subantorbital ramus of the maxilla (Figs. 7 and 8A). The first clearly identifiable of these is placed directly dorsal to the alveolar margin just posterior to the base of the fifth maxillary tooth. Another large foramen might be present dorsal to the mid-length of this tooth, but it is unclear whether this might not simply be a break. The next three foramina are placed posterior to the bases of the sixth, seventh and eighth maxillary tooth, respectively, but consecutively higher on the maxillary body, so that the fourth foramen is found at about mid-height of the maxillary ramus. The last two foramina are placed posterior to the tooth row and again slightly lower on the maxillary ramus. They open posterolaterally, and especially the last foramen, which is set at about the mid-length between the last maxillary tooth and the lacrimal, is anteroposteriorly elongate and has a well-developed groove continuing posterior to it over a short distance. The laterally twisted posterior end of the subantorbital ramus of the maxilla shows a well-developed longitudinal groove for the contact with the jugal, which reaches anteriorly to approximately the level of the anterior end of the antorbital fenestra, becoming narrower and shallower anteriorly (Fig. 8). Medial to the jugal contact, the posterior end of a long and gradually anteriorly expanding medial palatal shelf of the maxilla is visible (Fig. 8), as it is also present in other paravian theropods (*Ostrom, 1969*; *Currie, 1985*; *Makovicky et al., 2003*; *Currie & Varricchio, 2004*).

The left **jugal** is present, but poorly preserved (Figs. 6, 7 and 9). As in the Eichstätt (*Wellnhofer, 1974*; *Elzanowski & Wellnhofer, 1996*), Thermopolis (*Mayr et al., 2007*) and 11th (*Foth, Tischlinger & Rauhut, 2014*) specimens of *Archaeopteryx* it is an anteroposteriorly long, slender bone that is lowest at about the level of the mid-length of the orbit and becomes slightly higher anteriorly and posteriorly. The central part of the jugal is collapsed, indicating that it had a medial longitudinal furrow, as it is present in the jugal of the Munich specimen (BSPG 1999 I 50) and the 11th specimen (*Foth, Tischlinger & Rauhut, 2014*), and in *Anchiornis* (*Hu et al., 2009*). *Bambiraptor* seems to have an incipient stage of this character, as there is a longitudinal depression in the ventral half of the medial side of the anterior part of the jugal (*Burnham, 2004*). The anterior end is not preserved, but, as noted above, the facet on the maxilla indicates that it continued anteriorly to almost the anterior end of the antorbital fenestra. The posterior end shows the typical low, posteriorly inclined, triangular postorbital process with an incision from the lateral temporal fenestra at its base (Fig. 9), which *Foth, Tischlinger & Rauhut (2014)* interpreted as an autapomorphy of the genus *Archaeopteryx*. However, a similar incision seems to be present in *Microraptor* (*Pei et al., 2014*). As in other specimens and the dromaeosaurid *Bambiraptor* (*Burnham, 2004*), the quadratojugal process is very slender and low dorsoventrally (Fig. 9), but it is poorly preserved. Unlike the situation in most non-avialan theropods and at least the basal bird *Sapeornis* (*Wang et al., 2017a*), the end of the quadratojugal process does not seem to be forked for the reception of the anterior process of the quadratojugal.

For the first time in the available specimens, the postorbital process of the jugal is preserved in close association with the postorbital. The anterior margin of the postorbital process shows a well-developed facet for the slightly displaced ventral process of the postorbital, as in the Thermopolis specimen (*Rauhut, 2014*), clearly indicating a closed

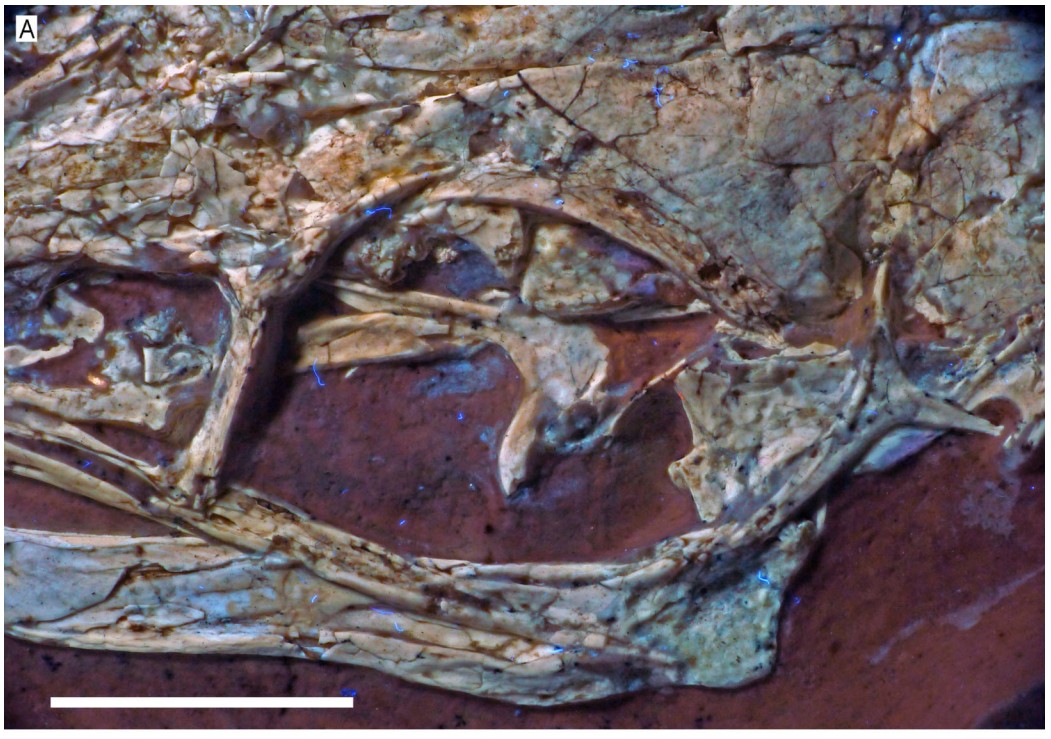

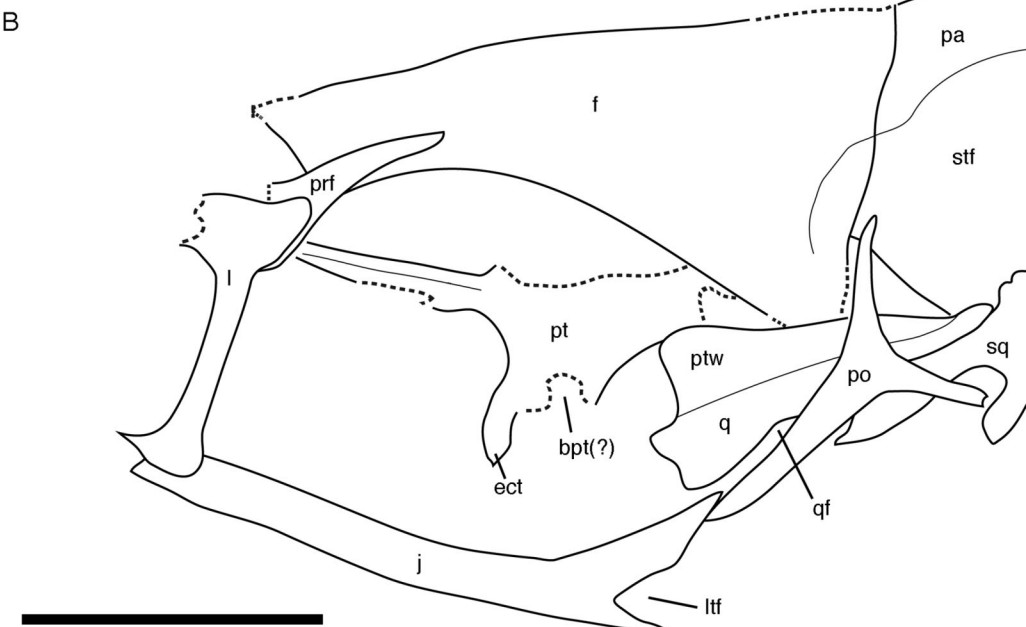

**Figure 9 Circumorbital region of the skull of the 12th specimen of *Archaeopteryx.* (A)** Photograph under UV light. **(B)** Interpretative drawing of cranial elements, with mandibles and several elements and fragments omitted for clarity. bpt, articular facet for the basipterygoid process of the basisphenoid; ect, ectopterygoid wing of pterygoid; f, frontal; j, jugal; l, lacrimal; ltf, incision of the lower temporal fenestra into the jugal; pa, parietal; po, postorbital; prf, prefrontal; pt, pterygoid; ptw, pterygoid wing of the quadrate; q, quadrate; qf, quadrate foramen; sq, squamosal; stf, supratemporal fossa. Scale bars are 10 mm.

postorbital bar, as in *Confuciusornis* (*Peters & Ji, 1998*; *Chiappe et al., 1999*) and non-avialan theropods.

Only the tip of the postorbital process of the right jugal is visible posterior to the ventral process of the right postorbital (Fig. 10). The process shows the tapering dorsal end with the facet for the postorbital and the beginning of the anteroventral incision of the infratemporal fenestra, thus confirming the observations on the left jugal.

Only fragments of the **nasals** are preserved and do not present much information (Figs. 6 and 8A). The intranasal suture is straight and the element seems to widen slightly posteriorly, as in the Thermopolis specimen (*Mayr et al., 2007*).

The left **lacrimal** is present, but poorly preserved and provides only limited information (Figs. 6–9). As in other maniraptorans, it is a slender, T-shaped element. As in the Thermopolis specimen (*Rauhut, 2014*), a lacrimal recess seems to have been present, but largely overlapped dorsolaterally by a lamina of the lacrimal roof (Fig. 8); its anterodorsal margin is visible through a break in the collapsed lamina. This overlapping lamina is obviously broken away in the Thermopolis specimen (*Rauhut, 2014*), giving the impression of a large, laterally opening recess. However, it is also present in the Eichstätt specimen (JME SOS2257), where it overhangs the lacrimal recess, which thus has little lateral exposure. Whereas the anterior process of the lacrimal is directed slightly anteroventrally, the posterior process is stouter and inclined steeply posterodorsally, as in *Confuciusornis* (*Chiappe et al., 1999*).

The ventral process is straight and seems to have a sharp edged lateral margin. A narrow rim of the antorbital fossa is present along its entire anterior margin and becomes slightly wider ventrally, where the ventral process expands anteriorly. This ventral expansion forms an anteriorly pointed ventral footplate that contacts the posterior part of the maxilla medial to the facet for the jugal (Fig. 8). A marked posterior expansion of the ventral process seems to be absent.

Not all of the posterodorsal extension continuing from the lacrimal seems to be part of this bone; a slender slip of bone at the anterolateral margin of the orbit probably represents a separate **prefrontal** ossification (Figs. 7–9), which is absent in many Pennaraptora (*Chiappe et al., 1999*; *Xu & Wu, 2001*; *Osmólska, Currie & Barsbold, 2004*; *Makovicky & Norell, 2004*; *Norell et al., 2006*; *Balanoff et al., 2009*; *Pei et al., 2014*). The refrontal is elongate and triangular in outline and forms about one-third of the prefrontal part of the dorsal orbital margin, separating the lacrimal from the frontal (Figs. 8 and 9). The anterior part of this bone is overlapped laterally by the lacrimal and thus is not visible in the new specimen, in which the lacrimal is mainly exposed in lateral view. The prefrontal seems to become thicker dorsoventrally in its anterior portion and can be seen both dorsal and ventral to the posterior process of the lacrimal; if the ventral portion represents a separate anterior ventral process, this process is short and restricted to the dorsal rim of the orbit, unlike the long ventral process in more basal theropods, which flanks the lacrimal medially over at least half of the anterior margin of the orbit (*Rauhut, Milner & Moore-Fay, 2010*: fig. 11).

Both **frontals** are preserved, but broken and somewhat deformed (Figs. 7, 9 and 10). As in the Eichstätt (*Wellnhofer, 1974*; *Elzanowski & Wellnhofer, 1996*) and Thermopolis

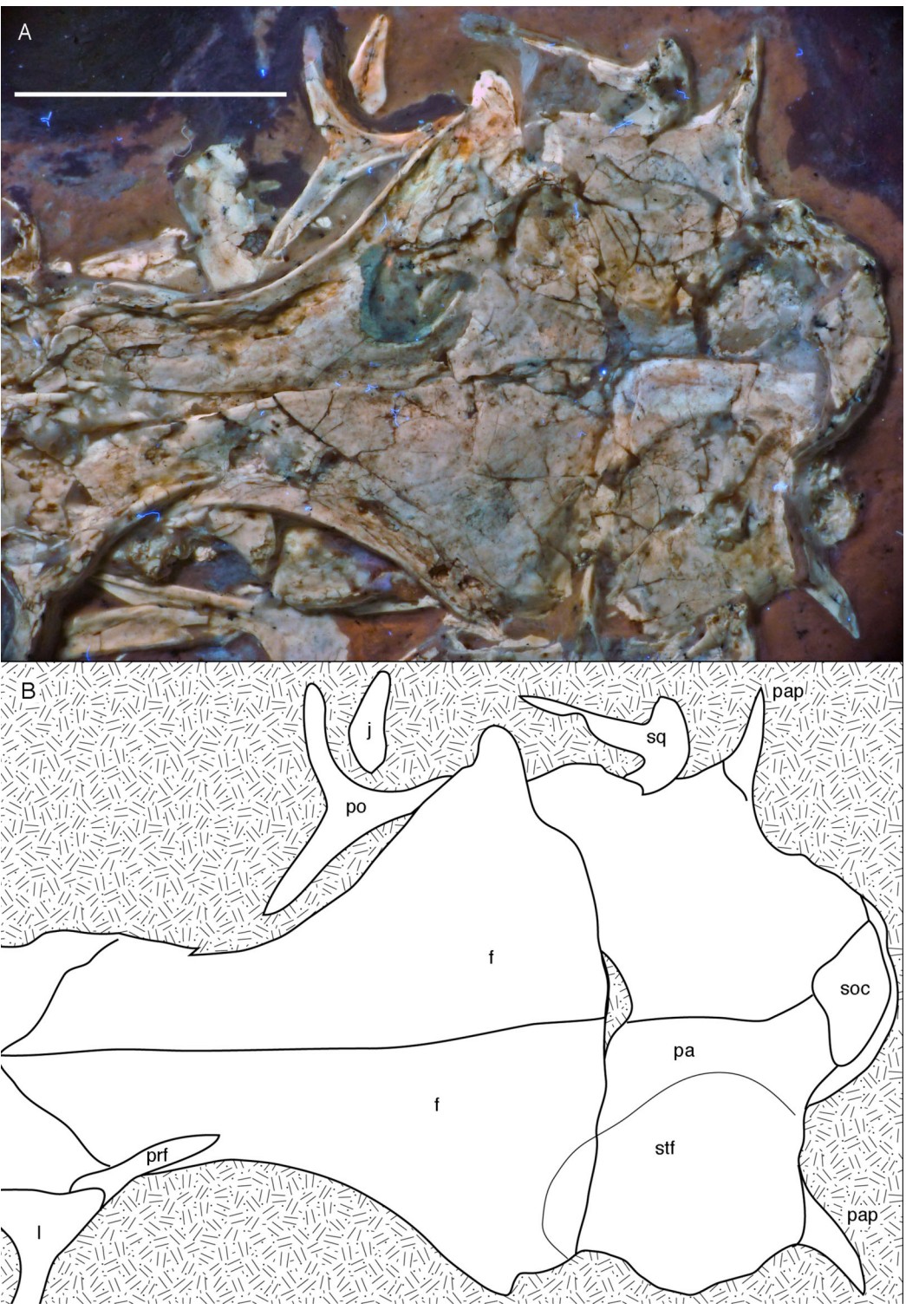

**Figure 10 Posterior part of the skull roof of the 12th specimen of *Archaeopteryx.*** (A) Photograph under UV light. (B) Interpretative drawing of dorsal skull roof elements. Some bone remains have been ommitted for clarity. f, frontal; j, jugal; l, lacrimal; pa, parietal; pap, paroccipital process; po, postorbital; prf, prefrontal; soc, supraoccipital; stf, supratemporal fossa. Scale bar is 10 mm.

(*Mayr et al., 2007*) specimens of *Archaeopteryx*, they are long and anteriorly slender elements, which are strongly transversely expanded posteriorly towards the contact with the parietal and postorbital. However, in contrast to some more basal coelurosaurs (e.g. ornithomimosaurs; *Osmólska, Roniewicz & Barsbold, 1972*; *Kobayashi & Lü, 2003*; *Troodon*; *Currie, 1985*) and at least some basal birds (e.g. *Sapeornis*: *Wang et al., 2017a*) the frontal is not triangular in dorsal outline, as the narrowest part of the bone is placed slightly anterior to the mid-length of the orbit and the bone expands slightly anterior to this point laterally (Fig. 10), similar to the situation in *Anchiornis* (BMNHC PH804; *Pei et al., 2017*), *Mei* (*Xu & Norell, 2004*), *Zanabazar* (*Norell et al., 2009*), the enantiornithine birds *Eoenantiornis* and *Longipteryx* (*O'Connor & Chiappe, 2011*), and as also seen in the Thermopolis specimen (*Mayr et al., 2007*; *Rauhut, 2014*). The postorbital process is preserved in the right element as a short, tongue-shaped lateral process posterior to the orbit (Fig. 10). Thus, the total length of the frontal is c. 22 mm, its maximal width, at the postorbital contact, c. 12 mm, the minimal width above the orbit c. 4 mm, and the anterior width approximately 4.5 mm. The anterior end of the frontal is oblique, so that the articulated frontals form an anteriorly pointing arrow, with the angle between the intrafrontal suture and the anterior margin being approximately 45°. The intrafrontal suture seems to be straight, and not interdigitating over its entire length. As in the Eichstätt (JM SOS2257), London (*Dominguez Alonso et al., 2004*), Daiting (*Tischlinger, 2009*) and Thermopolis specimens (*Mayr et al., 2007*; *Rauhut, 2014*), but also *Anchiornis* (PKUVP 1068, *Pei et al., 2017*), *Mei* (*Xu & Norell, 2004*), *Zanabazar* (*Norell et al., 2009*), and *Jeholornis* (*Lefèvre et al., 2014*), the orbital rim is slightly raised, resulting in a shallow, curved groove along the orbital rim on the dorsal surface of the frontal, in which numerous small pits or foramina are placed (Figs. 7, 9 and 10). In contrast, the posteromedial portion of the frontal, which forms the roof of the cerebrum (*Dominguez Alonso et al., 2004*), is dorsally vaulted, as in other specimens of *Archaeopteryx* (*Wellnhofer, 1974*; *Dominguez Alonso et al., 2004*). The suture with the parietal is straight and more or less strictly transversely oriented, as far as this can be discerned. The supratemporal fossa expands onto the posterolateral surface of the frontal (Figs. 7, 9 and 10). In contrast to more basal theropods, but as in many advanced coelurosaurs (*Clark, Norell & Rowe, 2002*; *Norell et al., 2006*, *2009*; *Lautenschlager et al., 2014*) the fossa does not have a pronounced rim and faces posterodorsally and not strictly dorsally.

The **parietals** are both present but strongly affected by compression so that they are crossed by many breaks (Figs. 7 and 10). As in the Thermopolis and Eichstätt specimens, the parietals are broad and bulbous elements that are convex transversely. The border of the supratemporal fossa is marked on the left parietal by a notable step that curves from the anterolateral facet on the frontal posteromedially and continues straight towards the nuchal crest (Figs. 7, 9 and 10). In contrast to more basal theropods, in which the dorsal parietal roof is usually offset from the border of the fossa by an almost straight angle, the parietal body lateral to this step slopes lateroventrally, owing to the enlarged brain proportions in *Archaeopteryx* (*Dominguez Alonso et al., 2004*). The dorsal roof of the parietal between the supratemporal fossae is broad and flat to slightly vaulted dorsally (Fig. 10), as in ornithomimosaurs (*Osmólska, Roniewicz & Barsbold, 1972*; *Kobayashi & Lü,*

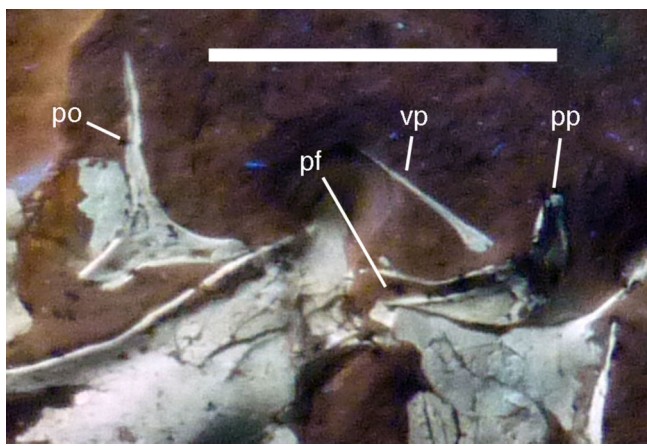

**Figure 11 Details of right postorbital and squamosal of the 12th specimen of *Archaeopteryx*.** Photograph under UV light. pf, postorbital facet on the anterior process of the squamosal; po, postorbital; pp, posterior process of the squamosal; vp, ventral process of the squamosal. Scale bar is 10 mm.

*2003*), therizinosaurs (*Lautenschlager et al., 2014*) and more derived avialans (*Chiappe et al., 1999*; *Zhou & Zhang, 2003b*; *O'Connor & Chiappe, 2011*), but in contrast to the sharp midline saggittal crest in troodontids (*Currie, 1985*; *Norell et al., 2009*) or dromaeosaurids (*Barsbold & Osmólska, 1999*; *Xu et al., 2015a*). Posteriorly, the parietal contributes to a low nuchal crest that extends laterally onto the slender squamosal processes of this bone, which, as in other theropods, are mainly oriented vertically (Fig. 10).

Posterior to the middle part of the skull roof, the **supraoccipital** has been pressed upwards and is visible behind the parietals (Fig. 10). As in the London specimen (*Dominguez Alonso et al., 2004*), its dorsal end is broad transversely and rounded dorsally. The bone is roughly diamond-shaped in outline and broader transversely than high dorsoventrally.

Both **postorbitals** are preserved, the left in its natural position in lateral view on the left side of the skull and in partial articulation with the jugal (Figs. 7 and 9), and the right in medial view slightly disarticulated from the right orbit (Figs. 10 and 11). As in all non-avian theropods, the postorbital is a triaradiate bone, with the ventral (jugal) process being the longest and the anterior (frontal) process the shortest of the three processes. In contrast to many non-avialan theropods, in which the postorbital is approximately T-shaped, the bone is Y-shaped in *Archaeopteryx*, as the anterior and posterior processes are slightly dorsally directed, the former more so than the latter. A similar Y-shaped postorbital is also present in the *Shuvuuia* (*Chiappe, Norell & Clark, 2002*) and *Confuciusornis* (*Chiappe et al., 1999*). Both the anterior and posterior processes are slender and triangular in outline and taper to a point (Fig. 9). The anterior process is slightly flexed, with the anterior end pointing anterodorsally. The posterior process is also very slightly flexed, but considerably less so than the anterior process. The ventral process is long and tapers ventrally, being very slightly flexed anteroventrally (Fig. 9). Its distal end would overlap the postorbital process of the jugal anteriorly, as in other theropods, but is slightly displaced so to lie posterior to the postorbital process of the jugal. The lateral

surface of the postorbital is smooth, and there is no marked thickening on the orbital margin or a depression on the confluence of the processes, as it is present in many basal theropods. In medial view, the ventral process has a slightly raised posterior margin, resulting in the presence of a shallow depression in at least its dorsal part.

Both **squamosals** are preserved, but slightly incomplete (Figs. 7 and 9–11). The right squamosal is preserved within the right supratemporal fenestra and is exposed in lateral view (Fig. 10). The left element is exposed in medial view (Fig. 9). It is incomplete anteriorly and its posterior process is covered by the parietal and the ventral process partially overlain by the postorbital.

The anterior part of the right squamosal shows a well-developed groove along the ventral margin of the anterior process for the reception of the posterior process of the postorbital (Fig. 11), as in other theropods. The groove is narrow with subparallel margins posteriorly, but widens in its anterior half, with its ventral margin showing a small kink at about its mid-length. The ventral process of the squamosal is slender, long and slightly flexed anteriorly. Whereas it remains of subequal anteroposterior width over most of its length, its distal end tapers and is not expanded anteroposteriorly, as it is the case in many basal theropods (*Rauhut, 2003*). The posterior process is flexed posteroventrally, so that the posterodorsal margin of the squamosal is strongly convex (Fig. 10). The process is short and blunt and forms the posterior border of a narrow, U-shaped incision between the ventral and posterior processes, which housed the dorsal head of the quadrate. This incision is considerably narrower than reconstructed by *Elzanowski & Wellnhofer (1996)* and the morphology of the quadrate articulation corresponds closely to that seen in many more basal theropods. In medial view, a stout ridge arises on the ventral process along its posterior margin at about its mid-length and becomes higher dorsally. It forms the anterior border of the contact with the quadrate and curves into the medial wall of the dorsal socket for the reception of the quadrate head dorsally. The ventral process is long and slender and tapers to a point ventrally (Fig. 10).

The left **quadrate** is partially exposed at the posterior margin of the orbit (Fig. 7). The dorsal portion is partially overlapped by the postorbital, but the dorsal end is visible dorsal to this bone within the supratemporal fenestra, in near articulation with the left squamosal (Fig. 9). The pterygoid wing and the mandibular condyle are visible in anterolateral view in the orbit (Fig. 9). The pterygoid wing is dorsoventrally extensive and triangular in outline, expanding gradually ventrally, as in *Tsagaan* (*Norell et al., 2006*) and *Bambiraptor* (*Hendrickx, Araújo & Mateus, 2015a*). The ventral margin is straight and set at an angle of approximately 90° towards the quadrate shaft anteriorly, but gently curves towards the quadrate condyle in its proximal part. The margin flexes slightly medially over its entire length. A small foramen is present on the medial side of the quadrate shaft at the base of the pterygoid wing and faces anterolaterally. The anterodorsal margin of the pterygoid wing is straight and seems to extend all the way to the dorsal head. At the shaft, the bone flexes laterally to form the lateral wing of the quadrate shaft. Just above the level of the ventral margin of the pterygoid wing, there is a dorsoventrally large, shallow embayment in the lateral wing that probably represents the medial margin of the quadrate foramen (Fig. 9). If so, the quadrate foramen was large and

clearly incised into the lateral quadrate wing, as in oviraptorosaurs (*Maryanska & Osmólska, 1997*) and many other theropods. The mandibular condyle is transversely expanded and has a low, rounded medial portion that is offset from an oblique ridge that expands from anteromedially posterolaterally by a shallow and broad concavity, similar to the situation in oviraptorids (*Maryanska & Osmólska, 1997*). The condyle extends slightly more ventrally laterally than medially (Fig. 9).

Of the palate, the left pterygoid and palate are visible through the orbit and antorbital fenestra. The **pterygoid** is shifted anterodorsally and exposed in dorsal view, with the quadrate wing being collapsed into the plane of the ectopterygoid wing and being largely overlain by the frontal (Figs. 7 and 9). The ectopterygoid wing is tongue-shaped and anteroposteriorly slender, although it is more expanded than in allosauroids (*Madsen, 1976*; *Eddy & Clarke, 2011*). It extends far laterally and is considerably larger than reconstructed by *Elzanowski & Wellnhofer (1996)*. Its anterior margin is gently curved, whereas the posterior margin is straight and slightly thickened. From the junction of the ectopterygoid wing with the quadrate wing, a long and slender anterior (palatal) process extends anteriorly and disappears under the dorsal skull roof at the anterodorsal end of the orbit. The medial margin of this process is slightly thickened and raised as a distinct dorsal ridge (Fig. 9), as in *Deinonychus* (*Ostrom, 1969*). The quadrate wing of the pterygoid was large and anteroposteriorly expanded, as in most theropods. Together with the ectopterygoid wing it encloses an obviously narrow facet for the articulation with the basipterygoid process of the braincase (Fig. 9).

The left **palatine** is visible in dorsolateral view through the antorbital fenestra and the anterior part of the orbit (Figs. 7 and 8). It is an elongate, slender element, as in the Munich specimen (*Wellnhofer, 1993*; *Elzanowski & Wellnhofer, 1996*), but misses most of the vomerine process. On the basis of the isolated left palatine of the Munich specimen, *Elzanowski & Wellnhofer (1996)* argued that the palatine in *Archaeopteryx* is triradiate, as in birds, with the lateral ramus, which contacted the maxilla, being almost confluent laterally with the pterygoid wing. This is unlike the situation seen in basal theropod dinosaurs, which usually have a posteriorly pointing jugal process on the lateral ramus, resulting in a tetraradiate palatine. In contrast, *Mayr et al. (2007)* identified such a process, though shorter and stouter than in most non-avialan theropods, which contacted the jugal in the Thermopolis specimen. The new specimen seems to show a somewhat intermediate condition between that reconstructed for the Munich specimen by *Elzanowski & Wellnhofer (1996)* and the process illustrated by *Mayr et al. (2007)*. As in the Munich specimen, the maxillary process is slender, elongate and tapers anteriorly (Fig. 8), which contrasts with the broader process in the Thermopolis specimen (*Mayr et al., 2007*). However, the posterior part of the lateral ramus, which is slightly disarticulated from the maxilla and jugal, is thickened and forms a bluntly rounded posterolateral edge that is clearly offset from the pterygoid wing of the palatine, although a posteriorly pointing jugal process is absent (Fig. 8). A depression is present on the posterior part of the dorsal surface of the lateral ramus, as in the Munich and Thermopolis specimens, and it is mainly defined here by the dorsally raised lateral and posterior margin of the lateral ramus. Medially, this depression seems to be offset from another, deeper depression on the base of
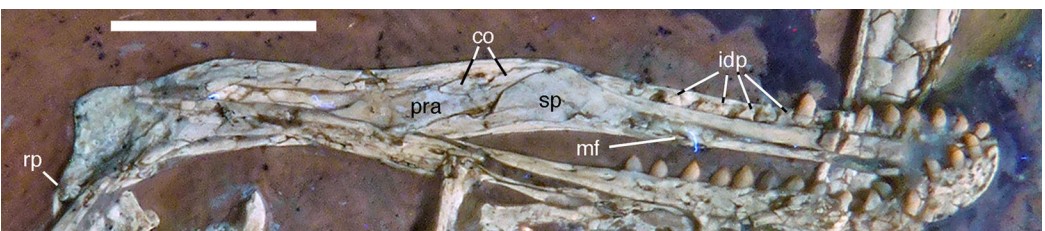

**Figure 12** **Left mandible of the 12th specimen of *Archaeopteryx* in medial view.** Photograph under UV light. co, coronoid; idp, interdental plates; mf, myliohyoid foramen; pra, prearticular; rp, retro-articular process; sp, splenial. Scale bar is 10 mm.

the pterygoid wing by a transverse ridge, as in the Munich and Thermopolis specimens, but the medial part of the dorsolateral side is here overlain by an unidentified bone fragment. This depression extends over the entire length of the dorsolateral side of the pterygoid wing, but becomes shallower posteriorly. The pterygoid wing is slender, tongue-shaped and strongly elongate (Figs. 7 and 8), being almost twice the length (9.4 mm) of the lateral ramus (5 mm), as in the Munich specimen, but unlike the situation in non avialan theropods, which usually have a rather short (as long as or shorter than the lateral wing) and often broad pterygoid wing (*Ostrom, 1969*; *Madsen, 1976*; *Currie & Zhao, 1993a*; *Barsbold & Osmólska, 1999*; *Currie, 2003*; *Lautenschlager et al., 2014*). Information on the palate of Mesozoic birds is limited, but the pterygoid wing is slender and elongated in the enantiornithine *Gobipteryx* (*Chiappe, Norell & Clark, 2001*) and the hesperornithiform *Hesperornis* (*Elzanowski, 1991*), so this might be an avialan synapomorphy. Nothing can be said about the morphology of the vomerine process in the new specimen, as it is largely broken and its remnants are hidden by an unidentified bone fragment.

## Mandible

Both mandibles are preserved in articulation, but compressed and therefore largely crushed. Due to the compression, the right mandible is exposed in lateral view, whereas the left is seen in medial view (Figs. 6 and 7). The better exposed, complete left mandible has a total length of 45.5 mm (Fig. 12). The mandible largely corresponds to the morphology seen in other specimens, such as the Eichstätt (*Wellnhofer, 1974*) and Munich specimens (*Wellnhofer, 1993*; *Elzanowski & Wellnhofer, 1996*), so only a few additional comments will be offered here.

The fact that the **dentaries** are preserved with the symphysis in articulation, despite the compression and slight disarticulation in other parts of the skull (Fig. 7), is in accordance with the notion that the symphysial syndesmosis was rather strong in *Archaeopteryx* (*Elzanowski & Wellnhofer, 1996*), but remained unfused, as in *Confuciusornis* (*Chiappe et al., 1999*). However, the mandible of the Munich specimen, which is preserved in medial view, shows that the symphysis is small and smooth (*Holliday & Nesbitt, 2013*). As in other specimens, the dentaries are long and slender, being considerably lower in dorsoventral height than the postdentary portion of the mandible. There are 13 tooth positions in the left dentary, which is one more than in the Munich specimen, the only other specimen in

which the dentary tooth count can be established with certainty (*Elzanowski & Wellnhofer, 1996*). For the Eichstätt specimen, *Wellnhofer (1974)* even assumed the presence of only 11 dentary teeth; given that these three specimens are of consecutively larger size (length of the mandible 36.5 mm in the Eichstätt specimen, 40 mm in the Munich specimen and 45.5 mm in the new specimen), it is possible that there was a gradual increase in dentary tooth number in *Archaeopteryx* during ontogeny, as assumed for allosauroids (*Rauhut & Fechner, 2005*), therizinosaurids (*Kundrát et al., 2008*) and *Byronosaurus* (*Bever & Norell, 2009*), although individual variation cannot be ruled out. Dentary teeth are widely spaced, with the spacing between teeth increasing in the posterior part of the dentary, as in the Munich specimen (BSPG 1999 I 50). Thus, whereas the space between individual teeth in the anterior half is approximately half of the mesiodistal width of the individual teeth, it is subequal to this width in the posterior part (Fig. 12). In contrast, dentary teeth seem to be more narrowly spaced in the 11th specimen (*Foth, Tischlinger & Rauhut, 2014*). The first dentary tooth is offset from the anterior end of the bone by approximately one tooth width, as in the premaxilla.

The anterior end of the dentary is gently rounded, as in the Berlin (*Tischlinger, 2005*; *Wellnhofer, 2008*) and Eichstätt specimens (*Wellnhofer, 2008*), but unlike the straight anterior margin figured for the Solnhofen (*Wellnhofer, 1992*, *2008*) and Munich specimens (*Elzanowski & Wellnhofer, 1996*). However, the latter interpretations might be erroneous due to poor preservation (Solnhofen specimen) and the articulated mandibular symphysis being exposed in medial view (Munich specimen), respectively. Several nutrient foramina are present in the anterior end of the dentary, apparently arranged in two rows, one along the alveolar margin, and a second following the curvature of the anterior end and continuing posteriorly along the ventral margin, though slightly diverging from the latter posteriorly. Unfortunately, it cannot be said how far this ventral row extended posteriorly, as only the anteriormost 5 mm of the right dentary are sufficiently well preserved to see this row. In the posterior part of the dentary, the alveolar row of foramina is placed in a narrow and shallow groove in the dorsal third of the bone, as in the Eichstätt specimen (*Wellnhofer, 2008*), *Microraptor* (*Pei et al., 2014*) and *Anchiornis* (*Pei et al., 2017*). A narrow, but deep Meckelian groove is present on the ventral part of the medial side of the dentary in its anterior half (Fig. 12), as in the Munich specimen, but, in contrast to the reconstruction of the latter by *Elzanowski & Wellnhofer (1996)*, who figured it as of subequal width up to the 11th dentary tooth, it gradually expands dorsoventrally in its posterior part from the ninth tooth position onwards; this expansion seems to be even more anterior in the 11th specimen (*Foth, Tischlinger & Rauhut, 2014*). In the posterior part of the dentary, the Meckelian groove is overlapped medially by the splenial.

Above the Meckelian groove, the medial side of the dentary is slightly dorsoventrally convex between the groove and the paradental lamina. The latter is broad and placed notably ventral to the lateral alveolar margin, forming the medial wall of the alveoli. In between individual dentary teeth, low, triangular to polygonal interdental plates are present, but these plates are widely separated, exposing the medial side of the upper part of the roots of the teeth above the paradental lamina (Fig. 12). Towards its posterior end,

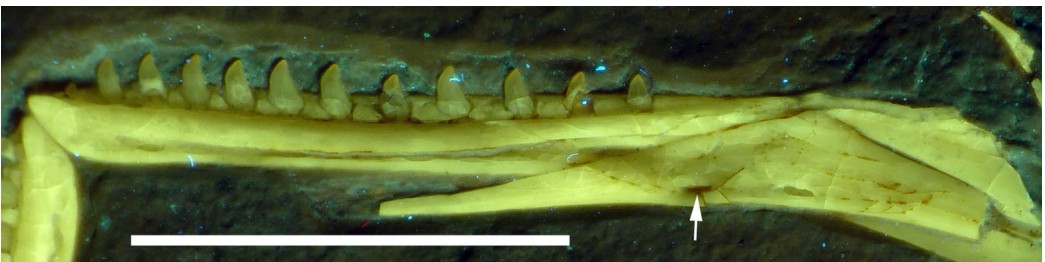

**Figure 13 Right mandibular ramus of the Munich specimen of *Archaeopteryx* (SNSB BSPG 1999 I 50).** Photograph under UV light, showing myliohyoid foramen in the splenial (arrow). Scale bar is 10 mm.

the dentary becomes high and plate-like, but this part is poorly preserved in the right and largely covered by other elements in the left side.

The **splenial** is a large, anteroposteriorly elongate triangular element that covers the medial side of the posterior half of the dentary and extends posteriorly below the prearticular (Figs. 7 and 12), as in the Munich specimen (*Wellnhofer, 1993*; *Elzanowski & Wellnhofer, 1996*). However, the anterior ramus seems to be shorter than in the latter, and tapers to a point below the tenth tooth. In the Munich specimen, the splenial reaches more anteriorly, approximately to the seventh dentary tooth. The highest part of the splenial reaches the level of the dorsal margin of the lateral side of the mandible at the posterior end of the dentary. A large myliohyolid foramen seems to have been present in the ventral part of the splenial below the last dentary tooth, but most of its ventral and anterior margins are broken. No myliohyoid foramen was illustrated for the Munich specimen by either *Wellnhofer (1993)* or *Elzanowski & Wellnhofer (1996)*. However, this opening is clearly visible in UV-photographs of this specimen, but placed slightly more posteriorly, posterior to the last dentary tooth position (Fig. 13). In contrast to dromaeosaurids (*Rauhut, 2003*), the foramen is closed ventrally and anteroposteriorly elongate, as in the troodontid *Zanabazar* (*Norell et al., 2009*).

A **coronoid** was said to be absent in *Archaeopteryx* (*Elzanowski & Wellnhofer, 1996*), but a thin, anteroposteriorly elongated element posterior to the apex of the splenial and dorsal to the anterior end of the prearticular in the new specimen obviously represents this element (Figs. 7 and 12). The bone is tongue-shaped, being higher posteriorly, and seems to taper to a point anteriorly, unlike the triangular coronoid in more basal theropods (*Brochu, 2003*). The elongate shape is similar to the bone identified as coronoid in *Ichthyornis* (*Clarke, 2004*), indicating that the reduction of this element, or its fusion with other mandibular elements happened later and, possibly, several times independently in avialan evolution (*contra Elzanowski & Wellnhofer, 1996*).

Only the anterior end of the **prearticular** is visible, as the posterior part of the left mandible is partially hidden by the jugal (Fig. 12). The anterior end of the bone is rather slender and spatulate and extends from posteroventral anterodorsally towards the dorsal rim of the mandible. It is overlapped dorsally by the coronoid and anteriorly by the posterodorsal margin of the splenial. A small break in the latter margin shows

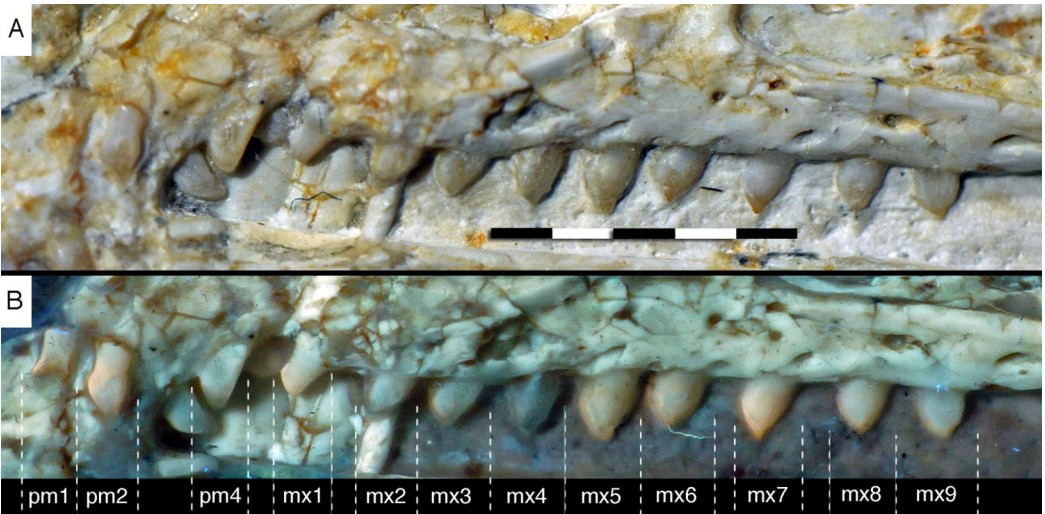

**Figure 14 Dentition of the upper jaw of the 12th specimen of *Archaeopteryx.*** (A) Photograph under normal light. (B) Photograph under UV light. mx, maxillary teeth; pm, premaxillary teeth; numbers denominate tooth positions. Scale bar is in mm.

that the anterior border of the prearticular flexes slightly laterally to form a facet for the splenial.

The **surangular** forms most of the dorsal border of the mandible posterior to the apex of the splenial. This dorsal border is slightly thickened anteriorly, where its medial edge borders the dorsal margin of the coronoid. Posteriorly, it becomes progressively more thickend, until it forms a medial shelf anterior to the mandibular articulation. The dorsomedial surface of this shelf exhibits a weak concavity anterior to the glenoid, which is laterally bordered by a raised ridge. The medial side of the surangular forms the lateral wall of the mandibular fossa, but most of its ventral portions are hidden by the prearticular and jugal, and so is the contact to the angular.

The suture between the surangular and the articular is unclear. The glenoid region is poorly preserved, so nothing can be said about its detailed morphology. The retroarticular process is short and directed posteroventrally (Fig. 12), as in the Munich specimen of *Archaeopteryx* (*Wellnhofer, 1993*). The attachment area for the *m. depressor mandibulae* is thus dorsoventrally concave and more posteriorly than dorsally directed.

## Dentition

As noted above, there are probably four tooth positions in the premaxilla, nine in the maxilla and 13 in the dentary. The number of teeth is thus rather high in comparison with other specimens of *Archaeopteryx*; nine maxillary teeth are otherwise only (probably) present in the Thermopolis specimen (*Mayr et al., 2007*), whereas the Berlin and Eichstätt specimen have only eight (*Howgate, 1984a*; *Wellnhofer, 2008*, *2009*). Likewise, as noted above, 13 teeth in the dentary is the highest number yet recorded (see discussion below).

The first two premaxillary teeth are slender. The second tooth is well preserved, whereas only an impression of the apical part of the crown of the first tooth is present (Fig. 14).

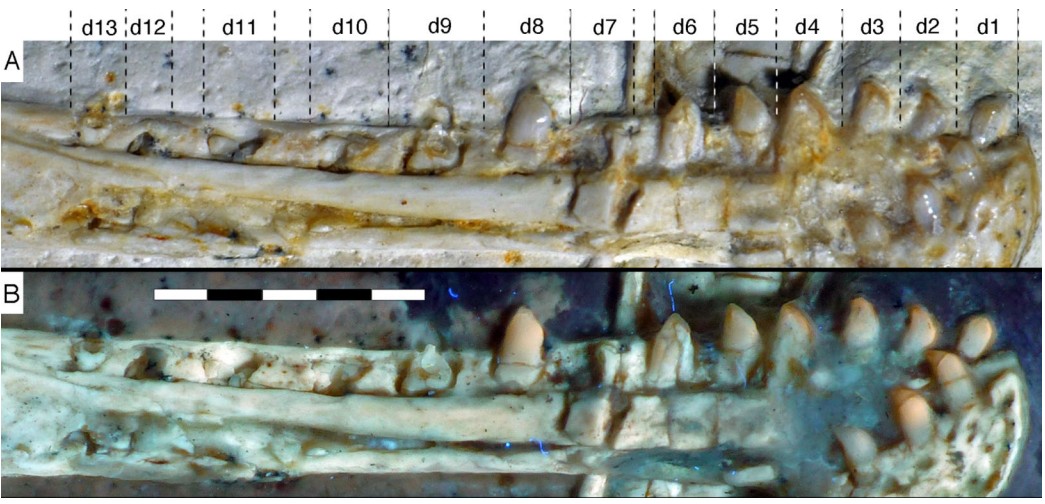

**Figure 15 Dentition of the lower jaw of the 12th specimen of *Archaeopteryx*.** (A) Photograph under normal light. (B) Photograph under UV light. d, dentary tooth position; numbers denominate tooth positions. Scale bar is in mm.               

However, this impression indicates that both teeth were closely comparable in morphology. There is a constriction of the tooth crown at about one-third of its height. Apical to this constriction, the crown expands both mesially and distally and is strongly convex mesiodistally. In the apical third of the crown, the tip is backturned, so that the anterior margin of the crown is straight and the posterior margin slightly concave. However, in contrast to most non-avialan theropods, the tip of the crown is placed approximately directly apical to the posterior margin of its base and not distal to it. The last premaxillary tooth is slightly more robust and seems otherwise to be generally similar, but is poorly preserved.

The maxillary teeth are slightly procumbent, at least to the seventh tooth (Fig. 14). The first maxillary tooth is similar to the premaxillary teeth, but the expansion of the central part of the crown is less pronounced, especially distally, where the distal margin is straight to very slightly concave over its entire length. From the second maxillary tooth onwards, the crowns become more massive, being only slightly higher apicobasally than long mesiodistally (Fig. 14). Both the medial and distal margins are strongly convex over almost their entire length, with a small concavity being present only in the apical third of the distal margin. The crowns are strongly convex mesiodistally. A pointed tip extends apically from the bulbous body of the crown and is placed approximately above the mesiodistal mid-length of the tooth. The fourth to sixth maxillary teeth are the most massive elements and also seem to be slightly more narrowly spaced than the more mesial and especially the more distal teeth.

Dentary tooth variability largely mirrors that of the teeth of the upper tooth row, but only one more distal tooth (eighth) is currently preserved in the left mandibular ramus (Fig. 15), and three of the probably most posterior teeth of the right ramus are visible (Fig. 16). The anterior dentary teeth are slender, slightly waisted in the basal third of the crown and have a recurved tip. However, in comparison to the premaxillary teeth, the expansion of the apical part of the crown is less pronounced, and the tip more strongly

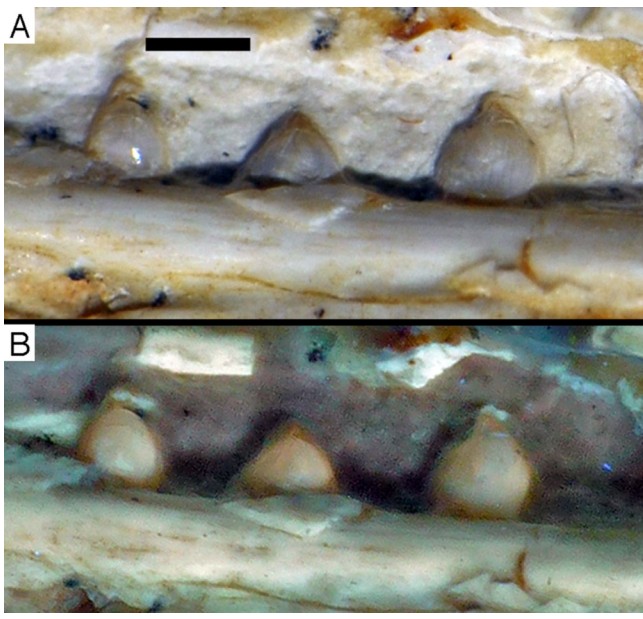

**Figure 16 Preserved teeth in the posterior portion of the right lower jaw of the 12th specimen of _Archaeopteryx._** (A) Photograph under normal light. (B) Photograph under UV light.

recurved, placing the apex slightly distal to the base of the tooth (Fig. 15). This tooth morphology is retained to at least the fifth tooth position, with the teeth becoming slightly more robust towards the distal positions. The transition between these anterior teeth and the robust, bulbous posterior teeth seems to be more gradual than in the upper jaw, but the eighth dentary tooth is closely comparable in shape to the mid-maxillary crowns. In this tooth, low, unserrated carinae are present mesially and distally as in the Munich specimen (_Wellnhofer, 1993_; _Weigert, 1995_). In contrast to the Munich specimen, the carinae of the 12th specimen do not extend to the apex of the tooth, but end a short way below it, and the apex is placed slightly labial to them. Translucent areas along the margins of some of the maxillary teeth indicate that similar carinae might be present here as well. A part of the root of the eighth dentary tooth is exposed and shows a longitudinal depression along its mid-width. The distalmost crowns of the right mandibular ramus are very low, anteroposteriorly massive, and have a small, pointed apical tip (Fig. 16).

## Postcranium

Although much of the postcranial skeleton is preserved in articulation, several areas were affected by breakage and loss of elements prior to or at the time of discovery (Fig. 5). This includes the presacral vertebral column, the pelvis, and the forelimbs. The skull is slightly disarticulated from the vertebral column and much of the anterior part of the cervical vertebral column is missing or hidden by the cranium. Both shoulder girdles and forelimbs are preserved in articulation, but have been disarticulated from the rest of the skeleton and moved slightly away from the vertebral column.

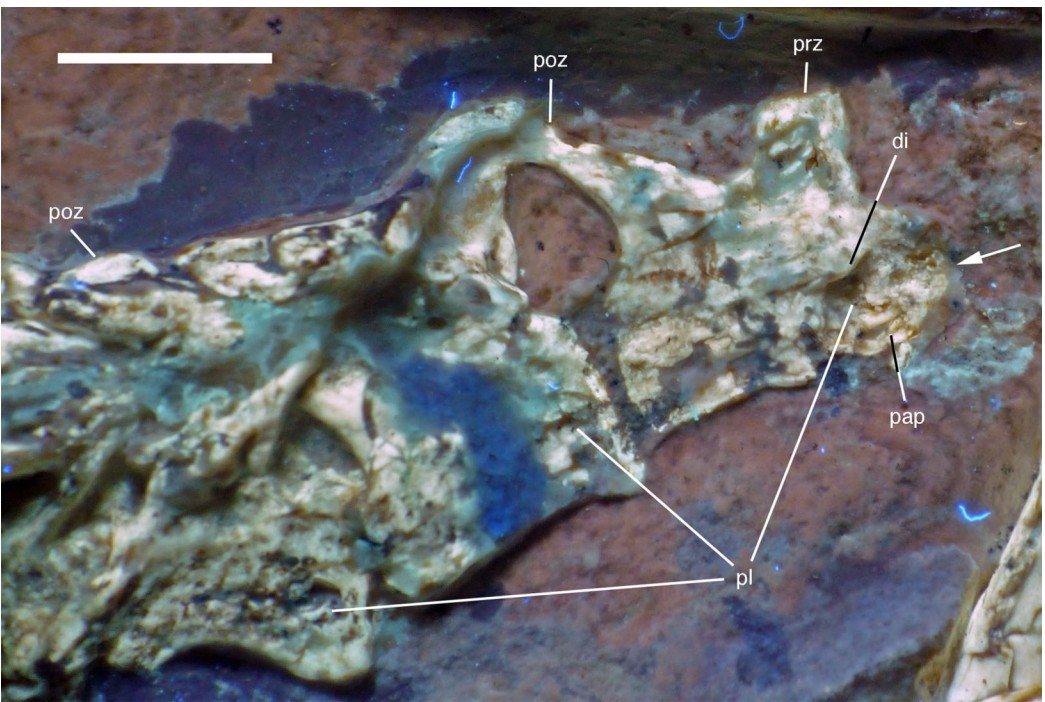

**Figure 17 Preserved last cervical and first dorsal vertebrae of the 12th specimen of *Archaeopteryx*.** Photograph under UV light. di, diapophysis; pap, parapophysis; pl, pleurocoel; poz, postzygapophysis; prz, prezygapophysis. Arrow points to strongly convex anterior articular end. Scale bar is 5 mm.

### Axial skeleton

The neck was obviously disrupted by the left forelimb, so that the eighth **cervical vertebra** (assuming a cervical count of nine vertebrae, as in other specimens of *Archaeopteryx*; *Wellnhofer, 2008, 2009*) is disarticulated from the more anterior elements (Fig. 5). Anterior to this vertebra, only one cervical is recognisable on the other side of the left humerus, close to the skull. However, this vertebra (presumably a posterior mid-cervical) is very poorly preserved, and most of the lateral wall of the element is missing, so that the interior of the vertebra is exposed (Fig. 6B). The vertebral centrum is elongate (more than 7 mm long at an anterior centrum height of c. 2.4 mm) and seems to be at least slightly convex anteriorly, although the articular surface is too poorly preserved to determine this with certainty. The prezygapophysis is marked as a slender process that overhangs the vertebral centrum anteriorly and is flexed ventrally in its anterior part. Both the vertebral centrum and neural arch are strongly pneumatised as found in the Berlin specimen (*Britt et al., 1998*); in the case of the neural arch small cavities extend even into the stalk of the prezygapophysis (Fig. 6B).

The last two cervical vertebrae are preserved in articulation with the dorsal vertebral column (Fig. 17).

The length decreases rapidly from the eighth (8.1 mm) to the ninth cervical (6.1 mm). The more anterior of these two cervicals has an elongate centrum that is approximately 2.8 times as long as its anterior height. The centrum is only slightly constricted in lateral

view, and the articular ends are not offset from one another. The anterior end of the centrum is strongly convex, indicating an opisthocoelous condition for at least the posterior cervical vertebrae of this specimen (Fig. 17). *Dames (1884)* assumed an amphicoelous ('biconcave') condition for the cervical vertebrae of the Berlin specimen of *Archaeopteryx*, although he noted that this condition cannot be established with certainty. *Wellnhofer (2008, 2009)* considered the vertebrae to be platycoelous in this specimen. However, as the cervical vertebrae of this specimen are preserved in articulation, the condition is difficult to assess, and this is also true for the Eichstätt specimen, which also preserves a complete cervical vertebral column (*Wellnhofer, 1974, 2008*). In the 11th specimen, the neck is also preserved in articulation (*Foth, Tischlinger & Rauhut, 2014*), but at the strongest flexure of the neck, the anterior end of the seventh cervical is slightly disarticulated from the sixth element, and also seems to be at least slightly convex. In only slightly more dervied avialan taxa, such as *Confuciusornis* (*Chiappe et al., 1999*) and *Sapeornis* (*Zhou & Zhang, 2003b*), the cervical vertebrae are already hetercoelous; the condition is currently unknown in *Jeholornis* (*Zhou & Zhang, 2003a*; *O'Connor et al., 2012*; *Lefèvre et al., 2014*).

In the anterior end of the centrum of the eighth cervical, the lateral wall of the bone is missing, so the presence of a pneumatic foramen can only be assumed by the presence of a cavity in this area. However, the centrum is clearly strongly pneumatized, with the preserved structures of the broken anterior end indicating a camellate structure, as present in *Aerosteon*, carcharodontosaurids, ornithomimosaurs, tyrannosaurids, oviraptorosaurs, and modern birds (*Britt, 1993*; *Sereno et al., 2008*; *Evers et al., 2015*). The neural arch of this vertebra is poorly preserved, but the postzygapophysis seems to overhang the centrum posteriorly, while the prezygapophysis seems to be placed entirely above the anterior end of the centrum. The probably last cervical vertebra is poorly preserved, and little can be said about its morphology (Fig. 17). The anterior end of the centrum seems to be flat, but this might be an artifact of preservation. The neural arch is high, being approximately as high as the centrum.

In contrast to most other theropods, *Archaeopteryx* (including the new specimen) possesses 14 **dorsal vertebrae**. The first dorsal vertebra is slightly shorter (5.3 mm) than the last cervical vertebra. As in other specimens of *Archaeopteryx* (*Britt et al., 1998*; *Christiansen & Bonde, 2000*), it has a well-developed pneumatic foramen on the anteroventral side of the centrum (Fig. 17). This foramen seems to be placed below the parapophysis, which also seems to be the case in the 11th specimen. A ventral keel or hypapophysis is absent. As in the last cervical vertebra, the neural arch is anteroposteriorly short and high.

The remaining dorsal vertebrae are poorly preserved and/or overlapped by the heads of the dorsal ribs, and little can be said about their morphology (Fig. 18). The posteriormost dorsal and the sacral vertebrae are largely missing due to a break going through the slab here, so that details of the transition between the two vertebral regions cannot be studied. The centra and parts of the neural arches of the 7th to 10th dorsal vertebrae are exposed (Fig. 19). The centra are amphi- to platycoelous and elongate, the eighth dorsal being c. 6.2 mm long and 4 mm high posteriorly. The centra are evenly constricted

**Figure 18 Articulated dorsal vertebral column of the 12th specimen of *Archaeopteryx*, including dorsal ribs and gastralia.** Photograph under UV light. Scale bar is 10 mm.

ventrally in lateral view and have a flattened ventral surface. No pneumatic foramina are present in these centra, but a large, shallow, anteroposteriorly elongate depression is present on the dorsal part of the lateral side. A part of the lateral lamination of the neural arch is preserved in the eighth dorsal (Fig. 19). The anteroposteriorly extensive transverse process is supported ventrally by a short, but stout and vertically oriented posterior centrodiapophyseal lamina (pcdl), which is placed above the posterior third of the centrum. Anteriorly, a well-developed, though more slender paradiapophyseal lamina (ppdl) extends posterodorsally at an angle of approximately 45° in relation to the long axis of the centrum and meets the base of the transverse process well in front of the pcdl, so that there is a large centrodiapophyseal fossa in between these laminae that is roofed dorsally by the transverse process. A well-developed prezygodiapophyseal lamina (prdl) is present anterior to the point where the ppdl meets the transverse process. Anterior to the ppdl, but originating ventrally at approximately the same point as the latter, there is a thin, but well-developed prezygoparapophyseal lamina (prpl), which extends anterodorsally at an angle of slightly more than 90° towards the ppdl and reaches the anterior end of the prezygapophysis anteriorly (Fig. 19). The prezygapophysis slightly overhangs the vertebral centrum anteriorly. The neural spines of the mid-dorsal vertebrae are anteroposteriorly long (70–75% of centrum length in D9), rectangular in outline, and slightly lower dorsoventrally than long anteroposteriorly (Figs. 18 and 19). They are placed over the posterior part of the centrum, so that their posterior margin is

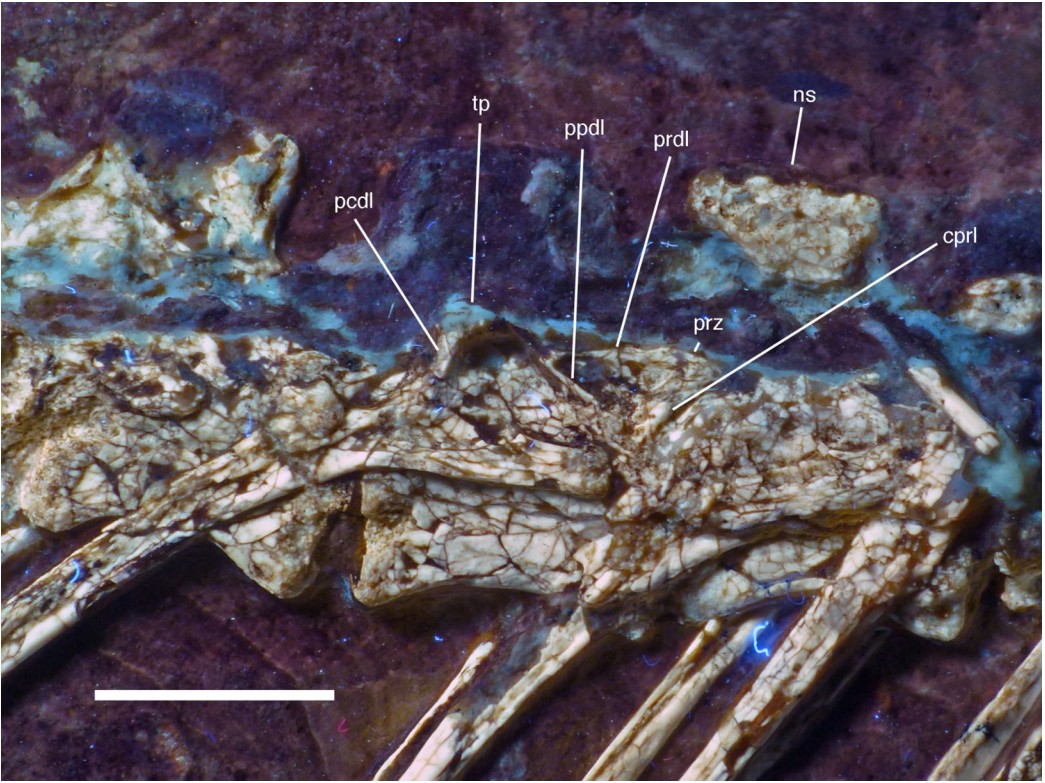

**Figure 19 Dorsal vertebrae seven to nine of the 12th specimen of *Archaeopteryx*.** Photograph under UV light. cprl, centroprezygapophyseal lamina; ns, neural spine of the 7th dorsal; pcdl, posterior centrodiapophyseal lamina; ppdl, paradiapophyseal lamina; prdl, prezygodiapophyseal lamina; prz, prezygapophysis; tp, transverse process. Scale bar is 5 mm.

approximately flush with the posterior end of the latter. The postzygapophysis extends posterior to the base of the neural spine and thus overhangs the centrum for its entire length. Its dorsal margin is connected to the neural spine by a short, curved spinopostzygapophyseal lamina (spol).

Because the sacral region is highly damaged the number of **sacral vertebrae** cannot be estimated. Only a few fragments of the neural arches of the sacral vertebrae are discernible. The neural spines of the sacral vertebrae were obviously similar to those of the dorsal vertebrae and are well separated (Fig. 18).

Due to the damage in the sacral region, the proximalmost **caudal vertebrae** are very poorly preserved and mainly indicated by impressions. Thus, the transition between the sacral and caudal vertebrae is difficult to establish, but we take a rather abrupt change in length of the vertebrae to indicate this transition. Based on this assumption, there are 22 caudal vertebrae preserved (Fig. 20A), as in the Eichstätt (*Wellnhofer, 1974*, *2008*, *2009*) and 11th specimen. However, it cannot be ruled out that a very small 23rd vertebra might originally have been present posterior to the last preserved element. When compared, *Eosinopteryx* possesses 20 caudals (*Godefroit et al., 2013a*), *Anchiornis* has approximately 31–32 vertebrae (*Pei et al., 2017*), while the caudal series of *Jeholornis* varies between 22 and 27 vertebrae (*Zhou & Zhang, 2002*, *2003a*; *O'Connor et al., 2012*).

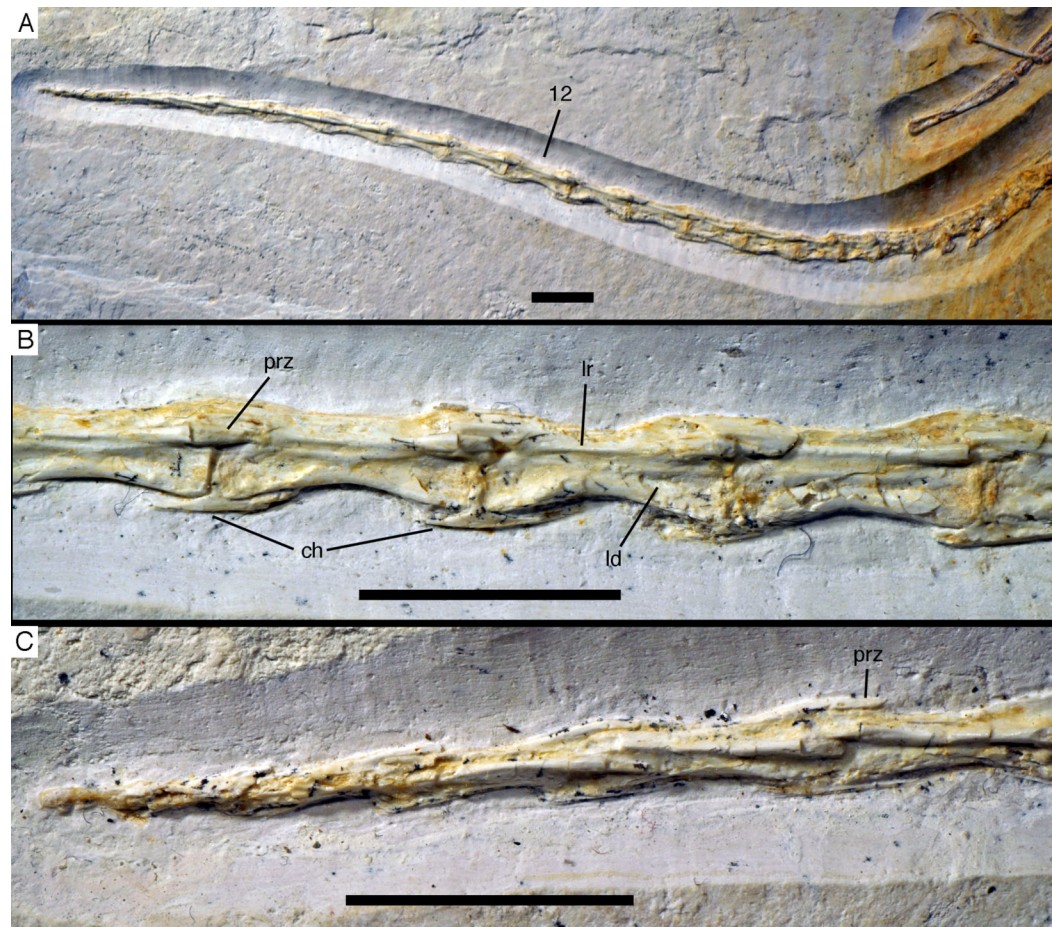

**Figure 20 Caudal vertebrae of the 12th specimen of *Archaeopteryx*.** (A) Complete caudal vertebral column as preserved in right lateral view. (B) Close-up of ninth (right, incomplete) to 13th caudal vertebra. (C) Distal end of tail. 12, denominates the 12th caudal vertebra as reference point; ch, chevron; ld, lateral depression; lr, lateral ridge; prz, prezygapophysis. Scale bars are 10 mm.

The proximal caudals have rather short centra, as in *Jeholornis* (*O'Connor et al., 2012*), *Anchiornis* (*Xu et al., 2009a*), and several other paravians, but nothing can be said about their morphology. The length of the caudal vertebrae increases rapidly from 6.2 mm in the sixth to 9.7 mm in the ninth vertebra. The longest caudal vertebrae are the 11th and 12th elements, which measure 10.1 mm each and in which the vertebral centrum is more than five times as long as high, being thus more elongate than in *Jeholornis* (*O'Connor et al., 2012*). From here, caudal vertebral length decreases gradually to 8.7 mm in the 18th caudal, and then rapidly to 2.4 mm in the 22nd vertebra (Fig. 20C). Vertebral centra are generally spool-shaped and only slightly constricted in the middle in lateral view (Fig. 20). A shallow longitudinal depression is present on the lateral side in the middle and distal caudal vertebrae (at least to caudal 17) and become more pronounced towards the articular ends (Fig. 20B). These depressions might be slightly exaggerated by compression, as a damaged mid-caudal vertebra shows that the elements were hollow. However, similar lateral depressions have also been noted in *Anchiornis* (*Hu et al., 2009*).

The anterior mid-caudals (e.g. CA6) have short and stout prezygapophyses that point anterodorsally at an angle of slightly less than 45° towards the long axis of the centrum and do not overhang the centrum anteriorly. In contrast, the postzygapophyses are slender, point more posteriorly than dorsally and overhang the centrum posteriorly. From the eighth caudal onwards, the prezygapophyses become more slender, point more anteriorly than dorsally, and overhang the centrum anteriorly. In the distal caudal vertebrae, the prezygapophyses are developed as thin, anteriorly pointing processes that overlap the posterior end of the preceeding centrum. However, in contrast to many tetanuran theropods, in which this overlap often accounts for one-third or more of the length of the centrum, it reaches maximally 24% in this specimen of *Archaeopteryx*. The postzygapophyses form a broad, posteriorly pointing and posteriorly forked platform in the middle and distal caudals that overhangs the following centrum approximately as much as the prezygapophysis. From the seventh vertebra onwards, the pre- and postzygapophyses are connected by a pronounced longitudinal ridge on the laterodorsal side of the neural arch (Fig. 20). A small, apparently triangular transverse process is present at about mid-length of the centrum in the sixth caudal vertebra. This transverse process is transversely short and placed just below the mid-height of the centrum. Transverse processes are also still present in the seventh and eighth caudal, where it is only a small triangular flange of bone. These processes seem to be completely absent from the ninth caudal onwards.

The neural spine is first visible in the eighth caudal vertebra, where it is developed as a low, dorsally straight dorsal ridge that extends over almost the entire length of the vertebra, but is maximally as high as the dorsal edges of the zygapophyses. The last vertebra with a spine is the 10th caudal; posterior to this element, the dorsal surface of the neural arch is flat.

Several **dorsal ribs** are at least partially preserved. As in all dinosaurs, the rib head is two-headed (Fig. 18). Where the rib heads are preserved, in the mid-dorsal section, they have a long and slender capitulum that is continuous with the proximal curvature of the rib shaft, and a much shorter, dorsally pointing tuberculum (Fig. 19). As in other specimens of *Archaeopteryx* (*Wellnhofer, 1974*), the first dorsal rib is short and spike-like. The anterior dorsal (thoracic) ribs have long shafts that are almost straight proximally, but curve strongly medially in their distal half (Fig. 18). This morphology is also found in the Eichstätt (*Wellnhofer, 1974*), Solnhofen (*Wellnhofer, 1992*) and 11th specimen, whereas the ribs are only gently and more gradually curved in the Berlin and Munich specimens (*Wellnhofer, 1993*, *2008*, *2009*). As all specimens are flattened into two dimensions, it is unclear in how far these differences might be due to compression and differences in the orientation of the ribs during burial and which of these morphologies represents the original condition. However that may be, the more posterior ribs of the 12th specimen are more gradually curved. Many of the rib shafts seem to have a narrow and shallow longitudinal furrow anterolaterally (Fig. 18). This furrow is also present in the 11th specimen, *Microraptor* (*Pei et al., 2014*) and *Anchiornis* (*Pei et al., 2017*). Uncinate processes, as they are present in *Microraptor* (*Hwang et al., 2002*), some other dromaeosaurids (*Norell & Makovicky, 1999*) and oviraptorids (*Clark, Norell & Chiappe,*

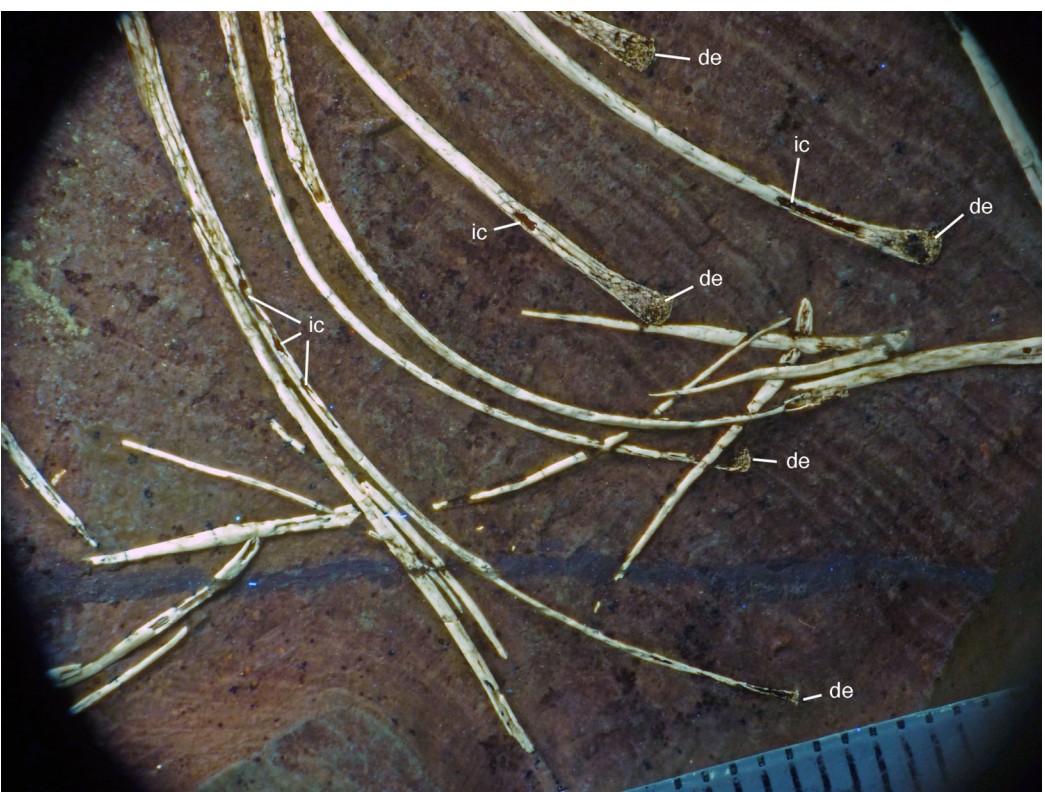

**Figure 21 Distal ends of anterior dorsal ribs and anterior gastralia of the 12th specimen of _Archaeopteryx._** Photograph under UV light. de, distal expansion; ic, internal cavity revealed by breaks. Scale bar in 1 mm increments.

_1999_), are absent in this and all other specimens of _Archaeopteryx._ Several broken sections of rib shafts indicate that these bones were hollow (Fig. 21). As in several other specimens of _Archaeopteryx_ (_Wellnhofer, 2008_, _2009_), the distal end of several of the anterior dorsal ribs is notably thickened (Figs. 18 and 21), indicating the attachment of cartilagenous sternal ribs (see _Foth, 2014_).

The anterior **caudal chevrons** are poorly preserved, but the element between the fourth and fifth caudal vertebra is dorsoventrally short and expanded distally and thus corresponds to the morphology figured by _Wellnhofer (2008)_. Posterior to this element, a recognisable chevron is first present between the 9th and 10th caudal vertebrae (Fig. 20B). This chevron is skid-like, with a longer anterior than posterior process. The anterior process is more slender than the posterior process and tapers to a point, whereas the posterior process is rounded posteroventrally. The contact to the chevron facets of the vertebrae is developed as a small, triangular flange. The anteroposterior length of this chevron is approximately 5.5 mm (3.8 mm anterior and 1.7 mm posterior process), whereas its dorsoventral height is only 1 mm. More posteriorly placed chevrons are generally similar in morphology, but more symmetrical (Fig. 20B). The last chevron seems to be present between the 21st and 22nd caudal (Fig. 20C). In contrast to _Jeholornis_ (_O'Connor et al., 2012_) and dromaeosaurids (_Ostrom, 1969_; _Norell & Makovicky, 1999_), the adjacent chevrons do not overlap.

**Gastral ribs** are present in the thoracic region and all the way to the pubis in the abdominal region (Figs. 4, 5 and 18), as in the Eichstätt and Munich specimens of *Archaeopteryx* (*Wellnhofer, 2008*, *2009*). The anterior elements are more robust than posterior elements and the medial elements are more robust than lateral gastralia (Fig. 21). Whereas the latter remain of subequal width throughout their length, the medial gastralia taper laterally. In contrast to the situation in *Velociraptor* (*Norell & Makovicky, 1997*), *Sinornithoides* (*Russell & Dong, 1993*) and *Linheraptor* (*Xu et al., 2010*), but similar to the condition in *Microraptor* (*Pei et al., 2014*), the lateral elements seem to be subequal in length to the medial elements.

### Appendicular skeleton

As noted above, the forelimbs are disarticulated from the rest of the body, but kept their natural articulation between the forelimb elements, especially in the left limb. Unfortunately, however, a break ran through the right shoulder girdle and forelimb when the specimen was discovered, and parts of the left forelimb were lost due to erosion or when the slab was split originally. Furthermore, all limb bones are strongly compressed and cracked, and their preservation is thus rather poor.

As in most other specimens of *Archaeopteryx*, the shoulder girdle comprises the scapula, coracoid and furcula (Fig. 22), but an ossified sternum is not present, as it is the case in most non-pennaraptoran theropods, *Anchiornis, Mei, Jianianhualong,* and *Sapeornis* (*Zheng et al., 2014*; *Foth, 2014*; *Xu et al., 2017*). The scapulocoracoid is described in reference to its original position, with the scapula being oriented more or less parallel to the vertebral column, and the coracoid at an angle of 100–110° towards it (as in the Thermopolis (*Mayr et al., 2007*) and 11th specimen). The **scapula** is long and slender (Figs. 22 and 23), with the ratio of scapular length (43 mm) to minimal shaft width (2.8 mm) being more than 15 in the left scapula. The shaft is of subequal width through most of its length, although a small distal expansion to maximally 4.4 mm is present in the right scapula (Fig. 22; the distal end of the left scapula is crushed into the vertebral column and therefore poorly preserved). The shaft is gently curved in its proximal two-thirds, but less so than reconstructed by *Wellnhofer (2008*: fig. 6.10*)*. The shaft is transversely broader ventrally than dorsally, and a shallow longitudinal depression is present on the lateral side in its proximal two thirds, although this might be exaggerated by compression, especially proximally (Fig. 23). Whereas the ventral margin of the shaft is rounded transversely over most of its length, it becomes sharp edged towards the margin of the glenoid, and a ventrally and slightly medially facing flat surface is present medial to it.

Proximally, the scapula expands ventrally towards the glenoid, with a maximal expansion of 2.5 mm below the scapular shaft (Fig. 23). In contrast, there is only a very slight dorsal expansion towards the acromion process, which is more anteriorly than dorsally directed, as in other paravians (*Forster et al., 1998*; *Norell & Makovicky, 1999*; *Xu, Wang & Wu, 1999*; *Xu et al., 2002*, *2011*; *Gianechini & Apesteguia, 2011*; *Brusatte et al., 2013*; *Lefevre et al., 2014*). Thus, the suture between the scapula and coracoid is an oblique line extending from the anterior margin of the glenoid anterodorsally, but its anterior extend is uncertain due to breakage in this area. Most of the glenoid articular surface is

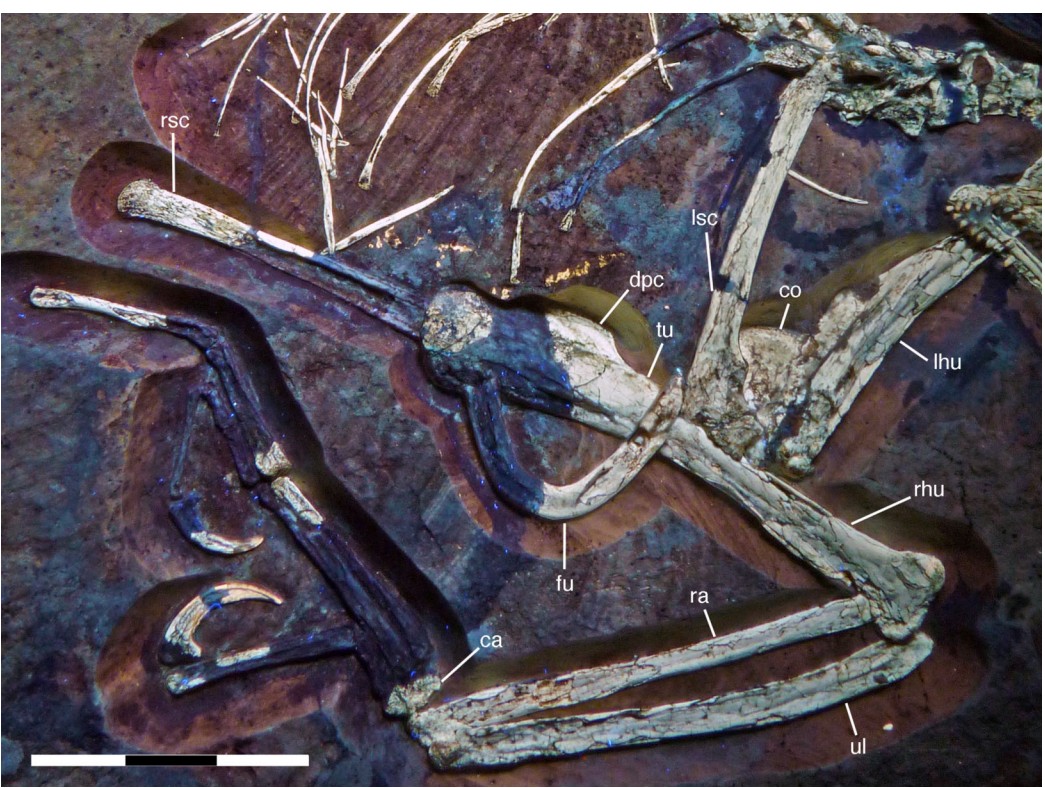

**Figure 22 Preserved elements of the shoulder girdle and right forelimb of the 12th specimen of *Archaeopteryx*.** Photograph under UV light. ca, carpus; co, coracoid; dpc, deltopectoral crest; fu, furcula; lhu, left humerus; lsc, left scapula; ra, radius; rhu, right humerus; rsc, distal end of right scapula; tu, tubercle; ul, ulna. Scale bar is 30 mm.

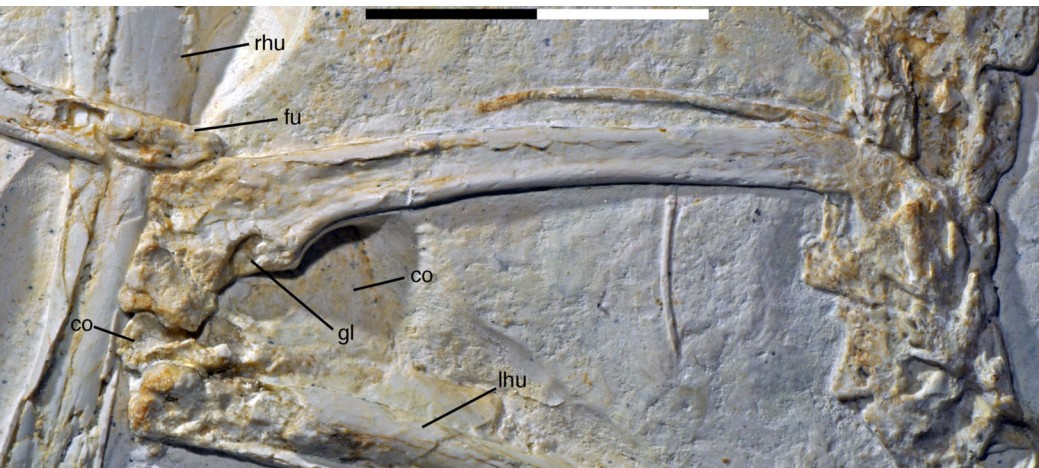

**Figure 23 Left scapulocoracoid of the 12th specimen of *Archaeopteryx*.** co, coracoid; fu, furcula; gl, glenoid; lhu, left humerus; rhu, right humerus. Scale bar is 20 mm.

therefore placed on the scapula, where the rim of the glenoid forms a semioval arch between the ventral expansion at the base of the shaft and the coracoid suture (Fig. 23). The articular surface of the scapular glenoid faces more laterally than ventrally; given the

position of the scapula high on the ribcage parallel to the dorsal vertebrae, this orientation in life was certainly even more or entirely lateral, as in *Unenlagia* (*Novas & Puerta, 1997*), *Rahonavis* (*Forster et al., 1998*) and *Sinornithosaurus* (*Xu, Wang & Wu, 1999*). The dorsal rim of the glenoid lacks any ledge and is even very slightly depressed in its central part. Only towards the suture with the coracoid, which is developed as a lateral ridge in this area, the rim of the glenoid expands somewhat laterally. The supraglenoid fossa on the lateral side of the proximal scapula is large and elongate triangular in outline (Fig. 23), as in other paravians, but its margins are not clearly defined, which might be due to compression. Dorsally, the low acromion process is largely overlapped by the very robust epicleideal process of the furcula, the contact with which extends to almost the level of the posterior margin of the glenoid (Fig. 23). Although the furcula is preserved in close association with the scapula, it is slightly displaced from its facet on the latter. The facet is placed on the laterodorsal edge of the anterior part of the acromion process and extends slightly more ventrally anteriorly. This indicates that the furcula bridged the gap between the left and right should girdles, as in other pennaraptorans (*Nesbitt et al., 2009*), but was not as strongly ventrally directed as reconstructed by *Elzanowski (2002)* and *Wellnhofer (2008)*.

Only the proximalmost portion of the left **coracoid** is preserved in articulation with the scapula. The rest of the bone is broken and was flattened below the rest of the scapulocoracoid. Thus, only the medial side of the main body of the coracoid is visible, and its anterior and posterior margins are overlapped by the scapula and the humerus, respectively (Fig. 23). The portion of the coracoid preserved in articulation with the scapula indicates that this element was angled at approximately 100–110° towards the latter. Thus, the angle between these two elements is smaller than in *Velociraptor* (*Norell & Makovicky, 1999*), but larger than reconstructed by *Elzanowski (2002)* and *Wellnhofer (2008)*. The coracoid was a rather large, plate-like element, as in other specimens of *Archaeopteryx* (*Wellnhofer & Tischlinger, 2004*; *Mayr et al., 2007*), troodontids (*Russell & Dong, 1993*), dromaeosaurids (*Norell & Makovicky, 1999*), and *Sapeornis* (*Zhou & Zhang, 2003b*; *Provini, Zhou & Zhang, 2009*), with an estimated dorsoventral length of 9.5–10 mm. In contrast, *Confuciusornis* and *Jeholornis* possess an elongated coracoid, as present in modern birds (*Chiappe et al., 1999*; *Zhou & Zhang, 2003a*). Proximally, the coracoid forms only a small part of the anterior margin of the glenoid fossa. The coracoid glenoid facet is triangular in outline and faces mainly posteriorly and only slightly laterally.

Only the left ramus of the **furcula** is preserved (Figs. 22 and 24), although the preserved portion includes the ventral bend of the element and the medialmost part of the right ramus (Fig. 24), in contrast to the indication in the reconstructed element, which implies that exactly half of the bone is present. As in other specimens of *Archaeopteryx*, it was a rather robust, U-shaped element. The bone is slightly convex anteriorly; this curvature has certainly been reduced by compression, and might originally have been similar to that seen in other pennaraptorans (*Nesbitt et al., 2009*). A very small, rounded flange is present at the flexure point ventrally (Fig. 24) and might indicate an incipient hypocleidium (though see *Nesbitt et al., 2009*, who questioned the homology of such tubercles, which are

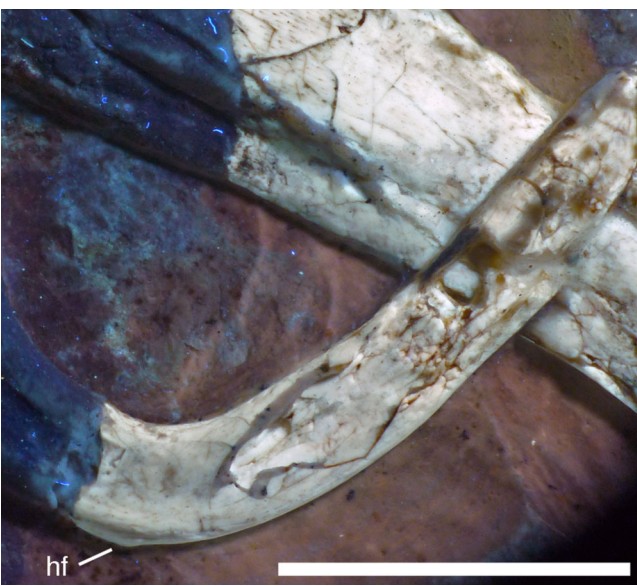

**Figure 24 Preserved left ramus of the furcula of the 12th specimen of *Archaeopteryx*.** Photograph under UV light. hf, hypocleidal flange. Scale bar is 10 mm.

common in theropods, with the hypocleidium of birds). The epicleideal process of the furcula is slightly flattened and triangular in outline, tapering to a rounded point mediodorsally. The largely collapsed ramus of the furcula indicates that this element might have been hollow (Fig. 24), as in *Aerosteon* (*Sereno et al., 2008*) and *Buitreraptor* (*Makovicky, Apesteguia & Agnolin, 2005*).

Both **humeri** are crushed and rather poorly preserved. As far as can be established, their morphology seems to correspond closely to that found in other specimens of *Archaeopteryx* (see *Wellnhofer, 2008*, *2009*). The right humerus is exposed in posterior view, but most of the proximal end is missing. The deltopectoral crest is well developed and its distal end raises rather abruptly out of the shaft (Fig. 22). Assuming a similar length of the left and right humerus, the deltopectoral crest reached approximately for 30% of the length of the humerus down the shaft, as in *Balaur* (*Brusatte et al., 2013*), while it measures only 25% in *Anchiornis* (*Pei et al., 2017*). A slightly raised, elongate oval tubercle is present on the lateral side of the shaft at the distal end of the deltopectoral crest (Fig. 22). The distal end is transversely expanded, apparently more so medially than laterally. Its posterior side is flat to very slightly concave, but a large posterior depression adjacent to the articular end, as it is found in many non-avialan theropods and also in the ninth specimen of *Archaeopteryx* (Ottmann & Steil specimen, so called 'chicken wing'), is absent. The medial side of the distal end, which is exposed in the left humerus, is flattened. Based on the more complete left humerus, which is exposed in medial view, the shaft seems to be less flexed posteriorly in its proximal portion than in other specimens of *Archaeopteryx*.

As with the humeri, the radia and ulnae are poorly preserved, and little detail can be gained from them. As with most of the other appendicular elements, the shafts of the

bones are collapsed, indicating that they were hollow. The more complete right ulna and radius are exposed in medial and/or anteromedial view (Fig. 22).

The **ulna** is slightly longer than the radius, and both elements are shorter than the humerus, as in other specimens of *Archaeopteryx*, non-avialan theropods and *Confuciusornis* (*Chiappe et al., 1999*), whereas the ulna is as long or longer than the humerus in *Jeholornis* (*Lefèvre et al., 2014*), *Sapeornis* (Zhou & Zhang, 2003; *Provini, Zhou & Zhang, 2009*) and more derived avialans. The ulna is slightly bowed posteriorly in its proximal half, while the distal half of the shaft is straight (Fig. 22). The proximal end is very slightly expanded anteroposteriorly and there is a pronounced angle of slightly less than 90° between the articular surface and the shaft, but an olecranon process is absent. A short, sharply defined ridge extends from the proximal end distally at about the mid-width of the bone. Given the compaction of the element, it seems plausible that this ridge marks the rather sharply defined boundary between the anterior and the medial side of the ulna, so that both sides are exposed here, pressed into a single plane. The distal end of the right ulna is exposed in medial view. There is no anteroposterior expansion and the distal articular surface is strongly convex, extending proximally in a small, triangular process medially, as in *Balaur* (*Brusatte et al., 2013*). The left ulna, the distal end of which is exposed in posterolateral view, shows furthermore that the articular end was strongly convex anteroposteriorly.

The **radius** is slightly more slender than the ulna and straight (Fig. 22). The proximal end is not preserved in the left element and hidden by the humerus in the right bone. The distal end shows little to no expansion, but is too poorly preserved to say anything about its morphology.

Both carpi and mani are poorly preserved, and only few elements are present. Of the **carpus**, only the poorly preserved semilunate carpal of the right manus is preserved (Fig. 22), but does not provide much additional information on the carpus of *Archaeopteryx* than that known from other specimens (*Ostrom, 1976*; *Wellhofer, 2008*, *2009*). A conspicuous proximodistally oriented ridge extends across the dorsal surface of the carpal and divides the smaller medial portion, that would have articulated with metacarpal I, from the wider lateral part that articulated with metacarpals II and III. The articular surface for metacarpal I is very slightly angled medially in respect to that for metacarpals II and III.

Only large parts of **metacarpals** II and III are preserved in the left hand (Fig. 25). As in the Berlin (*Wellhofer, 1985*, *2008*), Eichstätt (*Wellhofer, 1974*), Munich (*Wellhofer, 1993*), and Thermopolis (*Mayr et al., 2007*) specimens of *Archaeopteryx*, metacarpal III is subequal in length to metacarpal II, but only slightly more than half its width, straight, and closely appressed to the latter over its entire length. In contrast, the third metacarpal of the London (BSPG cast) and Solnhofen (*Wellhofer, 1988b*) specimens are slightly more curved and separated from metacarpal II over most of their length by a notable gap.

Only parts of the **phalanges** are, largely poorly, preserved, and they do not show much detail (Figs. 22 and 25). As in other specimens of *Archaeopteryx*, the phalanges of the hand were slender and elongate, showing no signs of laterally or medially located longitudinal furrows. The unguals are strongly recurved, with strongly expanded proximal flexor

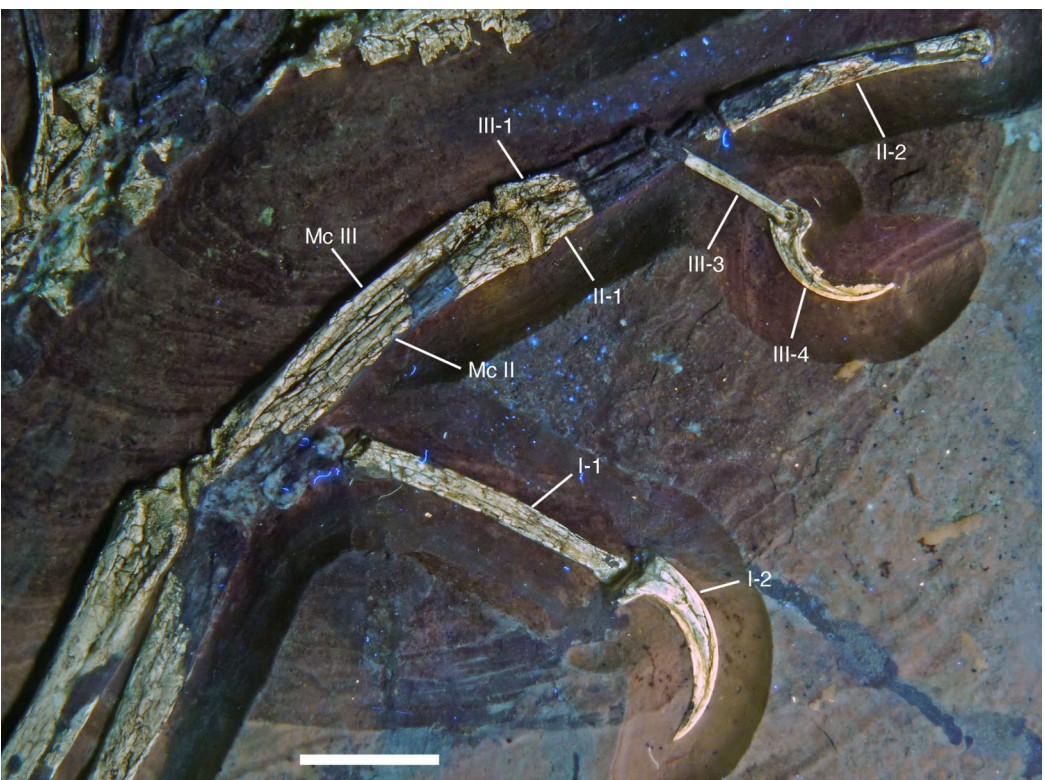

**Figure 25 Left manus of the 12th specimen of *Archaeopteryx* as preserved.** Photograph under UV light. Roman numerals denominate digits, Arabic numbers indicate phalange numbers. Mc, metacarpal. Scale bar is 10 mm.

tubercles (Figs. 25 and 26). Unlike the situation in many maniraptoran theropods (*Rauhut & Werner, 1995*), there is only a small, incipient dorsal 'lip' on the proximal articular end (Fig. 26). This is also different from the situation in *Xiaotingia* (*Xu et al., 2011*) and *Anchiornis* (*Pei et al., 2017*) and most other Pennaraptora, but similar to *Balaur* (*Brusatte et al., 2013*). The unguals preserve parts of their horny sheaths, which enlarge the bony core for approximately one third of the total length of the unguals and add to the pronounced curvature of these elements (Fig. 26).

Of the pelvis, only the shafts and distal ends of the pubis and ischium are preserved (Figs. 27 and 28). As in all non-avialan maniraptoran theropods, with the possible exception of Scansoriopterygidae (*Zhang et al., 2008*; *Czerkas & Feduccia, 2014*), the **pubis** is considerably longer than the ischium (Fig. 28). As in the Berlin, Eichstätt, Solnhofen, Thermopolis, and 11th specimen (*Dames, 1884*; *Wellnhofer, 1974*, *1992*, *2008*, *2009*; *Mayr et al., 2007*; *Foth, Tischlinger & Rauhut, 2014*), the pubic shafts are parallel to the ischium and directed posteroventrally in respect to the long axis of the vertebral column, indicating that this might have been the natural orientation, in contrast to the more ventral orientation advocated by *Wellnhofer (1985*, *2008*, *2009)*. This interpretation is supported by the preserved gastralia, which extend up to the margin of the pubis in the Eichstätt (*Wellnhofer, 1974*, *2008*), 11th (*Foth, Tischlinger & Rauhut, 2014*) and the current specimen (Fig. 28), which should not be the case if the pubis was rotated posteriorly

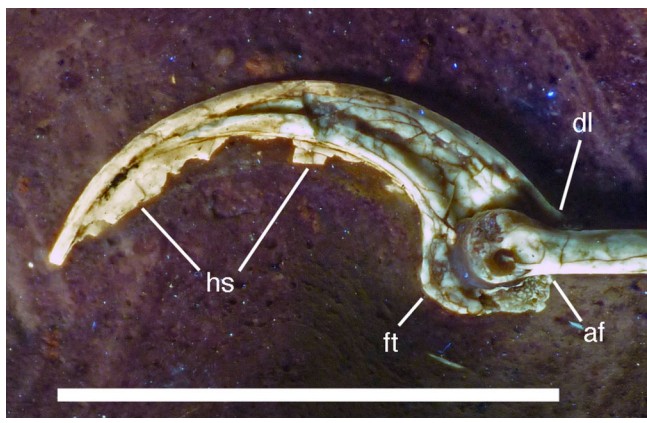

**Figure 26 Left manual ungual III of the 12th specimen of *Archaeopteryx*.** Photograph under UV light. af, articular facet; dl, dorsal lip; ft, flexor tubercle; hs, horny sheath. Scale bar is 10 mm.

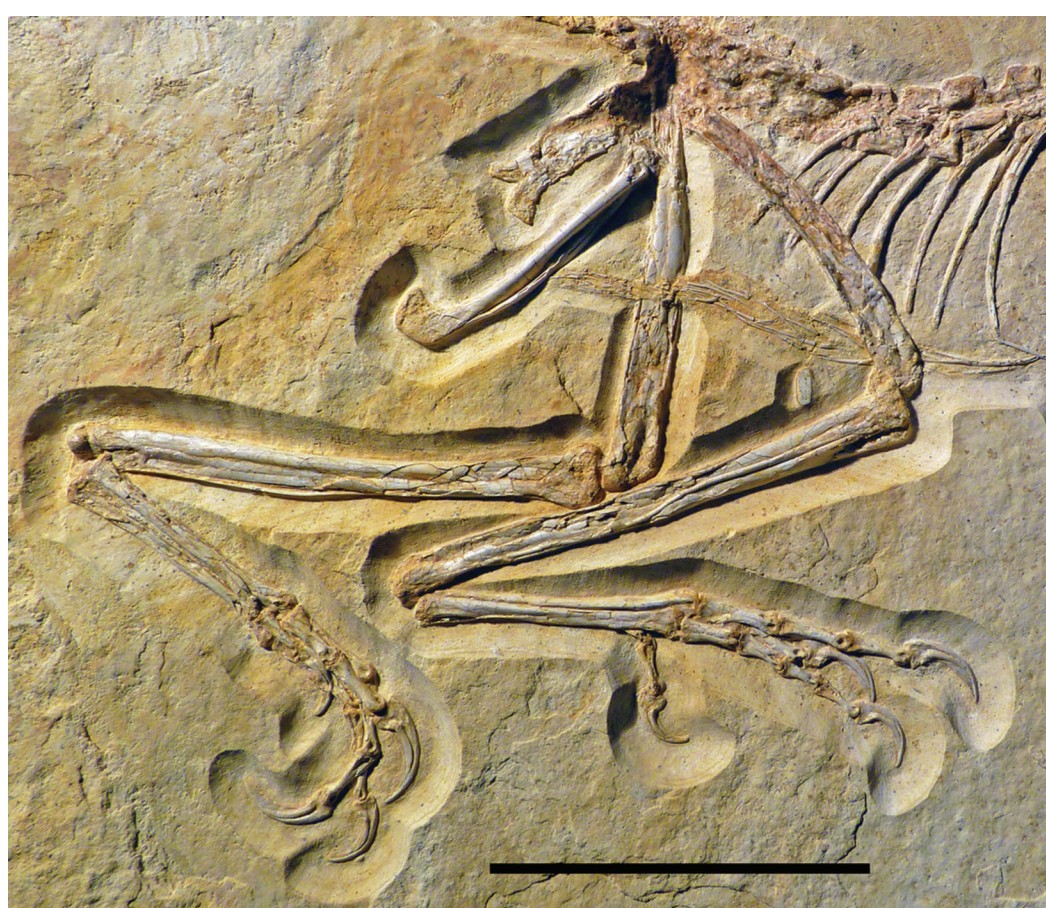

**Figure 27 Preserved elements of the pelvis and hindlimbs of the 12th specimen of *Archaeopteryx*.** Scale bar is 50 mm.

post-mortem. Curiously, however, in all of these specimens, as well as the Munich (*Wellnhofer, 1993*) and Haarlem specimens (*Ostrom, 1970*; *Wellnhofer, 2008*), the gastralia are placed at about the half length of the pubis, not at its distal end, as would be suspected

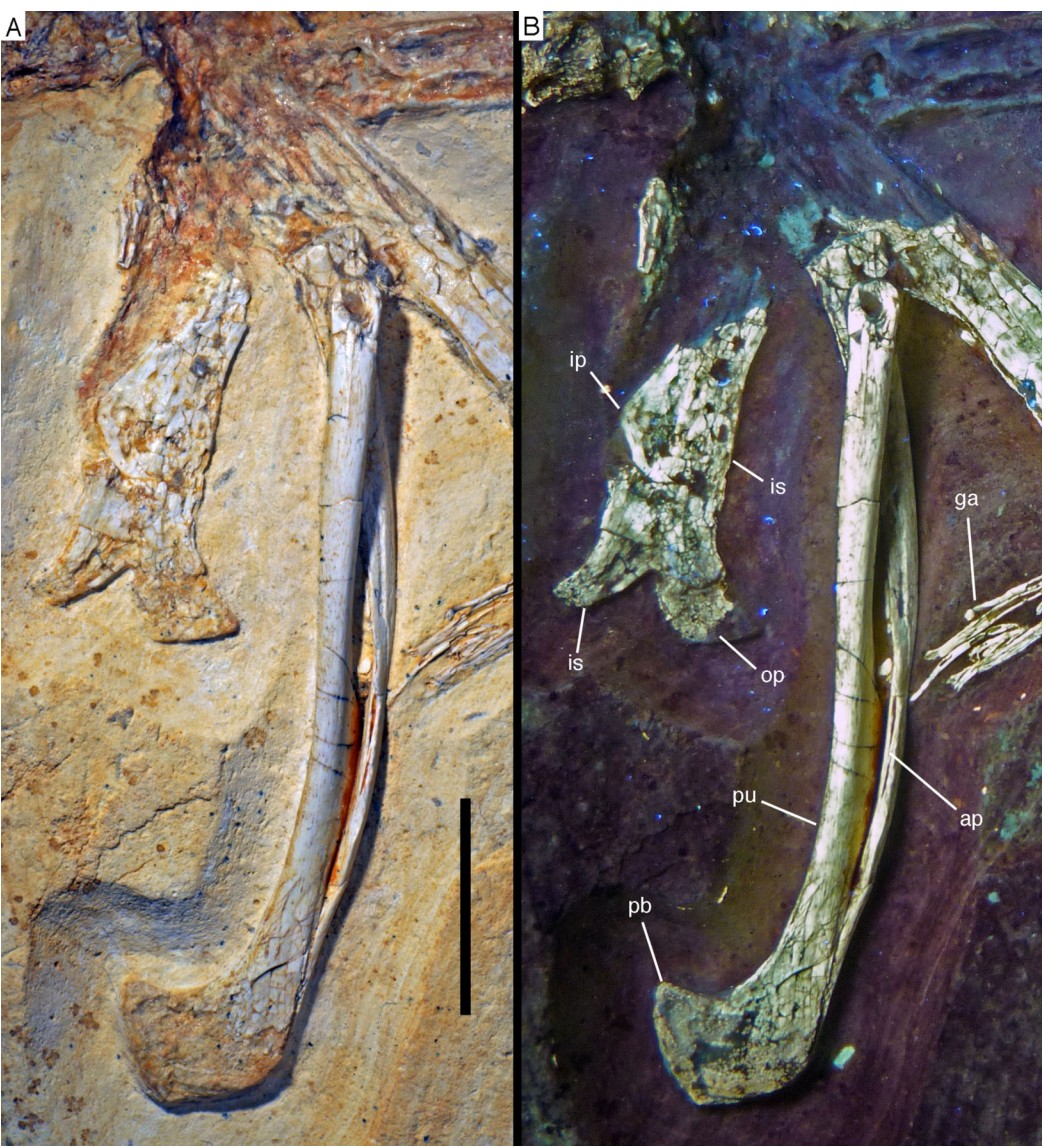

**Figure 28 Pelvic elements of the 12th specimen of _Archaeopteryx_.** (A) Photograph under normal light. (B) Under UV light. ap, medial surface of the pubic apron of the left pubis; ga, gastralia; ip, 'intermediate process;' is, ischium; op, obturator process; pb, pubic boot; pu, pubis. Scale bar is 10 mm.

(see also discussion). This is also the case in _Jianianhualong_ (_Xu et al., 2017_), _Anchiornis_ (_Zheng et al., 2014_; _Pei et al., 2017_), _Sapeornis_ (_Zhou & Zhang, 2003b_) and various dromaeosaurids (_Hone et al., 2010_; _Xu et al., 2010_; _Han et al., 2014_; _Lü & Brusatte, 2015_).

The pubic shafts are slender and very slightly flexed posteroventrally. The proximal break of the right pubis shows that the bone was hollow (Fig. 28). _Xu et al. (2011)_ stated that a lateral expansion of the pubic shafts was present in the Solnhofen specimen. Such an expansion is clearly absent in the current specimen. Moreover, examination of the Solnhofen specimen by two of us (OR, CF) indicates that the only exposed right pubic shaft is only slightly convex laterally, and this might furthermore be partially or even

entirely caused by the compaction of the femur onto the lateral side of the proximal pubic shaft. Thus, a noted lateral expansion, as it is found in microraptorine dromaeosaurids (*Hwang et al., 2002*; *Longrich & Currie, 2009*; *Pei et al., 2014*), is clearly absent.

The pubic apron is narrow and extended for approximately half the length of the pubic shafts (Fig. 28), as in the London and Thermopolis specimens (*de Beer, 1954*; *Mayr et al., 2007*). The extent of the pubic apron is thus considerably less than in non-coelurosaurian theropods (*Gilmore, 1920*; *Galton & Jensen, 1979*; *Madsen, 1976*; *Currie & Zhao, 1993a*), slightly less than in non-maniraptoran coelurosaurs (*Osmólska, Roniewicz & Barsbold, 1972*; *Brochu, 2003*), and comparable to the situation in dromaeosaurids (*Norell & Makovicky, 1999*; *Hwang et al., 2002*). In more derived avialans, the pubic apron is further reduced; it extends over slightly less than half the length of the pubis in *Jeholornis* (*O'Connor et al., 2012*), about a third in *Confuciusornis* (*Chiappe et al., 1999*) and *Sapeornis* (*Zhou & Zhang, 2003b*), roughly the distal fourth in *Sinornis* (*Sereno, Rao & Li, 2002*), and the left and right pubes are separated over their entire length in Ornithurae (*Clarke & Norell, 2002*). As in most theropods, the apron is placed anteriorly on the medial side of the pubic shafts. Towards the proximal end of the median contact of the aprons of the left and right pubis, the medial edge of the flange becomes sharp edged and flexes very slightly posteriorly.

Whereas the left and right pubes have slightly disarticulated over most of the length of the pubic apron, exposing a longitudinal furrow in the edge of the apron of the left pubis proximally, they remain in contact distally, and no suture is visible at the distal boot. The distal boot only expands posteriorly and has a hook-like appearance, with an angled ventral margin and a slight proximal expansion posteriorly (Fig. 28). An anterior expansion of the pubic boot is also absent in some (e.g. *Velociraptor*: *Norell & Makovicky, 1997*; *Unenlagia*: *Novas & Puerta, 1997*; *Microraptor*: *Pei et al., 2014*; *Zhenyuanlong*: *Lü & Brusatte, 2015*), but not all (e.g. *Deinonychus*: *Ostrom, 1976*; *Achillobator*: *Perle, Norell & Clark, 1999*) dromaeosaurids, *Anchiornis* (*Hu et al., 2009*; *Pei et al., 2017*), and many basal avialans that retain a pubic boot (e.g. *Sapeornis*: *Zhou & Zhang, 2003b*; *Sinornis*: *Sereno, Rao & Li, 2002*). The lateral surface of the pubic boot is poorly preserved but seems to be flat.

The **ischia** are approximately half the length of the pubis and considerably more massive (Fig. 28), which might be slightly exaggerated by compression. Only the distal end of the right ischium seems to be preserved, whereas the more proximal portion visible obviously belongs to the underlying left ischium. Thus, the ischia were in close contact distally, but obviously not fused. A posteriorly rounded, triangular 'intermediate process' is present at about half length of the ischial shaft (Fig. 28), as in all other specimens of *Archaeopteryx* where the ischium is preserved (*Elzanowski, 2002*; *Foth, Tischlinger & Rauhut, 2014*), as well as in *Sinovenator* (*Xu et al., 2002*), *Jianianhualong* (*Xu et al., 2017*), *Microraptor* (*Hwang et al., 2002*), and *Xiaotingia* (*Xu et al., 2011*). It projects posteriorly and is proximodistally more elongated than anteroposteriorly expanded.

As in all specimens of *Archaeopteryx*, the distal end of the ischium is bifurcated, with the posterior ramus representing the ischial shaft, whereas the anterior flange corresponds to the obturator process (Fig. 28). Thus, the bifurcated appearance of these structures stems from a well-developed distal incision between the ischial shaft and the obturator

process, similar to the morphology in many basal, non-coelurosaurian theropods (*Rauhut, 2003*). Within coelurosaurs, in which the obturator process usually disappears gradually into the ischial shaft distally, such an incision is only present in *Anchiornis* (IVPP V 14378; *Xu et al., 2009a*, *2011*), *Serikornis* (*Lefèvre et al., 2017*) and, to a lesser degree, in *Buitreraptor* (*Makovicky, Apesteguia & Agnolin, 2005*; *Agnolin & Novas, 2013*). The distal end of the ischial shaft is slightly flexed posteriorly and tapers to a rounded tip. The obturator process is damaged, but it was obviously considerably larger than the distal tip of the ischial shaft; photographs taken prior to preparation of the specimen show several bone fragments anterior to the preserved part, indicating that the process was considerably expanded anteriorly, as in the Eichstätt, Munich and Thermopolis specimens (*Wellnhofer, 1974*, *1993*, *2008*, *2009*; *Mayr et al., 2007*). *de Beer (1954*: figs. 6*)*, *Mayr et al. (2007*: fig. 11C*)* and *Agnolin & Novas (2013*: fig. 3.5f*)* illustrated the London specimen with a much smaller, ventrally rather than anteriorly directed process. However, the ischium of the London specimen is not entirely exposed in lateral view, but somewhat inclined into the slab anteriorly, and the anterior end of the obturator process is covered by the pubis (BSPG cast of NHMUK 37001; see also *Wellnhofer, 1985*: fig. 2), so nothing can be said about its shape. Likewise, in the Berlin specimen, for which *de Beer (1954*: fig. 7*)* also illustrated a small, ventrally directed process, the ischium is poorly preserved, but UV photographs show that the obturator process is similarly anteriorly expanded as in other specimens (*Tischlinger & Unwin, 2004*; *Wellnhofer, 2008*: fig. 5.55).

The **femora** are poorly preserved and largely collapsed (Fig. 27). The proximal ends of both elements are missing. The shaft is slender, slightly curved posterodistally and seems to become more robust distally, being approximately 120–130% of the width of the proximal shaft just proximal to the distal expansion. A slightly raised ridge on the posteromedial edge of the proximal shaft of the left femur probably represents the fourth trochanter, which was thus strongly reduced, as in most coelurosaurian theropods. The collapsed shafts show that the femora were hollow and thin-walled.

Both **tibiae** are preserved, the right element being exposed in lateral and the left element in medial and, partially anteromedial views (Figs. 27 and 29). Both elements are preserved in articulation with the fibulae and the proximal tarsals. The tibiae are straight and slender elements, being longer than the femur. The proximal end is expanded, more so posteriorly than anteriorly, to a maximal anteroposterior width of 8.3 mm (measured on the left tibia). The posterior expansion forms a rounded posterior tubercle of c. 1.5 mm anteroposterior length that abruptly arises from the shaft directly distal to the proximal end, as also figured for the Eichstätt specimen by *Wellnhofer (1974*: fig. 11*)*. The proximal end of the right tibia shows that this expansion is entirely formed by the medial part of the proximal tibia, whereas the posterior margin of the fibular condyle seems to be more or less flush with the posterior margin of the proximal tibial shaft and thus considerably offset anteriorly from the posterior end of the medial side. Two probably nutrient foramina (see *Seymour et al., 2012*) enter the tibia on the posterolateral side of the proximal end: a smaller, slightly more distally placed foramen at the level of the distal end of the posterior expansion that opens laterally from the

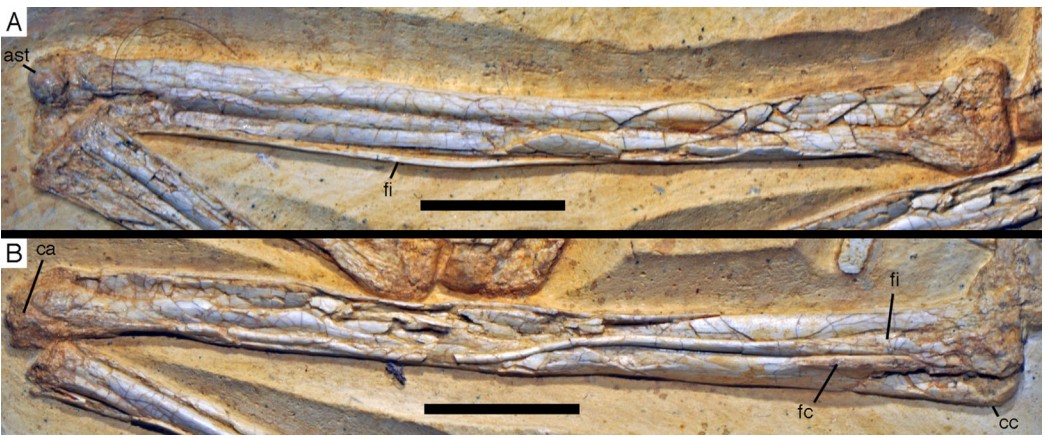

**Figure 29 Tibiae, fibulae and proximal tarsals of the 12th specimen of *Archaeopteryx*.** (A) Left tibia, fibula and astragalus in medial and anteromedial view. (B) Right tibia, fibula and calcaneum in lateral view. ast, astragalus, ca, calcaneum; cc, cnemial crest; fc, fibular crest; fi, fibula. Scale bars are 10 mm.

base of this expansion, and a larger, more proximally placed foramen in the posterior end of the base of the fibular condyle, which opens laterodistally.

The cnemial crest is small, expanding c. 1.3 mm anteriorly from the fibular condyle, and only offset from the shaft by a small concavity in the anterior margin of the tibia distally (Fig. 30), as also figured by *Wellnhofer (1974)* for the Eichstätt specimen. In contrast to many basal theropods, its proximal end does not extend proximally beyond the proximal surface of the main body of the tibia. A weakly developed longitudinal lateral ridge is present close to the anterior end of the cnemial crest, as in most theropods, and helps to define a well-developed incisura tibialis.

A well-developed lateral longitudinal crest of the contact with the fibula is present on the anterior part of the lateral side of the shaft (Figs. 29 and 30), as in all theropods (*Gauthier, 1986*). The crest is straight and slender and 7.7 mm long. It becomes lower proximally, but is clearly offset from the proximal end, its proximal end being placed some 7 mm below the articular surface.

Distal to the cnemial crest, the shaft of the tibia gradually narrows distally, but it exact shape and outline cannot be established due to compression. The distal end is clearly considerably broader transversely than deep anteroposteriorly. However, only the lateral malleolus is slightly expanded from the shaft, whereas the medial malleolus is more or less continuous with the latter. As in most theropods (*Currie & Zhao, 1993a*; *Brochu, 2003*; *Burnham, 2004*), the medial side of the distal end is somewhat deeper anteroposteriorly than the lateral side. Whereas the medial side of the distal tibia is flattened, the lateral side is anteroposteriorly gently rounded. A short, but well-developed longitudinal ridge is present at the medial edge of the anterior side of the distal end of the tibia and braces the ascending process of the astragalus medially. The ridge expands medially distally and its medial edge curves onto the medial side of the bone. Such a ridge is also present in the Thermopolis (*Mayr et al., 2007*) and the 11th specimens (*Foth, Tischlinger & Rauhut, 2014*).

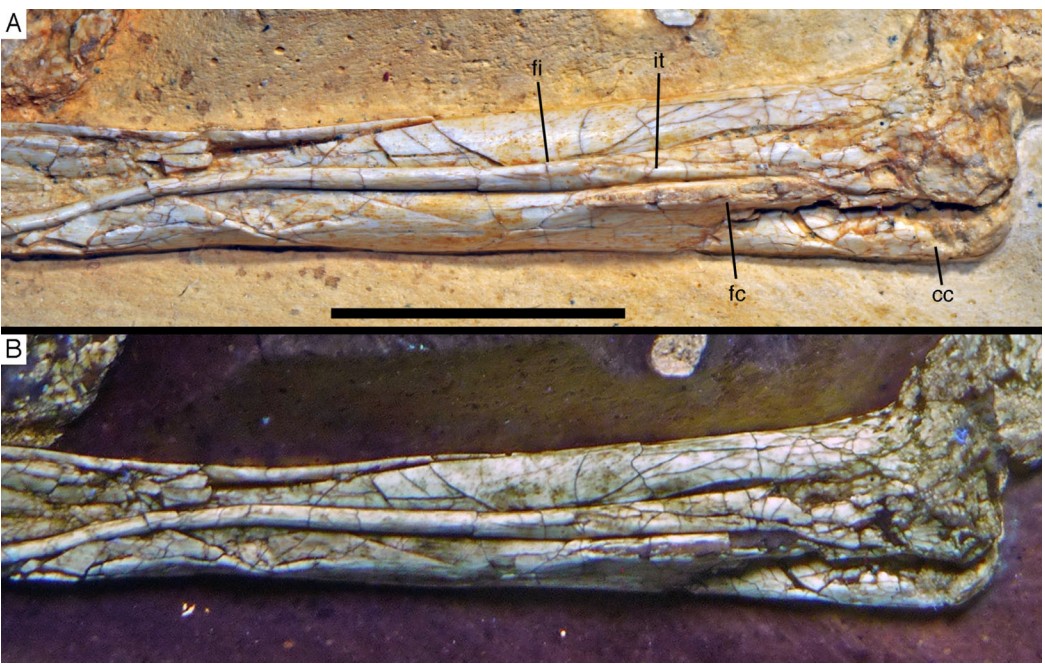

**Figure 30 Proximal ends of right tibia and fibula of the 12th specimen of *Archaeopteryx*.** (A) Photograph under normal light. (B) Photograph under UV light. cc, cnemial crest; fc, fibular crest; fi, fibula; it, iliofibularis tubercle. Scale bar is 10 mm.

The **fibula** is extremely slender and closely attached to the tibia. Both fibulae are preserved, but only the distal half is visible of the left element. The proximal end is expanded to a maximal anteroposterior width of 5.6 mm, or approximately 67% of the proximal width of the tibia. As in most theropods, the bone is here more strongly posteriorly than anteriorly expanded. Parts of the central part of the proximal expansion are broken away, revealing that the bone was very thin in this area, considerably more so than at the base of the shaft or the proximal end, indicating that a large and extensive medial groove was present in the fibula, as it is the case in many coelurosaurian theropods (*Rauhut, 2003*).

The shaft rapidly narrows in its proximal part towards the iliofibularis tubercle. The latter is developed as a low ridge on the anterolateral side of the fibula that becomes wider and less conspicuous distally. Its position roughly coincides with the area of the fibula that contacts the lateral fibular crest of the tibia. In contrast to many maniraptorian theropods (*Rauhut, 2003*: fig. 48), the fibular shaft does not narrow abruptly distal to the tubercle, but gradually decreases in width from this area to approximately the mid-shaft. Thus, whereas the shaft is 1 mm wide at the level of the beginning of the iliofibularis tubercle, its minimal anteroposterior width is 0.6 mm. The shaft is flattened, so that its transverse width is even less. Distally, the fibula is closely appressed to the anterolateral side of the tibia, and its distal end is slightly and gradually expanded from the shaft to a maximal width of 0.9 mm. The distal end is rounded and sits in a concave facet of the calcaneum.

Astragalus and calcaneum are preserved in close contact with the tibia and fibula, but neither seems to be fused with the elements of the lower leg or with each other. The left
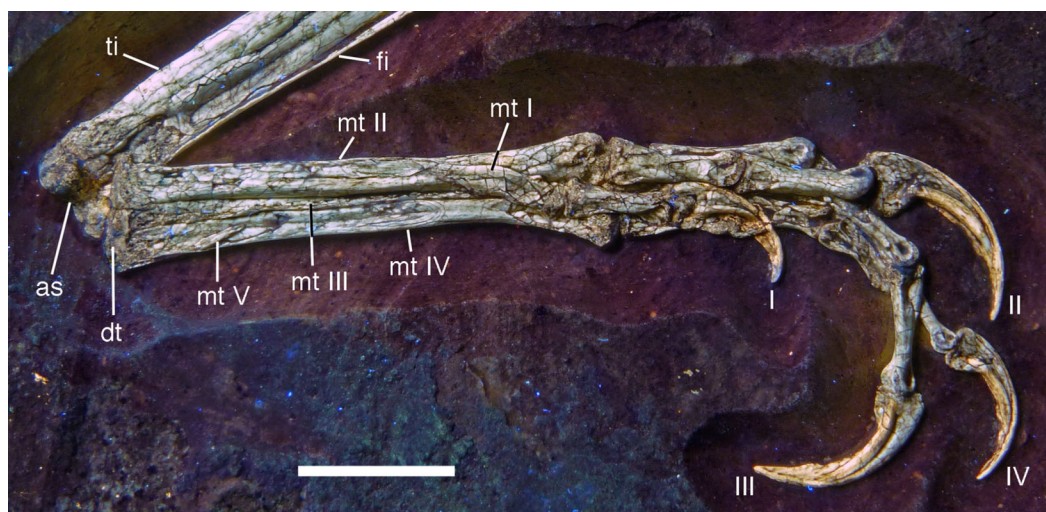

**Figure 31 Left tarsus and pes of the 12th specimen of *Archaeopteryx*.** Photograph under UV light. as, astragalus; fi, fibula; dt, distal tarsal; mt, metatarsal; ti, tibia; Roman numerals denominate digits of the pes. Scale bar is 10 mm.

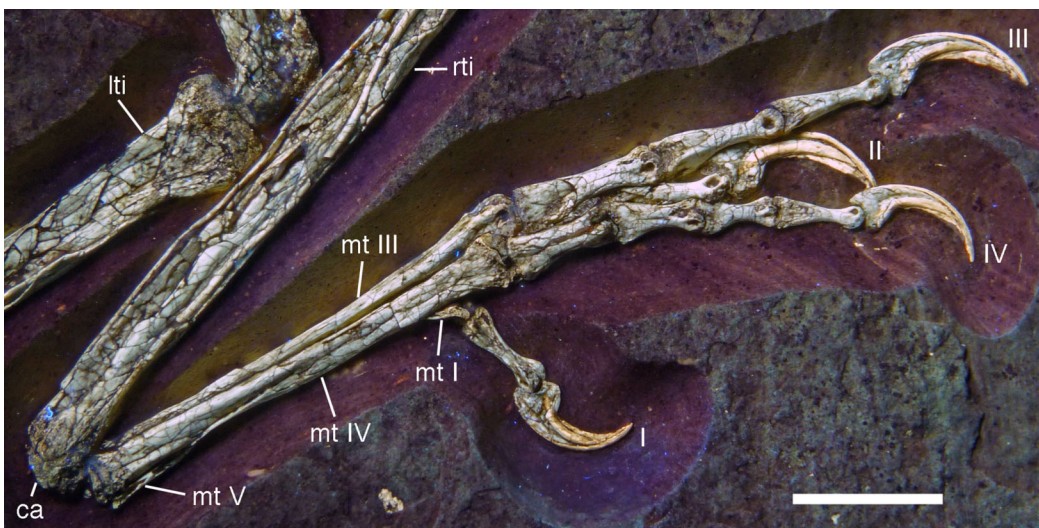

**Figure 32 Right tarsus and pes of the 12th specimen of *Archaeopteryx*.** Photograph under UV light. ca, calcaneum; lti, left tibia; mt, metatarsal; rti, right tibia; Roman numerals denominate digits of the pes. Scale bar is 10 mm.

**astragalus** is exposed in anteromedial and partially distal view (Fig. 31), whereas the right tarsus is visible in lateral and slightly posterior view (Fig. 32). As in the Thermopolis specimen (*Mayr et al., 2007*), the bodies of the astragalus and calcaneum are rather low and the medial condyle of the astragalus is bulbous and more pronounced than the lateral condyle (Fig. 31). Both condyles are separated by a deep concavity anteriorly and distally. A transverse groove across the condyles is absent; such a groove is present in many non-maniraptoran theropods (*Rauhut, 2003*), and probably represents a remnant of the suture between separate ossifications of the astragalar body and the ascending process (*Rauhut & Pol, 2017*). Although the condyles of the astragalus face anterodistally, as it is usual in

tetanuran theropods (*Sereno et al., 1994*; *Rauhut, 2003*), the medial condyle is placed entirely distal to the tibia, as in basal theropods (*Rauhut & Carrano, 2016*). Laterally, the tibia extends slightly further distally so that the lateral condyle is at least partially placed anterodistal to the tibia. On the medial side of the astragalar body, a well-developed depression is present. This depression becomes deeper posteriorly and is bound posterodistally by a raised ridge.

The ascending process of the astragalus is largely present, but poorly preserved (Fig. 31). As in most coelurosaurian theropods (*Rauhut, 2003*), its base extends approximately over the entire width of the astragalar body, and it is considerably higher (6.9 mm) than the latter (2.4 mm) and thus slightly more than 10% of the length of the tibia. Anteriorly, it is separated from the astragalar condyles by a well-developed groove. Proximally, a portion of the ascending process is broken away, revealing a well-developed facet on the anterior side of the distal tibia adjacent to the medial ridge.

*Mayr et al. (2007)* noted that, in the Thermopolis specimen, the ascending process of the astragalus was slightly separated from the medial ridge on the anterior side of the distal tibia by a groove and the distal end of the tibia extended slightly medially to the body of the astragalus. They interpreted this as an original feature and compared it to the situation in ornithomimosaurs and *Jeholornis* (*Mayr et al., 2007*: 110). In the new specimen, however, the medial margin of the ascending process is closely appressed to the medial ridge and the medial side of the distal tibia is more or less flush with the medial side of the astragalar body, as in most theropods, indicating that the situation in the Thermopolis specimen might be due to a slight displacement of the tarsus.

Only the right **calcaneum** is visible in posterolateral view (Fig. 32), and most of its lateral surface has suffered from erosion. *Mayr et al. (2007)* noted that the calcaneum in the Thermopolis specimen was approximately as wide as the distal end of the fibula in anterior view. This also seems to be the case in the 12th specimen, but the calcaneum narrows posteriorly and is thus wedge-shaped in distal view. Posteriorly, the calcaneum does not reach the posterior edge of the tibia, but the astragalus expands below the latter to take part in the posterolateral edge of the tarsus, as in most tetanuran theropods. The facet for the tibia on the calcaneum is steeply inclined posteroproximally and is slightly concave proximodistally, so that much of the calcaneum is placed anterior to the tibia. On the anterior part of the proximal side of the calcaneum a smaller, anteroposteriorly concave facet for the fibula is present. It is placed more proximal than the distal end of the facet for the tibia and separated from the latter by a pointed proximal process of the calcaneum. Distally, the calcaneum is rounded.

The left **tarsometatarsus** is exposed in ventromedial view (Fig. 31), whereas the right is visible in lateral and slightly dorsal view (Fig. 32). The presence, identity and probable fusion of the distal tarsals with the metatarsals in *Archaeopteryx* have remained enigmatic. No separate distal tarsals were identified in the London specimen by *de Beer (1954)*, who left the question open if this might be due to fusion of these elements with the metatarsals. *de Beer (1954*: 29*)* noted that the latter state was reported for the Berlin specimen, but, unfortunately, did not cite by whom; *Dames (1884*: 37*)* only discussed the possible fusion of the metatarsals with each other. Likewise, *Heller (1959)* noted the

possible fusion of the proximal ends of the metatarsals of the Maxberg specimens, but did not comment on the distal tarsals. *Wellnhofer (2008*: fig. 5.72*)* figured the Maxberg specimen as having two unfused distal tarsals above metatarsal III and IV, based on the X-ray image of this specimen, but does not comment on this in the text. For the Eichstätt specimen, *Wellnhofer (1974*: 199*)* described a thin, plate-like bone proximal to metatarsal III, which he considered to be largely fused with the metatarsal, whereas a possible, small and rounded distal tarsal IV seems to have been separate from the metatarsal. However, in the right foot of the Eichstätt specimen, a clear suture between the distal tarsal and metatarsal III is visible (O. W. M. Rauhut and C. Foth, 2014, personal observation). For the Solnhofen specimen, *Wellnhofer (1988b, 1992)* identified two distal tarsals, but placed above metatarsals II and III and not fused to these elements. *Tischlinger & Unwin (2004)* identified an unfused lateral distal tarsal in the Berlin specimen. No distal tarsals are mentioned in the description of the Thermopolis specimen (*Mayr et al., 2007*). Finally, in his reconstruction of the skeleton of *Archaeopteryx*, *Wellnhofer (2008*: fig. 6.13*)* figures three separate distal tarsals, proximally overlying metatarsals II–IV.

The presence of three distal tarsals in *Archaeopteryx* would be extremely unusual, since only two such elements are generally present in non-avialan theropod dinosaurs (*Ostrom, 1969*; *Osmólska, Roniewicz & Barsbold, 1972*; *Madsen, 1976*; *Norell & Makovicky, 1997*). In these animals, the two distal tarsals are usually interpreted as distal tarsals III and IV, with the former mainly overlapping metatarsal III (and often partially or even largely metatarsal II; *Ostrom, 1969*) and the latter overlapping metatarsal IV. In avialans more derived than *Archaeopteryx*, the distal tarsals seem to be already completely fused with the metatarsals into a tarsometatarsus (*Chiappe et al., 1999*; *Zhou & Zhang, 2002*).

In the new specimen, there is a single large element capping the proximal ends of metatarsals III and IV, whereas the proximal end of metatarsal II is expanded slightly more proximally than the other metatarsals to be approximately level with the proximal surface of this distal tarsal (Fig. 31), as it is the case in *Mei*, for example (IVPP V 12733). A small indentation is present posteriorly in this single element above the contact between metatarsals III and IV. This is similar to the situation in a specimen of *Velociraptor* described by *Norell & Makovicky (1997)*, which also has a single distal tarsal with an indentation above the contacts of the two metatarsals. *Norell & Makovicky (1997)* interpreted this tarsal in *Velociraptor* as the fused distal tarsals III and IV, given that the relationship of this element with the metatarsals is the same as that of these two tarsals in other theropod dinosaurs, and the indentation above the contact between the metatarsals indicates a line of fusion. We agree with this interpretation and suggest that this is the same in at least this specimen of *Archaeopteryx*.

The distal tarsal is thus a blocky element, with an almost straight posterior margin, with a small indentation above the contact between metatarsals III and IV (Fig. 31). The proximal surface is slightly convex both anteroposteriorly and transversely over metatarsal IV, but slightly depressed above metatarsal III. In the right pes, the lateral edge of the distal tarsal is broken, revealing two large cavities in the interior of the element. As for the question of fusion between the distal tarsal and the metatarsals, the element in the new

specimen is closely appressed to the proximal ends of metatarsals III and IV, but a suture line is clearly visible both under normal and UV light, indicating that fusion, if present, is at best partial. Distal tarsals are fused to the metatarsals in various Pennaraptora (*Chiappe et al., 1999*; *Norell & Makovicky, 1999*; *Hwang et al., 2002*; *Zhou & Zhang, 2003b*; *Yuan, 2008*; *Balanoff & Norell, 2012*; *Brusatte et al., 2013*; *Lefèvre et al., 2014*), while they seem to be unfused in *Sinornithosaurus* (*Xu & Wang, 2000*), *Deinonychus* (*Ostrom, 1969*) and *Anchiornis* (*Pei et al., 2017*).

As in other specimens of *Archaeopteryx*, the **metatarsus** is long and slender, with metatarsals II to IV being closely appressed to each other over their entire length (Figs. 31 and 32). However, in contrast to the situation reported in the Maxberg and Solnhofen specimens (*Heller, 1959*; *Wellnhofer, 1988b*, *1992*), the metatarsals are not fused proximally, but separated over their entire length. The left metatarsus is exposed in mainly posterior and slightly medial view (Fig. 31), whereas the right metatarsus is exposed in lateral view (Fig. 32). As in all theropods, metatarsal I does not reach the tarsus proximally, and metatarsal V is reduced to a small splint of bone.

Metatarsal I is a short element, slightly more than one fourth of the length of metatarsal II, to which it is attached in the distal third of the latter (Fig. 31), as in other Pennaraptora. The proximal third of the bone tapers to a point; however, in contrast to many tetanuran theropods, in which metatarsal I tapers almost from the distal articular trochlea proximally with little or no shaft in between, there is a distinct shaft in between the articular end and the proximal point in *Archaeopteryx*, which accounts for slightly more than 1/3 of the length of the bone, as in *Sinornithosaurus* (*Xu & Wang, 2000*). Another unusual feature of metatarsal I is that the shaft is flexed posteromedially at about its mid-length, so that its medial margin is concave and its lateral margin convex longitudinally (Fig. 31). This curvature of the shaft of metatarsal I also seems to be present in the Solnhofen, Thermopolis (*Mayr et al., 2007*) and 11th specimens, but cannot be established for most specimens of *Archaeopteryx*, due to incomplete preservation or the way the feet are exposed.

The distal articular end of metatarsal I is proximodistally short and seems to have been slightly ginglymoidal. A well-developed, but not sharply defined extensor groove is present anteriorly, although this might be exaggerated by compression. The distal condyles are too poorly preserved to say anything about their detailed morphology.

A contentious issue concerning the foot of *Archaeopteryx* is the possible opposability of the first digit. Whereas many authors considered the first digit to be at least somewhat opposable towards the other digits of the pes (*Dames, 1884*; *de Beer, 1954*; *Wellnhofer, 1974*, *1988b*, *1992*, *1993*, *2008*; *Tarsitano & Hecht, 1980*), *Mayr, Pohl & Peters (2005)* and *Mayr et al. (2007)* recently argued that metatarsal I is attached medially, not posteromedially to metatarsal II and lacks the torsion that constitutes most to opposability in modern birds (*Middleton, 2001*), indicating that the condition in *Archaeopteryx* was similar to other non-avialan theropods in that the first digit is parallel to the other pedal digits. Specifically, *Mayr, Pohl & Peters (2005)* and *Mayr et al. (2007)* argued that, in the Thermopolis specimen, metatarsal I was attached medially to metatarsal II, with its distal end even being placed slightly dorsal (anterior) to the distal

ends of the other metatarsals, and that the first phalanx of digit I is exposed in dorsomedial view, similar to the other digits, which are exposed in dorsal view. The new specimen, however, seems to support the traditional view that the first digit was at least somewhat opposable. In the left foot, which is exposed in posterior view, metatarsal I is attached to the posteromedial side of metatarsal II (Fig. 31). Non-avialan theropods and also *Archaeopteryx* lack a distinct fossa for the attachment of metatarsal I on metatarsal II (the position of which is another indicator for opposability according to *Middleton, 2001*), so the exact position of the attachment should, however, be seen with caution, as post-mortem compression might have altered it. However, this argument is valid for both the posteromedial orientation seen in several specimens, including the new one described here, and the medial position in the Thermopolis specimen: if the digit was not fully opposed, but in a posteromedial position, compaction of the foot in the anteroposterior plane would move the metatarsal into a more medial position. This might be supported by the left foot of the Thermopolis specimen, which is exposed in slightly anteromedial view (WDC-CSG-100; *Mayr et al., 2007*). Whereas the observation by *Mayr et al. (2007)* that the distal end of metatarsal I is placed slightly dorsal (anterior) in relation to the other metatarsals is true for the right foot, the metatarsal is angled posterodistally in the left foot, displacing its distal end below the level of the other metatarsals.

However that may be, in the new specimen, the area medial to the attachment of metatarsal I as preserved is gently rounded and does not show a facet for the attachment of a metatarsal, indicating that, if displaced, the displacement is probably small. Furthermore, the curve in the shaft of metatarsal I would displace the distal end posteriorly even if the metatarsal was placed more medially on metatarsal II, and the same can be seen in the feet of the Eichstätt, Solnhofen, and 11th specimens. Furthermore, in the current specimen, the distal end of metatarsal I is visible in anterior view, whereas the other metatarsals are exposed in posteromedial view. Even accounting for slight rotation from a more medial position and compaction, the distal condyles would thus, at best, be placed at approximately 90° in relation to the distal condyles of the other metatarsals, facing laterally. Both of these observations, the posterior displacement and lateral orientation of the distal condyle, are confirmed by the right foot: although metatarsals III and IV are exposed in lateral and slightly anterior view, the distal end of metatarsal I is exposed posterior to the shaft of metatarsal IV, and its distal articular end faces laterally, with the supposedly posterior side (if aligned with the other metatarsals) facing even slightly anteriorly (Fig. 32). Thus, although metatarsal I of this specimen does not necessarily confirm a fully opposable digit I, it indicates that the orientation of this digit differed considerably from that of the other digits, and was not parallel to them (see also description of the phalanges below). A fully opposed first digit is present in all avians more derived than *Archaeopteryx*, such as *Jeholornis* (*Zhou & Zhang, 2002*) and *Sapeornis* (*Zhou & Zhang, 2003b*).

Metatarsals II to IV are closely appressed, with only their distal articular ends slightly diverging (Fig. 31). In posterior view, metatarsal II is the broadest of the metatarsals, followed by metatarsal IV (Fig. 31). This is different to the situation in troodontids (*Kurzanov & Osmólska, 1991*; *Currie & Peng, 1993*; *Xu et al., 2002*; *Xu & Wang, 2004*;

*Zanno et al., 2011*) and dromaeosaurids (*Ostrom, 1969*; *Norell & Makovicky, 1999*; *Hwang et al., 2002*), where metatarsal IV is significantly more robust than metatarsal II, but similar to the condition in *Confuciusornis* (*Chiappe et al., 1999*). Metatarsal III is slightly more slender than metatarsals II and IV at the proximal end, and almost completely pinched between the other two elements distally.

As noted above, metatarsal II extends more proximally than metatarsals III and IV to reach the same proximal level as the distal tarsal capping the other two metatarsals (Fig. 31). Its proximal end is slightly convex anteroposteriorly and shows a well-developed, smooth articular surface, bordered by a gently raised rim, distal from which the bone surface is slightly pitted. The proximal part of the shaft is flattened laterally for approximately two-thirds of its length, whereas the medial side is rounded anteroposteriorly. Just below mid-length, proximal to the attachment of metatarsal I, the posterolateral edge is drawn out into a marked ridge (Fig. 31). Distal to this ridge, the lateral side becomes more rounded. The distal end flexes slightly medially and seems to be slightly wider than deep, probably in contrast to the proximal part and mid-shaft. A slender, but well-developed flexor ridge is present on the posterior side of the distal end medially, but no collateral ligament groove seems to be present on the medial side of the metatarsal, although the bone is slightly damaged here. The distal end is well rounded anteroposteriorly, but does not seem to be ginglymoidal, in contrast to the situation in dromaeosaurids (*Ostrom, 1969*; *Norell & Makovicky, 1999*), although this cannot be said with absolute certainty due to articulation of the digit with the metatarsal.

Metatarsal III has a flattened posterior side proximally, but becomes triangular in cross-section towards the mid-shaft, tapering posteriorly (Fig. 31). The anterior side becomes flattened distally, and there seems to have been a shallow extensor groove on its anterior side at the distal end (Fig. 32). The distal articular end seems to be more posteriorly than anteriorly expanded and is well-rounded anteroposteriorly, being not or only very slightly ginglymoidal. A well-developed collateral ligament groove is present at least laterally and is placed slightly anterior to the mid-depth of the articular end. The distal end seems to be wider than deep, but not much can be said about the condition at mid-shaft. Just proximal to the ligament groove, a depression is present on the lateral side of the metatarsal, marking the attachment with the distal end of metatarsal IV.

The proximal end of metatarsal IV is flattened medially, where it contacts metatarsal III, but a posterior medial process overlapping the posterior side of the proximal end of metatarsal III, as it is present in many basal tetanurans (*Madsen, 1976*; *Currie & Zhao, 1993a*) is missing (Fig. 31). In posterior view, the shaft narrows gradually over its proximal three-fourths, until it reaches its narrowest point just where the shaft starts to deflect slightly laterally from metatarsal III. There seems to be a slight longitudinal depression on the posterior side of the shaft, although it is certainly exaggerated by compaction over most of the length of the bone. Thus the posterior side is rather flattened, being offset from both the medial and the lateral sides by marked edges. The distal end extends slightly further distally than the end of metatarsal II, but not as far as metatarsal III. It is slightly expanded both anteriorly and posteriorly and is transversely narrow anteriorly, whereas the posterior part is drawn out laterally (Fig. 32). A posterior tubercle is only present

medially, and a collateral ligament groove is absent laterally. A shallow, not clearly defined extensor groove is present on the anterior side. Both the shaft and the distal end are deeper anteroposteriorly than broad transversely.

Metatarsal V is a slender rod of bone that is attached posterolaterally to metatarsal IV and is slightly less than one-fourth of the length of the latter (Figs. 31 and 32), similar to the condition in other basal avialans (*Chiappe et al., 1999*; *Zhou & Zhang, 2002*, *2003b*). Its proximal end seems to be slightly deeper anteroposteriorly than wide transversely, but without being markedly flattened, as it is the case in many basal tetanurans (*Rauhut, 2003*). Distally, the shaft is slightly flexed medially and tapers (Fig. 31), lacking an articular end or phalanges, as in all neotheropods. The shaft is broken in both sides, revealing a thin-walled, hollow condition for the bone.

The **digits** of the feet are complete on both sides, but partially overlapping each other as exposed. The phalangeal formula is 2-3-4-5-0, as in the Berlin, Eichstätt, Munich, Thermopolis and 11th specimens (and most other theropods), but in contrast to the Solnhofen specimen, which only has four phalanges on the fourth digit in the only completely preserved right pes (*Wellnhofer, 1988b*, *1992*; *Elzanowski, 2001b*). The situation in the Maxberg (*Heller, 1959*) and Haarlem specimens is uncertain, although *Ostrom (1972)* and *Wellnhofer (2008*, *2009)* assumed five phalanges to be present in the latter. However, as most phalanges are preserved as impressions in this specimen, nothing can be said with certainty. For the London specimen, *de Beer (1954)* assumed four phalanges to be present, but as the fourth digit in the only preserved foot is mainly covered by the second digit, nothing can be said with certainty.

Digit III is the longest digit, followed by digit IV (88.5% of digit III) and digit II (74% of digit III). Digit I is the shortest, being slightly half the length of digit II.

Both digits of the first toe are preserved and visible on either side. Phalanx I-1 is slightly longer than the ungual of this digit. Its proximal end is expanded transversely, and the proximal articular facet seems to be twisted in relation to the distal articular end (Fig. 33), similar to the condition found in manual phalanx I-1 in theropods (*Galton, 1971*). This twist in the articular ends slightly displaces the distal end medially in relation to the proximal end and thus helps to further bring the ungual of digit I in opposition to the other digits. The distal end of phalanx I-1 is transversely narrow and ginglymoidal, with well-developed, dorsally displaced collateral ligament grooves. The ungual of the first digit has a large flexor tubercle proximally that accounts for approximately one-third of the height of the proximal end. The bony claw is moderately curved, so that the distal tip is placed ventral to the ventral border of the proximal end, being displaced for approximately half the height of the proximal end in relation to the latter (if the claw is held with the proximal articulation being vertical). A single, well-developed claw groove is present, and the part of the ungual ventral to the groove is wider than the portion dorsal to it. The horny sheath of the claw is preserved on both sides (Fig. 33). It extends the claw for approximately half the length of the bony claw and follows the curve of the latter, so that the distal end of the horny claw is placed slightly more than the proximal height of the bony claw ventral to the ventral margin of the flexor tubercle.

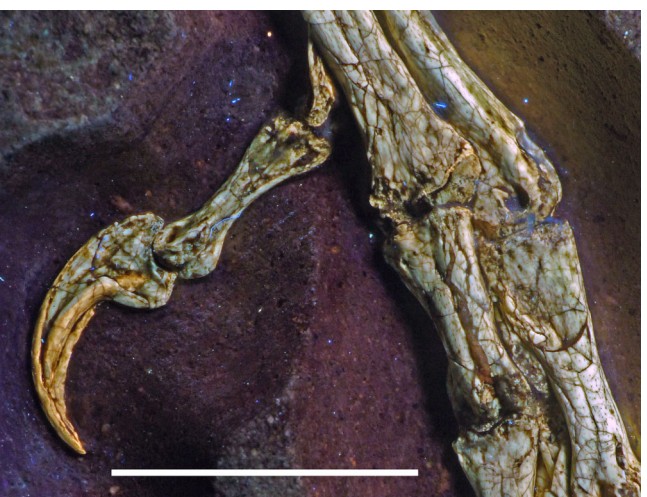

**Figure 33 Digit I of the pes of the 12th specimen of *Archaeopteryx*.** Photograph of the right digit I under UV light, showing the twist in phalanx I-1 and the horny sheath of the ungual. Scale bar is 10 mm.

In the second digit, the two non-ungual phalanges are subequal in length, which is different from the situation in various dromaeosaurids (*Ostrom, 1969*; *Forster et al., 1998*; *Xu & Wang, 2000*, *2004*) and troodontids (*Currie & Peng, 1993*; *Zanno et al., 2011*), in which the first phalanx is notably longer than the second. The first phalanx is elongate, but stout, and has a well-developed longitudinal ridge proximally on the medial margin of its ventral side; the lateral side of the phalanx is not visible in either foot. Distally, the phalanx has a strongly ginglymoidal articular end, which is extended proximally on the dorsal surface approximately as far as on the ventral surface, indicating a hyperextensible first toe, as in the Thermopolis (*Mayr, Pohl & Peters, 2005*; *Mayr et al., 2007*) and Eichstätt specimens (*Rauhut & Erickson, 2010*: fig. 5). The collateral ligament groove is centrally placed on the distal end of the medial side of the phalanx, and a broad, rounded, notable medial tubercle is present proximoventral to it. A well-developed extensor groove seems to be present on the dorsal side of the phalanx. The second phalanx is slightly more slender and has a flattened ventral surface, in contrast to many dromaeosaurids (*Ostrom, 1969*; *Rauhut & Werner, 1995*; *Norell & Makovicky, 1997*; *Forster et al., 1998*; *Novas et al., 2009*; *Turner, Makovicky & Norell, 2012*) and troodontids (*Currie & Peng, 1993*; *Zanno et al., 2011*), where PII-2 possesses a pronounced ventral heel proximally. The distal end is ginglymoidal, with widely and deeply separated, slender condyles, which extend considerably further proximally on the ventral than on the dorsal side. The collateral ligament groove is well-developed, seems to be deeper on the lateral than on the medial side and is displaced dorsally (Fig. 34). The ungual of the second digit is the largest ungual of the foot. Its morphology closely corresponds to that of the ungual of the first digit, but it is slightly more notably curved. In contrast to the situation in dromaeosaurids, which have an asymmetric arrangement of the claw grooves and a sharp ventral margin in the second pedal ungual (*Kirkland, Burge & Gaston, 1993*; *Rauhut & Werner, 1995*; *Norell & Makovicky, 1997*), the claw grooves are symmetrically or near-symmetrically arranged (based on a comparison of the left and right unguals, which are visible in medial and

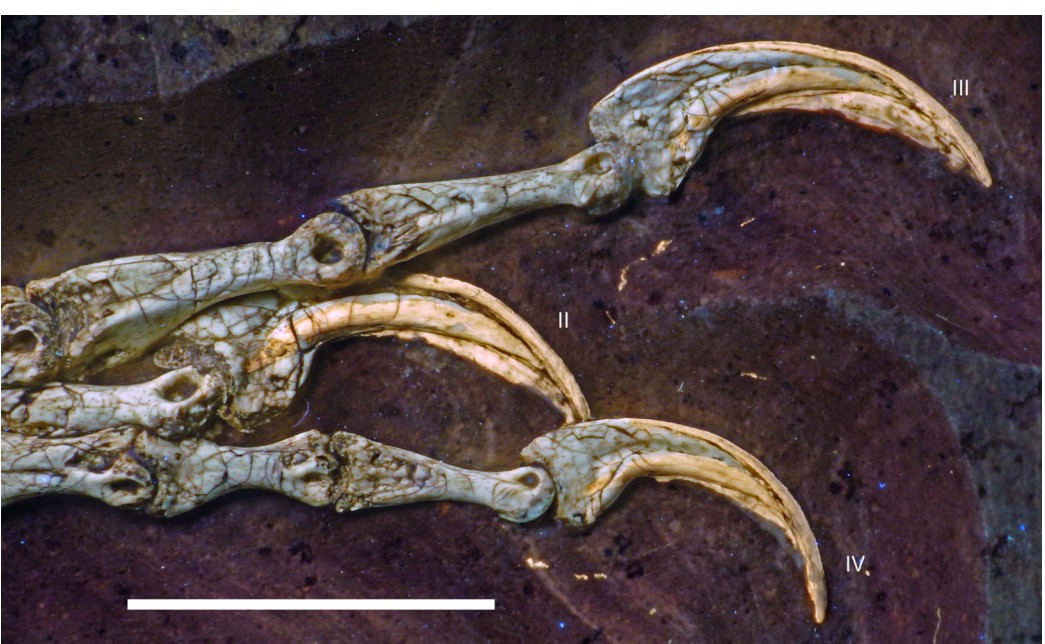

**Figure 34 Distal phalanges and unguals of the right pes of the 12th specimen of *Archaeopteryx*.** Photograph under UV light. Roman numerals denominate digits. Scale bar is 10 mm.

lateral view, respectively) and the ventral margin of the claw is broader than the dorsal margin and flattened.

The three non-ungual phalanges of the third digit become successively smaller, and phalanx III-1 is the largest of the foot. The phalanges are elongate, with strongly ginglymoidal distal ends. Whereas the collateral ligament grooves are centrally placed at the distal end of the lateral and medial side and are deeper on the lateral than on the medial side in the first two phalanges, the third phalanx has the grooves displaced dorsally (Fig. 34) and subequally developed on either side. Well-developed dorsal extensor grooves are present in phalanges III-1 and III-2, but apparently not in III-3. In at least the second and third phalanx of the digit, the ventral side is notably flattened proximally and offset from the lateral and medial sides by marked ventrolateral and ventromedial edges. The ungual of digit III is notably smaller than that of digit II and only slightly larger than ungual IV. It is also less strongly recurved and has a less bulbous and less notably expanded flexor tubercle than the ungual of the second digit (Fig. 34).

The four non-ungual phalanges of the fourth digit are less elongate than those of the second and third digit, as it is usual in theropods. The length successively decreases from phalanx IV-1 to IV-3, but then slightly increases again in phalanx IV-4. All phalanges are notably ginglymoidal and have well-developed collateral ligament grooves. The latter seem to be centrally placed in IV-1, but displaced dorsally in the other phalanges, most notably so in the penultimate phalanx (Fig. 34). Well-developed extensor grooves are present at least in phalanges IV-2 and IV-3 (Fig. 34), whereas only a shallow depression seems to be present in IV-1; nothing can be said in this respect about phalanx IV-4.

The ungual of digit IV is the smallest pedal ungual, subequal in size to ungual I, and has the least conspicuous flexor tubercle.

## DISCUSSION

### The monophyly of *Archaeopteryx* and referral of the new specimen to this taxon

Up to recently, the referral of new specimens from the Solnhofen Archipelago to the genus *Archaeopteryx*, or at least to the Archaeopterygidae (see *Elzanowski, 2002*) seemed unproblematic, as the known specimens (even more fragmentary ones, such as the Haarlem specimen or the ninth specimen) were all sufficiently similar to each other, and no other derived paravian theropod was known from the Late Jurassic. However, with the discovery of derived paravian and probably even avialan theropods that are similar to *Archaeopteryx* from the Late Jurassic of China in recent years (see *Xu et al., 2009a*, *2011*; *Hu et al., 2009*; *Godefroit et al., 2013a*, *2013b*), the question arises if the specimens referred to *Archaeopteryx* (or the Archaeopterygidae) really represent a monophyletic taxon. Indeed, a recent re-examination of the fourth (Haarlem) specimen of *Archaeopteryx* suggests that this specimen represents an anchiornithid rather than the genus *Archaeopteryx* (*Foth & Rauhut, 2017*).

*Elzanowski (2002)* gave a tentative diagnosis of the family Archaeopterygidae, which, in his work, included all specimens of *Archaeopteryx* then known, including the Solnhofen specimen, which he considered to be a separate genus, *Wellnhoferia* (*Elzanowski, 2001b*; but see also *Mayr et al., 2007*). In the same paper, *Elzanowski (2002)* also gave diagnoses for the two genera within the Archaeopterygidae, but these diagnoses were mainly aimed to separate *Wellnhoferia* from *Archaeopteryx*, and do not help too much in the question whether all the paravian specimens from the lithographic limestones of southern Germany represent a monophyletic taxon (see also discussion in *Mayr et al., 2007*: 113–114).

Furthermore, many of the characters listed in the diagnosis of the Archaeopterygidae (*Elzanowski, 2002*: 131–132) have a wider distribution among theropods or even represent the plesiomorphic condition within this clade. Thus, the presence of four premaxillary teeth, a premaxilla that projects anteriorly beyond the mandible, a furcula without hypocleideum, and the presence of 23 presacral and five sacral vertebrae all represent the normal character states for non-avialan theropods (see e.g. *Raath, 1977*; *Currie & Zhao, 1993a*; *Brochu, 2003*; *Norell & Makovicky, 2004*; *Nesbitt et al., 2009*; *Rauhut et al., 2012*), and a wing-shaped preglenoid process of the coracoid is the usual condition in non-avialan paravians and basal avialans (*Ostrom, 1974*; *Norell & Makovicky, 1999*; *Zhou & Zhang, 2002*, *2003a*, *2003b*; *Tsuihiji et al., 2014*).

Unusual features listed by *Elzanowski (2002)* include the 'intermediate process' on the ischium, the lack of a mandibular fenestra, the presence of only eight to nine maxillary teeth, and the basally rounded teeth with recurved tips. However, even most of these characters are also not unique to *Archaeopteryx*. As noted above, a triangular posterior process on the ischium ('intermediate process') is also found in the troodontids

*Sinovenator* (where it is, however, more proximally placed; *Xu et al., 2002*) and *Jianianhualong* (*Xu et al., 2017*), the dromaeosaurid *Microraptor* (*Hwang et al., 2002*), and, most significantly, in the possible close relative of *Archaeopteryx*, *Xiaotingia* (*Xu et al., 2011*).

The complete absence of a mandibular fenestra seems to be a rare character state in theropod dinosaurs; some taxa that were said to not have this opening, such as *Ornitholestes* (*Osborn, 1916*) and *Juravenator* (*Chiappe & Göhlich, 2010*), actually have a mandibular fenestra, which is difficult to identify due to compression of the specimens (O. W. M. Rauhut, 2015, personal observation in AMNH 619; see also *Paul, 1988*; O. W. M. Rauhut and C. Foth, 2014, personal observation in JME Scha 200). However, this opening is absent in *Compsognathus* (BSPG AS I 563; *Ostrom, 1978*; *Peyer, 2006*), and, more importantly, the paravian genera *Anchiornis* and *Eosinopteryx* have been figured as lacking a mandibular fenestra (*Hu et al., 2009*; *Godefroit et al., 2013a*). It should be noted, though, that these taxa are based on strongly compressed specimens with rather poor bone preservation, and although *Pei et al. (2017)* state that an external mandibular fenestra is absent in *Anchiornis*, their figures 5 and 8 might show parts of the margins of such an opening. Most other paravians have large mandibular fenestrae (*Chiappe et al., 1999*; *Turner, Makovicky & Norell, 2012*; *Tsuihiji et al., 2014*), so that the condition in *Archaeopteryx* is at least unusual, but if it represents a local autapomorphy of the taxon has to be shown by more detailed studies of the paravian taxa from the Tiaojishan Formation.

With eight to nine maxillary teeth in the known specimens, the number of teeth in this bone is unusually low for non-avialan theropods. However, a similar number of teeth is found in *Dromaeosaurus* (nine maxillary teeth; *Currie, 1995*), and most other dromaeosaurids have only a few teeth more (see *Xu & Wu, 2001*; *Currie & Varricchio, 2004*; *Pei et al., 2014*). On the other hand, most more derived avialan theropods that retain teeth in the maxilla seem to have less teeth (*O'Connor & Chiappe, 2011*; *Wang et al., 2017a*). Thus, the number of maxillary teeth might help to differentiate *Archaeopteryx* from other paravian theropods, but cannot be used as an autapomorphy of the taxon based on our current knowledge, as it might represent an intermediate stage between non-avialan paravians and more derived avialans.

The last character mentioned by *Elzanowski (2002)*, the tooth crowns with a bulbous or straight base and a recurved tip, was also used in slightly modified form by *Foth, Tischlinger & Rauhut (2014)* to refer the 11th specimen to *Archaeopteryx*. As discussed below, tooth morphology is quite variable between different specimens of *Archaeopteryx*. However, in all specimens where teeth are preserved, the premaxillary and anterior dentary teeth have a peculiar morphology that indeed seems to be very rare in theropods. In these teeth, the basal third to half of the crown has straight to slightly convex mesial and distal margins that slightly converge apically, before the apical part of the crown forms a bulbous mesial and less marked distal expansion. Whereas the distal margin of the crown is straight to slightly concave apical to this expansion, the mesial margin is strongly convex, so that the tip of the tooth is placed approximately above the margin of the distal expansion. This morphology is most marked in the premaxillary teeth; however, the anteriormost dentary teeth show a similar condition, just with the mesial expansion being

less marked than in the premaxillary teeth. Apart from *Archaeopteryx,* a similar tooth morphology is only found in the Early Cretaceous (Aptian) pygostylian avialan *Sapeornis* from Liaoning, China (*Wang et al., 2017a*). However, the teeth of this taxon differ in some details, such as the more triangular crowns and the presence of striations on the tooth crowns (*Wang et al., 2017a*). Thus, the exact tooth morphology of the premaxillary and anterior maxillary and dentary teeth might well be regarded as unique in *Archaeopteryx.* Fortunately, one premaxillary tooth with exactly this morphology is preserved in the London specimen (*Wellnhofer, 2008*: fig. 5.27), so the presence of this character can be demonstrated for the neotype specimen of *Archaeopteryx lithographica,* the type species of the genus *Archaeopteryx* (*ICZN, 2011*; see also *Bühler & Bock, 2002*; *Bock & Bühler, 2007*).

A probable cranial autapomorphy of *Archaeopteryx* is the presence of a depressed rim around the posterior margin of the trigeminal foramen, which was interpreted as a pneumatic cavity in the London specimen by *Walker (1985)*. This rim is also present in the Munich specimen (*Rauhut, 2014*), but has not been described or observed by us in any other theropod. Unfortunately, however, braincase descriptions for many paravian and especially basal avialan taxa are still sparse, and thus the interpretation of this character as an autapomorphy of *Archaeopteryx* should be seen as tentative. However, such a depressed rim is absent in the dromaeosaurids *Dromaeosaurus* (*Currie, 1995*), *Velociraptor* (*Norell, Makovicky & Clark, 2004*), and *Tsaagan* (*Norell et al., 2006*), the troodontids *Sinovenator* (*Xu et al., 2002*), *Stenonychosaurus* (*Currie, 1985*; *Currie & Zhao, 1993b*), *Byronosaurus* (*Makovicky et al., 2003*), and *Zanabazar* (*Norell et al., 2009*), and the more derived avialans *Enaliornis* (*Elzanowski & Galton, 1991*) and *Ichthyornis* (*Clarke, 2004*).

A combination of several unusual, although possibly not apomorphic characters might further help to distinguish *Archaeopteryx* from other paravian theropods. These characters include the presence of only eight to nine maxillary teeth, the lack of a mandibular fenestra, and the presence of an 'intermediate process' on the ischium, which were discussed above. Further unusual features include the marked incision between the postorbital and quadratojugal processes of the jugal, resulting in a rounded, triangular, sharply angled anteroventral margin of the infratemporal fenestra, and the very slender quadratojugal process of this bone, which were also used by *Foth, Tischlinger & Rauhut (2014)* to refer the 11th specimen to *Archaeopteryx.* However, as with the other characters discussed above, these features are not unique to *Archaeopteryx.* Whereas the normal condition of a quadratojugal process with a maximal dorsoventral height that is subequal to the minimal height of the jugal below the orbit is found in most paravians, such as *Dromaeosaurus* (*Currie, 1995*), *Deinonychus* (*Ostrom, 1969*), *Velociraptor* (*Barsbold & Osmólska, 1999*), *Tsagaan* (*Norell et al., 2006*), *Linheraptor* (*Xu et al., 2015a*), *Gobivenator* (*Tsuihiji et al., 2014*), and *Confuciusornis* (*Chiappe et al., 1999*), a few taxa show a very slender quadratojugal process, including *Microraptor* (*Pei et al., 2014*), *Bambiraptor* (*Burnham, 2004*), *Xiaotingia* (*Xu et al., 2011*), and *Anchiornis* (*Pei et al., 2017*). However, the incision of the infratemporal fenestra into the posterior margin of the jugal differs between the latter three taxa and *Archaeopteryx.* In *Bambiraptor, Xiaotingia* and *Anchiornis,* the margin of the fenestra curves gradually from the postorbital process onto the quadratojugal process. In contrast, in *Archaeopteryx* and *Microraptor* (*Pei et al., 2014*),

the incision between the two processes is sharp-angled, and the quadratojugal process is offset from the postorbital process. Given our current understanding of paravian phylogeny (*Turner, Makovicky & Norell, 2012*; *Foth, Tischlinger & Rauhut, 2014*), this morphology was probably acquired independently in these two taxa, and might thus represent a local autapomorphy of *Archaeopteryx*.

In the Munich (BSPG 1999 I 50), 11th (*Foth, Tischlinger & Rauhut, 2014*) and the new specimen, *Archaeopteryx* shows another unusual character in the jugal, the presence of a longitudinal groove on the medial side of the suborbital process of this bone. In the Thermopolis specimen, the compaction of this part of the jugal (which is exposed in lateral view) indicates that this character is also present. The only other theropod, for which such a longitudinal groove has been described, is *Anchiornis* (*Hu et al., 2009*), but a groove seems also to be present on the ventral part of the medial side in the anterior half of the suborbital process in *Bambiraptor* (*Burnham, 2004*: fig. 3.8B).

Finally, a possible autapomorphy of *Archaeopteryx* might be the number of cervical and dorsal vertebrae. As noted by *Wellnhofer (2008, 2009)*, well-preserved specimens of this taxon have nine cervical and 14 dorsal vertebrae, and the presence of 14 dorsals in the new specimen indicate that this was also the case here, although the cervical column is incomplete. In non-avialan theropods, the number of cervical vertebrae is sometimes difficult to establish due to the presence of one or more 'transitional' vertebrae at the base of the neck, sometimes called pectorals (see e.g. *Welles, 1984*), and several authors have assumed the number of cervical vertebrae and dorsal vertebrae to be nine and 14, respectively (*Lambe, 1917*; *Gilmore, 1920*; *Madsen, 1976*; *Zhao & Currie, 1993*; *Zhao et al., 2010*). However, all non-maniraptoran theropods in which the vertebral column and the ribs or rib heads are preserved have 10 cervical vertebrae, including coelophysoids (*Raath, 1977*; *Colbert, 1989*; *You et al., 2014*), ceratosaurs (*Bonaparte, Novas & Coria, 1990*; *O'Connor, 2007*), megalosauroids (*Sadleir, Barrett & Powell, 2008*; *Rauhut et al., 2012*), allosauroids (*Currie & Zhao, 1993a*; *Ortega, Escaso & Sanz, 2010*; *Evers, 2014*), tyrannosauroids (*Brochu, 2003*; *Brusatte, Carr & Norell, 2012*), compsognathids (*Currie & Chen, 2001*; *Peyer, 2006*), and ornithomimosaurs (*Osmólska, Roniewicz & Barsbold, 1972*; *Kobayashi & Lü, 2003*), and we assume this count to represent the plesiomorphic condition in coelurosaurs. Accordingly, the general number of dorsal vertebrae is 13, except in cases where the number of sacral vertebrae has been increased, in which case the last dorsal might be incorporated into the sacral complex (*O'Connor, 2007*; *Rauhut & Carrano, 2016*). In maniraptorans, the general number of 10 cervicals seems to be retained at least for basal members of most clades. Thus, the basal therizinosauroid *Jianchangosaurus* (*Pu et al., 2013*), the basal alvarezsauroid *Haplocheirus* (J. N. Choiniere, 2016, personal communication to OR), the basal troodontid *Liaoningvenator* (*Shen et al., 2017*), and dromaeosaurids, including the basal form *Microraptor* (*Pei et al., 2014*) and more derived taxa (*Norell et al., 2006*; *Xu et al., 2010*) have 10 cervical vertebrae. Likewise, cervical count is uncertain or has not been given for many basal avialans; however, in those taxa in which a count is given, there are 10 or more cervicals, e.g. in *Anchiornis* (10 cervicals; *Pei et al., 2017*), *Yanornis* (at least 10 cervicals; *Zhou & Zhang, 2001*), *Sapeornis* (10 cervicals; *Zhou & Zhang, 2003b*), *Archaeorhynchus* (at least 10

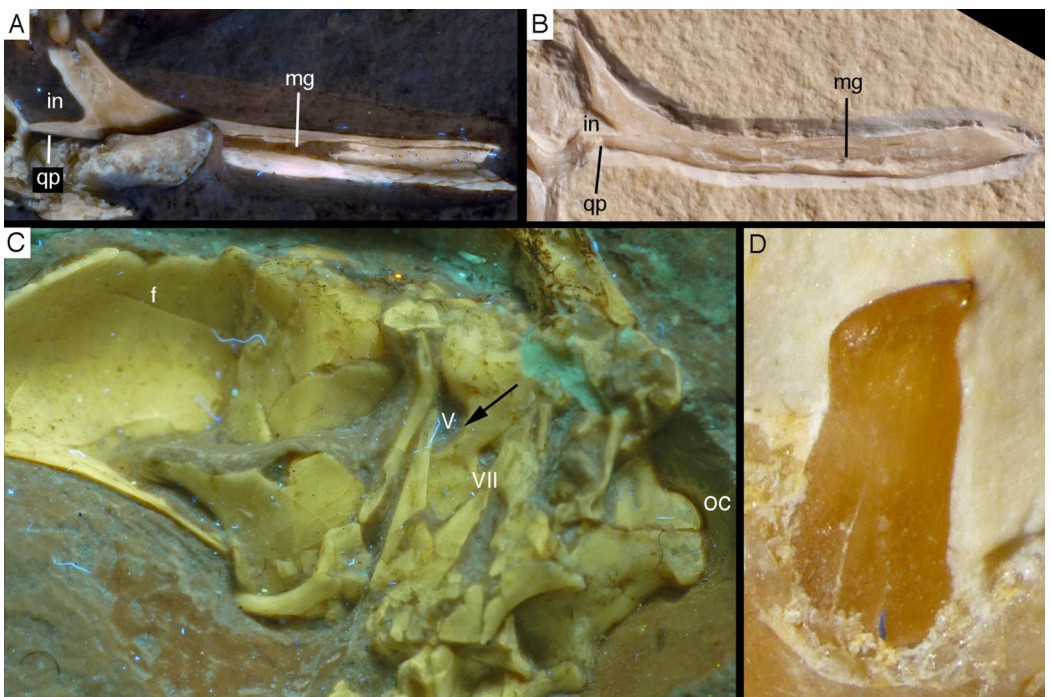

**Figure 35** **Diagnostic characters of the genus *Archaeopteryx*.** (A) Jugal of the 11th specimen in medial view under UV light. (B) Jugal of the 7th (Munich) specimen in medial view. The median longitudinal groove, sharply angled posterior incision and very slender quadratojugal process can be seen in both of these elements. (C) Braincase of the 7th (Munich) specimen under UV light, showing the posterior rim of the trigeminal foramen (arrow; see also *Rauhut, 2014*). (D) Fourth premaxillary tooth of the 11th specimen, illustrating the unique premaxillary tooth shape of *Archaeopteryx*. f, frontal; in, incision between the postorbital and quadratojugal process of the jugal; mg, medial groove; oc, occipital condyle; qp, quadratojugal process. Roman numbers denominate foramina for cranial nerves V and VII.

cervicals; *Zhou & Zhang, 2006*), *Eoenantiornis* (11 cervicals; *Zhou, Chiappe & Zhang, 2005*), and *Yixianornis* (probably 12 cervicals; *Clarke, Zhou & Zhang, 2006*).

In summary, the genus *Archaeopteryx* cannot be diagnosed on the basis of any single apomorphic characters, but only by an apomorphic set of combined characters. In consequence, we suggest the following **emended differential diagnosis** for the genus *Archaeopteryx*:

Maxilla with only eight to nine tooth positions; jugal with a longitudinal groove on the medial side of the suborbital process (Figs. 35A and 35B); anteroventral margin of the infratemporal fenestra forms a sharply angled incision separating the quadratojugal and postorbital processes in the jugal (Figs. 35A and 35B); quadratojugal process of the jugal very slender, less than half the height of the minimal height of the suborbital process (Figs. 35A and 35B); depressed rim around the posterior margin of the trigeminal foramen in the prootic (possibly autapomorphic; Fig. 35C); mandibular fenestra absent (probable local autapomorphy); premaxillary and anteriormost dentary teeth with straight basal part and bulbous, strongly recurved apical part (Fig. 35D); nine cervical and fourteen dorsal vertebrae; ischium with well-developed, triangular 'intermediate process' posteriorly at about mid-shaft (Figs. 28 and 36).

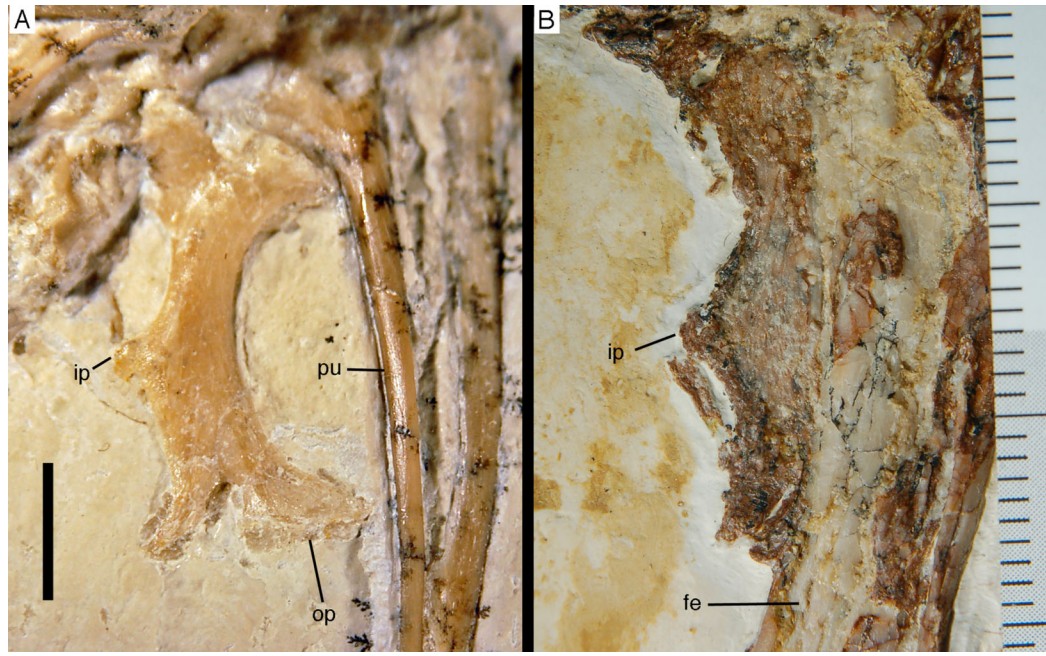

**Figure 36 The 'intermediate process' in the ischium of *Archaeopteryx*.** (A) Ischium of the 10th (Thermopolis) specimen. (B) Ischium of the 6th (Solnhofen) specimen. fe, femur; ip, intermediate process; op, obturator process; pu, pubis. Scale bar is 5 mm (A) and in mm intervals (B).

Due to the lack of most of the skull roof and lower jaw in the neotype of *Archaeopteryx*, the London specimen (*de Beer, 1954*; *Wellnhofer, 2008*, *2009*), only some of these characters can actually be established in this specimen. However, this specimen shows the highly unusual tooth morphology, the rim at the posterior margin of the trigeminal foramen and the 'intermediate process' of the ischium (*Walker, 1985*; *Wellnhofer, 2008*), which makes a correlation with several other specimens and thus a test for the remaining characters as possible unique character combination for *Archaeopteryx* possible.

Many of the other specimens can be referred to *Archaeopteryx* with certainty, as they show a combination of several of the characters listed above (Table 2). Thus, the Berlin specimen has only eight or nine maxillary teeth (*Dames, 1884*; *Wellnhofer, 2008*), shows the characteristic shape of the premaxillary teeth (*Howgate, 1984a*, *1984b*), has nine cervical vertebrae, and has an 'intermediate process' on the ischium (*Wellnhofer, 2008*). The Eichstätt specimen has eight maxillary teeth, a sharply angled anteroventral margin of the infratemporal fenestra, a very slender quadratojugal process of the jugal, lacks a mandibular fenestra, shows the typical premaxillary tooth morphology, nine cervicals and 14 dorsals, and has an 'intermediate process' on the ischium (*Wellnhofer, 1974*; *Howgate, 1984a*; *Elzanowski & Wellnhofer, 1996*). The Solnhofen specimen has the characteristic premaxillary dentition and an 'intermediate process' on the ischium (Fig. 36B; *Wellnhofer, 1988b*, *1992*). The Munich specimen shows a medial longitudinal groove in the jugal (Fig. 35B), a sharply angled incision between the postorbital and jugal processes of the jugal (Fig. 35B), a very slender quadratojugal process of the jugal (Fig. 35B), a depressed

**Table 2 Overview of specimens of *Archaeopteryx* referred to in the text and their taxonomic referral (up to genus level, as the species taxonomy of *Archaeopteryx* remains unclear).**

| No | Name | Diagnostic characters of *Archaeopteryx* preserved | Taxonomic identification | Important references[*] |
|---|---|---|---|---|
| 1 | London specimen | Shape of premaxillary teeth; rimmed posterior margin of trigeminal foramen; intermediate process of the ischium | *Archaeopteryx lithographica* (neotype of type species) | *Owen (1863)*, *de Beer (1954)*, *Ostrom (1976)*, *Walker (1985)*, *Dominguez Alonso et al. (2004)* |
| 2 | Berlin specimen | Eight to nine maxillary teeth; shape of premaxillary teeth; lack of mandibular fenestra; nine cervical vertebrae; intermediate process of the ischium | *Archaeopteryx* sp. | *Dames (1884)*, *Ostrom (1976)*, *Howgate (1984a)* |
| 3 | Maxberg specimen | None visible, but generally compatible with known anatomy of *Archaeotperyx* | cf. *Archaeopteryx* sp. | *Heller (1959)* |
| 4 | Haarlem specimen | None | Distinct genus, *Ostromia* | *Ostrom (1972)*, *Foth & Rauhut (2017)* |
| 5 | Eichstätt specimen | Eight maxillary teeth; shape of premaxillary teeth; sharply angled incision of infratemporal fenestra in jugal; very slender quadratojugal process of jugal; lack of mandibular fenestra; nine cervical vertebrae; intermediate process of the ischium | *Archaeopteryx* sp. | *Wellnhofer (1974)*, *Ostrom (1976)*, *Elzanowski & Wellnhofer (1996)*, *Howgate (1984a, 1985)* |
| 6 | Solnhofen specimen | Shape of premaxillary teeth; intermediate process of ischium | *Archaeopteryx* sp. | *Wellnhofer (1988b, 1992)*, *Elzanowski (2001b)* |
| 7 | Munich specimen | Shape of anterior dentary teeth; sharply angled incision of infratemporal fenestra in jugal; very slender quadratojugal process of jugal; longitudinal groove on medial side of jugal; rimmed posterior margin of trigeminal foramen; lack of mandibular fenestra; nine cervical vertebrae; intermediate process of the ischium | *Archaeopteryx* sp. | *Wellnhofer (1993)*, *Elzanowski & Wellnhofer (1996)*, *Wellnhofer & Tischlinger (2004)*, *Rauhut (2014)* |
| 8 | Daiting specimen | Nine maxillary teeth; shape of anterior maxillary teeth | *Archaeopteryx* sp. | *Tischlinger (2009)* |
| 9 | 'Chicken wing' | None preserved, but general anatomy is largely consistent with *Archaeopteryx*; unusually long metacarpal I | cf. *Archaeopteryx* sp. | *Wellnhofer & Röper (2005)* |
| 10 | Thermopolis specimen | Eight maxillary teeth; shape of premaxillary teeth; sharply angled incision of infratemporal fenestra in jugal; very slender quadratojugal process of jugal; longitudinal groove on the medial side of the jugal; intermediate process of the ischium | *Archaeopteryx* sp. | *Mayr et al. (2007)*, *Rauhut (2014)* |
| 11 | 11th specimen | Shape of premaxillary teeth; sharply angled incision of infratemporal fenestra in jugal; very slender quadratojugal process of jugal; longitudinal groove on medial side of jugal; intermediate process of the ischium | *Archaeopteryx* sp. | *Foth, Tischlinger & Rauhut (2014)* |
| 12 | 12th specimen | Nine maxillary teeth; shape of premaxillary teeth; sharply angled incision of infratemporal fenestra in jugal; very slender quadratojugal process of jugal; lack of mandibular fenestra; intermediate process of the ischium | *Archaeopteryx* sp. | |

**Note:**
[*] In addition to the review papers by *Elzanowski (2002)*, which dealt with the first to seventh specimen and the monographic treatment of the 1st to 10th specimen by *Wellnhofer (2008, 2009)*. Listed here are mainly important descriptive papers, not necessarily first mentionings or short descriptions.

rim of the trigeminal foramen on the prootic (Fig. 35C), lacks a mandibular fenestra, has the diagnostic tooth morphology in the anteriormost dentary teeth, nine cervical vertebrae, and an 'intermediate process' on the ischium (*Wellnhofer, 1993*; *Elzanowski & Wellnhofer, 1996*; *Rauhut, 2014*). The Daiting specimen does, unfortunately, not preserve most of the elements needed for a secure identification (*Tischlinger, 2009*), but the anteriormost preserved maxillary tooth shows a straight basal part with a strongly recurved, anteriorly bulging apical part of the crown (O. W. M. Rauhut, 2017, personal observation), and thus conforms to the characteristic premaxillary tooth morphology seen in other *Archaeopteryx* specimens. The Thermopolis specimen has eight maxillary teeth, a longitudinal groove on the medial side of the suborbital process of the jugal, a sharply angled incision between the postorbital and quadratojugal processes of the jugal, a very slender quadratojugal process of the jugal, the typical premaxillary tooth morphology, and a well-developed 'intermediate process' on the ischium (Fig. 36A; *Mayr et al., 2007*; *Rauhut, 2014*). Likewise, the 11th specimen also shows all of these characters, with the exception of the maxillary tooth count, which cannot be evaluated (*Foth, Tischlinger & Rauhut, 2014*). The new specimen can be referred to *Archaeopteryx* with certainty, as it shows the following diagnostic characters: nine maxillary tooth positions, longitudinal groove on the medial side of the suborbital part of the jugal, sharply angled incision between the postorbital and quadratojugal process of the jugal, very slender quadratojugal process of the jugal, lack of mandibular fenestra, diagnostic premaxillary tooth morphology, and presence of an 'intermediate process' on the ischium.

More problematic are those specimens of *Archaeopteryx*, in which none of the elements that show diagnostic characters are preserved or visible, including the Maxberg and ninth specimens (*Heller, 1959*; *Wellnhofer & Röper, 2005*; *Wellnhofer, 2008*, *2009*). Unfortunately, only published information on the Maxberg specimen (*Heller, 1959*; *Ostrom, 1976*; *Wellnhofer, 1985*, *2008*, *2009*) and casts are available, as the original specimen remains lost (*Wellnhofer, 2008*). Based on the descriptions and a cast (SNSB-BSPG 1992 I 9), the specimen seems to closely correspond to other specimens of *Archaeopteryx* in all visible details, as well as its skeletal proportions, and can thus also be tentatively referred to this genus, although a positive identification is currently not possible.

Finally, the ninth specimen ('chicken wing') only preserves the skeleton of the right forelimb (*Wellnhofer & Röper, 2005*). Although the specimen is comparable to other *Archaeopteryx* specimens in its anatomical details and general wing proportions (*Wellnhofer, 2008*), it differs in a few proportions, most notably the relatively longer metacarpal I if compared to certain *Archaeopteryx* specimens. Thus, any referral of this specimen to the genus *Archaeopteryx* should currently be regarded as tentative.

*Weigert (1995)* referred isolated teeth from the Kimmeridgian of Guimarota, Portugal, to cf. *Archaeopteryx* (see also *Wiechmann & Gloy, 2000*), which would represent the only record of this taxon outside the area of the Solnhofen Archipelago. These teeth are similar to the premaxillary, anterior maxillary and anterior dentary teeth of *Archaeopteryx* in that they have a rather straight basal and a recurved apical part, as well as a slight bulbous expansion mesially above the straight part. However, the teeth also

differ from those found in the known specimens of *Archaeopteryx* in the presence of a bulbous expansion distally in the basal part. Furthermore, *Wellnhofer (2008)* did not find any evidence for the strongly lingually curved mesial carina or the rudimentary denticles on the carina found in the Portuguese specimens on the medially exposed dentary teeth of the Munich specimen of *Archaeopteryx*, and our own observation of the medially exposed premaxillary and dentary teeth of the 11th specimen and the dentary teeth of the 12th specimen described here confirm this observation. Thus, although the potentially apomorphic similarities of the Portuguese teeth might indicate that these elements represent a closely related taxon, they cannot currently be referred to *Archaeopteryx*. A single tooth crown from the Beriasian of France was recently also referred to the Archaeopterygidae (*Louchart & Pouech, 2017*), due to general similarity to the teeth of *Archaeopteryx*. However, the tooth differs from the typical teeth of *Archaeopteryx* in the more compressed and less bulbous shape, and, given the presence of similar teeth in other basal avialans (*Wang et al., 2017a*), should currently only been regarded as an indeterminate probable avialan. Likewise, fragmentary bones from the Berriasian of Romania that were originally referred to *Archaeopteryx* (*Kessler, 1984*; *Kessler & Jurcsák, 1984*) have since been shown to represent undiagnostic remains (*Dyke et al., 2011*). Thus, the genus *Archaeopteryx* is so far only securely known from the lower Tithonian of the southern Franconian Alb in Bavaria, Germany.

## Considerations on the osteology of *Archaeopteryx*
### Cranial osteology

**Postorbital.** Although *Archaeopteryx* is a rather well-known taxon, and numerous anatomical descriptions have been published, the detailed analysis of the new specimen resulted in several insights into its osteology. Most importantly, for the first time, the postorbital is well preserved in a specimen of *Archaeopteryx* (Figs. 6, 7 and 9–11). Although the presence of this bone and of a closed postorbital bar had been assumed for this taxon several times (*Wellnhofer, 1974*; *Tischlinger, 2005*; *Rauhut, 2014*) many recent reconstructions of the skull show *Archaeopteryx* without or with a reduced postorbital and an open postorbital bar (*Ji & Ji, 2007*; *Wellnhofer, 2008*, *2009*; *Xu et al., 2011*; *Wang & Hu, 2017*), a condition that is important or even essential for the inference of an avian-style cranial kinesis in this taxon (see *Bühler, 1985*; *Wellnhofer, 2008*, *2009*). The new specimen confirms the presence of a triradiate postorbital, with a long ventral process that reaches the postorbital process of the jugal, as reconstructed by *Wellnhofer (1974)* and *Rauhut (2014)*. As discussed by *Rauhut (2014)*, this indicates that an avian style cranial kinesis was absent in *Archaeopteryx*.

The postorbital of *Archaeopteryx* conforms well with the same bone in other paravians in that the anterior process is upturned, resulting in a Y-shaped postorbital (Figs. 6, 7 and 9), rather than the T-shaped element seen in basal tetanurans, including basal coelurosaurs, such as *Zuolong* (*Choiniere et al., 2010*) and tyrannosauroids (*Currie, 2003*; *Xu et al., 2004*). However, in contrast to most dromaeosaurids (see e.g. *Ostrom, 1969*; *Barsbold & Osmólska, 1999*; *Norell et al., 2006*; *Xu et al., 2010*; *Turner, Makovicky &*

*Norell, 2012*), troodontids (*Currie, 1985*; *Norell et al., 2009*), *Anchiornis* (*Hu et al., 2009*; *Pei et al., 2017*), and *Aurornis* (*Godefroit et al., 2013b*), in which the postorbital is a rather plate-like element with a notably short ventral process, this bone is slender in *Archaeopteryx*, with the ventral process being the longest of the three processes. This resembles the condition in more basal taxa, such as oviraptorosaurs (*Clark, Norell & Rowe, 2002*; *Balanoff & Norell, 2012*), but also seems to be the case in at least some slightly more derived avialans, such as *Pengornis* (*Zhou, Clarke & Zhang, 2008*; *O'Connor & Chiappe, 2011*), and in the troodontid *Gobivenator* (*Tsuihiji et al., 2014*).

**Prefrontal.** A separate prefrontal ossification has so far usually been said to be absent in *Archaeopteryx* (*Elzanowski, 2001a*, *2002*; *Mayr et al., 2007*; *Wellnhofer, 2008*, *2009*). This bone has previously only been postulated to be present in the Eichstätt specimen (*Wellnhofer, 1974*), although the element identified as such was later correctly reinterpreted as part of the lacrimal (*Elzanowski & Wellnhofer, 1996*). However, the prefrontal is actually preserved in both the Eichstätt and the Thermopolis specimens (Fig. 37). In the former, it is an elongate, splint-like bone that has been slightly displaced from the medial side of the lacrimal and the orbital margin to slightly obliquely overlie the frontal, where it is visible in dorsal view (Fig. 37A). In the Thermopolis specimen, there is a similar, slender, posteriorly triangular slip of bone present at the anterior margin of the dorsal rim of the left orbit (Fig. 37B), which, according to its position and shape, corresponds closely with the elements in the Eichstätt and the new specimen. In general, the prefrontal seems to be similar in shape to the same element in the therizinosauroid *Erlikosaurus* (*Lautenschlager et al., 2014*), but more slender, resembling the condition in *Sinornithosaurus* (*Xu & Wu, 2001*).

The presence of a separate prefrontal ossification is unexpected in *Archaeopteryx*, as it is absent in the vast majority of pennaraptorans. However, a similar slip of bone above the orbital rim as seen in *Archaeopteryx* is present in *Jinfengopteryx* (*Ji et al., 2005*), and might represent a prefrontal as well, but the fossil is too poorly preserved to confirm this observation with certainty. Likewise, one of the few illustrated specimens of *Anchiornis*, in which this region is sufficiently preserved (*Pei et al., 2017*: fig. 5) shows an elongate triangular element that seems to be separate from the lacrimal above the orbit, and a similar element might be present in an undescribed specimen referred to *Anchiornis* (XHPM 1084; C. Foth, 2014, personal observation). Separate prefrontals have also been described for the dromaeosaurids *Deinonychus* (*Maxwell & Witmer, 1996*) and *Sinornithosaurus* (*Xu & Wu, 2001*). In other pennaraptorans, including oviraptorosaurs (*Clark, Norell & Rowe, 2002*; *Balanoff & Norell, 2012*), most dromaeosaurids (*Barsbold & Osmólska, 1999*; *Norell et al., 2006*; *Novas et al., 2009*; *Pei et al., 2014*), troodontids (*Currie, 1985*; *Xu & Norell, 2004*; *Norell et al., 2009*; *Tsuihiji et al., 2014*), and avialans (*Zhou, Clarke & Zhang, 2008*; *Zhang et al., 2013*), the lacrimal has a pronounced posterior process, resulting in a T-shaped outline of this bone. This process usually tapers posteriorly, thus mimicking the morphology of the prefrontal in *Archaeopteryx* and the few other paravians in which this bone is separate. This similarity and the fact that a separate prefrontal is sporadically present in paravians lends support to

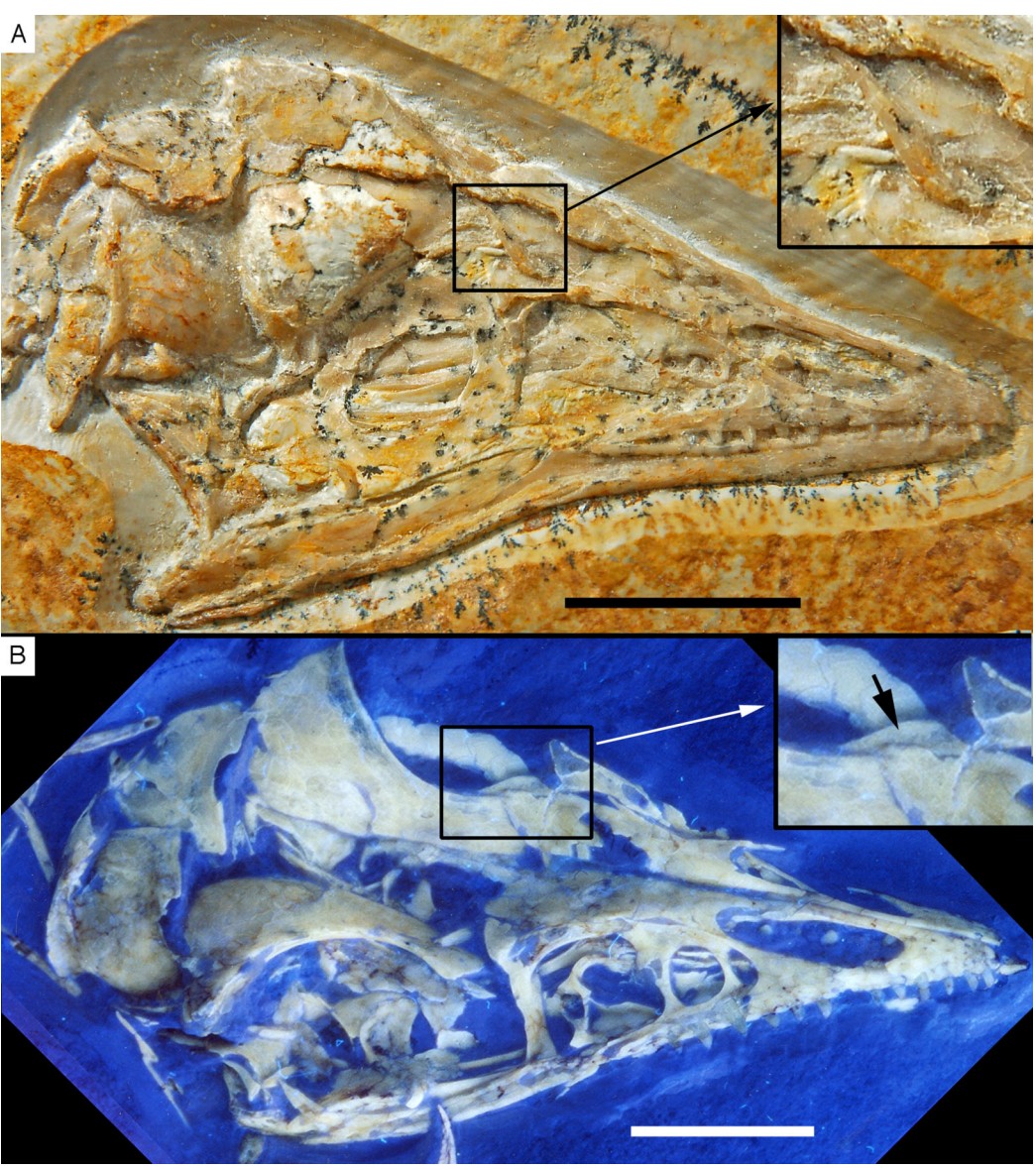

**Figure 37 The prefrontal in *Archaeopteryx*.** (A) Eichstätt specimen, inset shows the displaced prefrontal. (B) Thermopolis specimen; arrow in inset points to preserved prefrontal. Scale bars are 10 mm.

the hypothesis that the posterior process of the lacrimal in other taxa is indeed derived from a fusion of the lacrimal with the prefrontal (*Lautenschlager et al., 2014*). *Currie (1985)* interpreted a slender slip of bone along the lateral margin of the frontal as the prefrontal in *Troodon*, noting that this bone is fused with the frontal and the difference is mainly notable as a change in surface texture (*Currie, 1985*: 1645). However, given the absence of an expanded posterior process in paravians, the shape of the prefrontal in those taxa where it is present, and the fact that this bone does not exclude the frontal from the orbital margin as *Currie (1985)* argued to be the case in *Troodon*, we consider it to be more likely that the prefrontal fuses to the lacrimal. The thin slip of bone parallel to the orbital

margin with different surface texture in the specimen described by *Currie (1985)* might represent a slightly offset margin of the frontal, as it is also present in *Archaeopteryx*, and several other paravians (e.g. *Anchiornis*; *Hu et al., 2009*).

Why the prefrontal and lacrimal fail to coossify in a few taxa of paravians, including *Archaeopteryx*, remains unknown. In those taxa where an enlarged posterior process of the lacrimal is present, no suture is visible; even in a perinate specimen of the troodontid *Byronosaurus* described by *Bever & Norell (2009)*, the two bones are obviously already completely fused, without any visible suture. In oviraptorids that have a pronounced posterior process as adults, a separate prefrontal ossification might be present in early juvenile individuals: Although *Wang et al. (2016)* describe the lacrimal as having a large posterodorsal process in the embryonic oviraptorid skull of IVPP V20182, close inspection of their figure 5a actually indicates a thin line separating an elongate triangular supraorbital ossification from the posterodorsal boss of the lacrimal (also indicated in their drawing of the skull; *Wang et al., 2016*: fig. 5b), so that both the shape of the prefrontal (if it is this bone) as well as its connection with the lacrimal would coincide with the situation seen in therizinosaurids (*Lautenschlager et al., 2014*).

**Presence of coronoid.** As with the prefrontal, a separate coronoid ossification has been said to be absent in *Archaeopteryx* (*Elzanowski & Wellnhofer, 1996*), but the new specimen shows a slip-like coronoid on the internal side of the mandible (Figs. 7 and 12). This bone is absent in modern Aves, but present in most basal paravians, such as dromaeosaurids (*Currie, 1995*; *Norell et al., 2006*), troodontids (*Tsuihiji et al., 2014*), and also hesperornithiforms (*Clarke, 2004*). In non-paravian theropods, the coronoid seems to be absent in a few clades, such as ornithomimosaurs (*Osmólska, Roniewicz & Barsbold, 1972*), therizinosaurs (*Clark, Perle & Norell, 1994*; *Lautenschlager et al., 2014*), and at least some oviraptorosaurs (*Balanoff & Norell, 2012*). If the absence of the coronoid in the Munich specimen of *Archaeopteryx* is not an artifact of preservation (which does not seem to be the case, as no trace of the element can be seen in any of the two mandibular rami), this bone might have been lost or fused to another mandibular element independently of other theropod taxa during the evolution of this genus.

### Postcranial skeleton

**Shape of cervical vertebrae.** The new specimen for the first time provides evidence that at least some of the mid- to posterior cervical vertebrae of *Archaeopteryx* were opisthocoelous (Fig. 17). This contrasts with the usually platycoelous or amphi-platycoelous vertebrae of other basal paravians (*Ostrom, 1969*; *Novas et al., 2009*; *Tsuihiji et al., 2014*), whereas *Patagopteryx* and more derived avialans have heterocoelous cervicals (*Clarke, 2004*). More basal avialans, such as *Confuciusornis* or enantiornithines, show an incipient heterocoely (*Chiappe et al., 1999*). Opisthocoelous cervical vertebrae are usually correlated with large size in basal tetanurans (*Madsen, 1976*; *Britt, 1991*; *Currie & Zhao, 1993a*), but are also found in a few small-bodied coelurosaurs, such as *Compsognathus* (*Ostrom, 1978*; *Peyer, 2006*) and alvarezsaurids (*Chiappe, Norell & Clark, 2002*; *Xu et al., 2013*). Thus, the morphology of the cervical vertebral centra seems to be quite variable in tetanuran theropods. However, whereas convex-concave articulation

facets provide stability in large-bodied taxa (*Fronimos, Wilson & Baumiller, 2016*), their functional significance in small taxa, such as *Archaeopteryx* or derived alvarezsaurids, remains enigmatic. Given the general functional significance of these articulations in other taxa (*Salisbury & Frey, 2000*; *Fronimos, Wilson & Baumiller, 2016*), stabilisation of the cervical vertebral column during rapid head movements in pursuit of elusive prey may be one possible explanation, but further research of this topic is necessary. We should also note that the cervical vertebrae are rather poorly preserved, and the cervical column is incomplete in the new specimen, so further studies of the cervical vertebral column of *Archaeopteryx*, preferably using CT or Synchotron data would be necessary to confirm our findings and elucidate the extent of opisthocoely in the vertebral column of this taxon. If confirmed, the transformation to opisthocoelous cervicals might represent a first step towards heterocoely in more derived avialans, as this condition includes an aspect of opisthocoely (dorsoventrally convex anterior and concave posterior articular surfaces).

**Absence/presence of a sternum.** As in other *Archaeopteryx* specimens, no traces of an ossified sternum are preserved in the new specimen, which is similar to the condition in *Anchiornis*, *Mei*, *Sapeornis* (*Zheng et al., 2014*), *Jianianhualong* (*Xu et al., 2017*), and most non-pennaraptoran theropods (*Weishampel, Dodson & Osmólska, 2004*, see below). This has recently led to the conclusion that sternal elements were completely absent in these species, and that the *m. pectoralis*, responsible for the ventral movement of the forelimbs, was alternatively attached between the proximal end of the humerus and the anterior portion of the gastral basket (*Zheng et al., 2014*; *O'Connor et al., 2015a*). In contrast, most other Pennaraptora had large, ossified sternal plates (*Norell & Makovicky, 1997*; *Clark, Norell & Chiappe, 1999*; *Hwang et al., 2002*; *Burnham, 2004*; *Godfrey & Currie, 2004*; *Xu et al., 2010*; *Pei et al., 2014*), which were fused into a keeled sternum in Ornithothoraces (*Chiappe et al., 1999*; *Zhou, 2002*; *Clarke, Zhou & Zhang, 2006*; *O'Connor et al., 2011*; *Hu et al., 2015*). Based on the distribution of this character within Pennaraptora, *Zheng et al. (2014)* speculated further that the sternal elements of Oviraptorosauria, Dromaeosauridae and Pygostylia were not homologous with each other.

Apart from biomechanical difficulties related to forelimb movements and costal breathing (*Lambertz & Perry, 2015*), several specimens of *Archaeopteryx* (i.e. Munich, Thermopolis, 11th and 12th specimen) show distal cartilaginous expansions usually in four to five pairs of their anterior dorsal ribs (Fig. 21; *Wellnhofer, 2008*, *2009*; *Foth, 2014*). When the Munich and the 11th specimen are compared, these expansions seems to be present between the third (11th specimen) and eighth dorsal rib pair (Munich specimen). Similar expansion can be also found in basal bird *Sapeornis* (*Zhou & Zhang, 2003b*; *Provini, Zhou & Zhang, 2009*). According to *Wellnhofer (2008, 2009)* these expansions were most likely articular facets for sternal ribs, which would indicate the presence of a sternum. In addition, in various specimens of *Archaeopteryx* the anterior end of the gastral basket reaches only until the seventh (see Eichstätt and Thermopolis specimen) to ninth (see 11th specimen) dorsal vertebrae, leaving a large gap between the posterior end of the coracoid and the gastral basket. A similar gap is also present e.g. in *Anchiornis* (BMNHC PH822; *Pei et al., 2017*), *Mei* (*Xu & Norell, 2004*), *Jianianhualong* (*Xu et al., 2017*), *Sapeornis* (*Zhou & Zhang, 2003b*; *Provini, Zhou & Zhang, 2009*) and in various non-

pennaraptoran theropods (*Currie & Chen, 2001*; *Ji et al., 2007*; *Chiappe & Göhlich, 2010*; *Dal Sasso & Maganuco, 2011*; *Rauhut et al., 2012*). The anterior extension of the gastralia itself, however, is similar to Pennaraptora (except of *Archaeopteryx, Anchiornis, Mei, Sapeornis*), but in the latter the gap is absent due the development of ossified sternal plates (*Xu et al., 2010*; *Pei et al., 2014*; *O'Connor et al., 2015a*). As the gastralia do not reach the pectoral girdle anteriorly, this gap might be seen as an indication for the presence of cartilaginous sternal elements in *Archaeopteryx* and other non-pennaraptoran theropods.

Looking at early juvenile specimens of Enantiornithes from the Jehol Group, no plate-like sternum, but several small, isolated, ossified sternal loci are preserved, which were not connected to each other by bone tissue (*Chiappe, Ji & Ji, 2007*; *Zheng et al., 2012*; *O'Connor et al., 2015b*). Assuming a similar developmental mode as in recent birds, these ossified loci were connected with each other by cartilaginous tissue, which ossifies in later ontogenetic stages, forming the plate-like sternum (*Parker, 1867*). As no traces of cartilage can be found in the pectoral region of these juvenile enantiornithine specimens, a general bias in preservation between bone and cartilage seems to be present, which would explain the frequent absence of sternal elements in many non-avialan theropods, *Archaeopteryx* and *Sapeornis* (*Foth, 2014*). A triangular crystalline calcite structure found between the right humerus and the left coracoid, ventral to the glenoid fossa of the right scapula of the Berlin specimen of *Archaeopteryx*, however, may provide direct evidence for the presence of cartilaginous sternal elements, as in the Solnhofen limestones chondral elements are often replaced by crystalline calcite after their decay (*Tischlinger & Unwin, 2004*). Thus, following Wellnhofer's interpretation, *Archaeopteryx* probably possessed cartilaginous sternal elements, which were articulated with four to six sternal ribs. For comparison, in oviraptorosaurs and dromaeosaurids the number of sternal ribs varies between three and four (*Clark, Norell & Chiappe, 1999*; *Hwang et al., 2002*; *Godfrey & Currie, 2004*), while Confuciusornithidae and Ornithothoraces possess at least four (*Chiappe et al., 1999*; *Clarke, Zhou & Zhang, 2006*; *O'Connor et al., 2011*; *Hu et al., 2015*). Although we cannot say anything about the morphology of the sternal elements in *Archaeopteryx*, the supposed number of sternal ribs indicates that the sternal elements were potentially larger than in non-avialan Pennaraptora, similar to the condition in other basal birds. Based on these arguments we conclude that cartilaginous sternal elements were also present in non-pennaraptoran theropods. This is further supported by single findings of bony remains in some theropods, such as in the ceratosaur *Limusaurus* (*Xu et al., 2009b*), the abelisaurid *Carnotaurus* (*Bonaparte, Novas & Coria, 1990*), the allosauroid *Sinraptor* (*Currie & Zhao, 1993a*), the tyrannosaurid *Gorgosaurus* (*Lambe, 1917*), and the alverezsaurid *Mononykus* (*Perle et al., 1994*), which are interpreted as sternal elements. In the latter two species the sternal elements are even fused and keeled, resembling the condition of Ornithothoraces. Given that the sternal elements of *Carnotaurus, Gorgosaurus* and *Sinraptor* are rather small and inconspicuous it is further possible that these elements have been previously overlooked during excavations. The absence of sternal elements in more than 200 specimens of *Anchiornis* studied by *Zheng et al. (2014)* might thus reflect the difficult preservation of cartilage in the sediments of the Tiaojishan Formation: *Wang et al. (2017c)* noted numerous soft

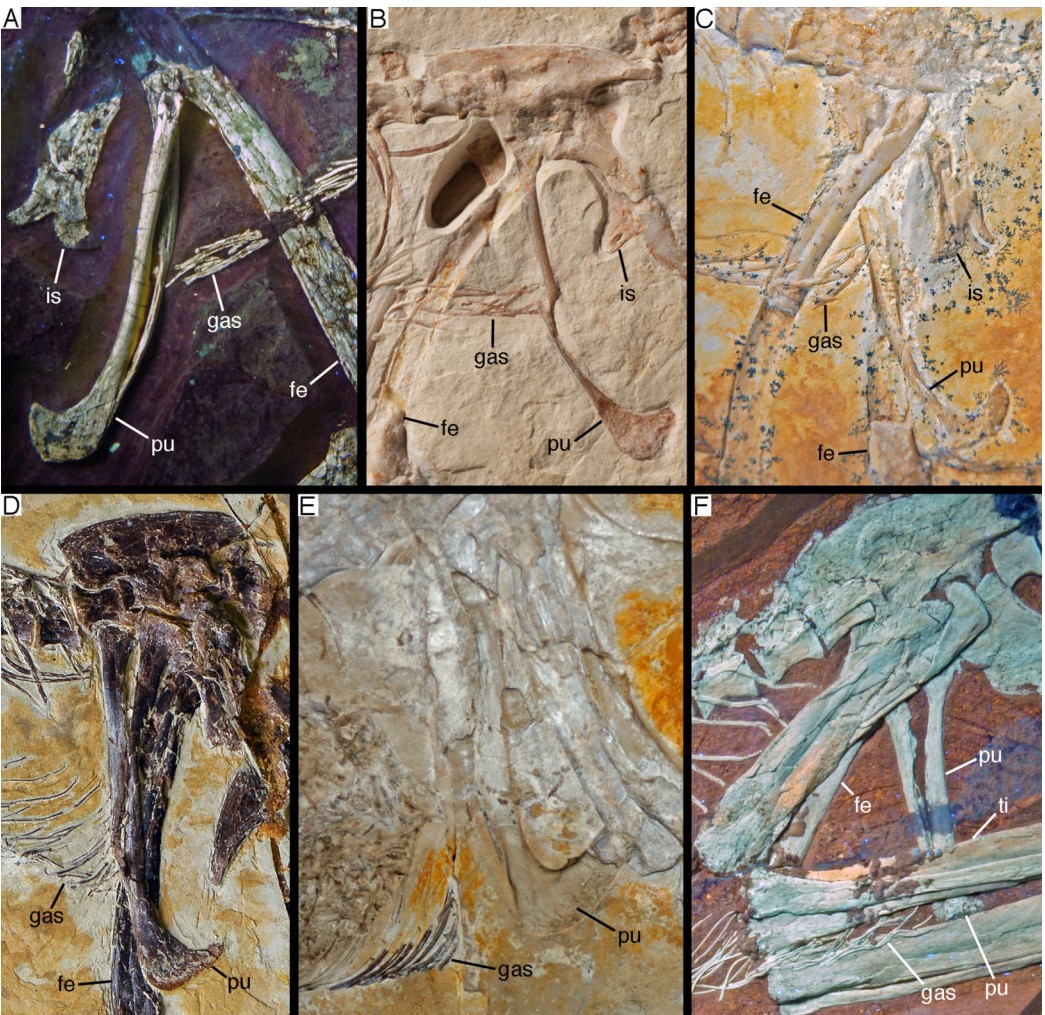

**Figure 38 Position of gastralia in respect to the pubis in several theropods.** (A) Twelfth specimen of *Archaeopteryx*, photograph under UV light. (B) Munich specimen of *Archaeopteryx*. (C) Eichstätt specimen of *Archaeopteryx*. (D) *Anchiornis* (STM0-52; photo courtesy of Wang Xiaoli). (E) *Compsognathus* (MNHN CNJ 79). (F) *Sciurumimus* (BMMS BK 11; photograph under UV light).

tissues that showed up in specimens of *Anchiornis* under laser-stimulated fluorescence (LSF). Interestingly, however, the presence of a cartilagenous pad at the end of the pubis was indicated by an area that did not fluoresce in between the distal end of the pubis and the integument (*Wang et al., 2017c*: fig. 4b), indicating that cartilage might not be visible in these specimens under special lighting methods, such as LSF or UV. Other lighting techniques might be necessary to visualise cartilage in these specimens, or cartilage might simply note be preserved at all.

**Position of gastralia in respect to pubis.** In all specimens of *Archaeopteryx*, in which both the gastral basket and the pubis are adequately preserved, the posterior end of the gastral basket is placed at about half-length of the pubis (Figs. 38A–38C), not at its distal end, as might be expected. This unusual preservation can be seen in the Berlin, Eichstätt, Solnhofen, Munich, 11th, and the new specimen (*Wellnhofer, 2008, 2009; Foth, Tischlinger &*

*Rauhut, 2014*); in the Thermopolis specimen, the gastralia do not reach the pubis, but their posterior end is placed approximately at the same relative level as in the other specimens (*Mayr et al., 2007*). As noted above, this elevated position of the gastralia in respect to the pubis is also seen in the troodontids *Jinfengopteryx* (*Ji et al., 2005*) and *Jianianhualong* (*Xu et al., 2017*), the avialans *Anchiornis* (Fig. 38D; *Zheng et al., 2014*; *Pei et al., 2017*), *Eosinopteryx* (*Godefroit et al., 2013a*), *Sapeornis* (*Zhou & Zhang, 2003b*), and *Sinornis* (*Sereno, Rao & Li, 2002*), and various dromaeosaurids (*Hone et al., 2010*; *Xu et al., 2010*; *O'Connor, Zhou & Xu, 2011*; *Han et al., 2014*; *Lü & Brusatte, 2015*); all of these taxa have opisthopubic, retroverted pubes. In contrast, the preservation is different in theropods with a pro/mesopubic anatomy, including megalosauroids (Fig. 38F; *Rauhut et al., 2012*), tyrannosauroids (*Lambe, 1917*; *Xu et al., 2012*), *Scipionyx* (*Dal Sasso & Maganuco, 2011*), compsognathids (Fig. 38E; e.g. *Currie & Chen, 2001*; *Hwang et al., 2004*; *Peyer, 2006*; *Ji et al., 2007*; *Hu, Wang & Huang, 2016*), ornithomimosaurs (*Osborn, 1916*; *de Klerk et al., 2000*; *Varricchio et al., 2008*), oviraptorosaurs (*Zhou et al., 2000*; *Balanoff & Norell, 2012*), and the troodontid *Gobivenator* (*Tsuihiji et al., 2014*), where the gastral basket reaches the distal pubic boot. This probably represents the natural association between these structures, as the gastralia are connected with the sternum and the pubis by midventral ligaments and are embedded in the *M. rectus abdominis* in recent reptiles (*Carrier & Farmer, 2000*; *Claessens, 2004*; *Fechner & Schwarz-Wings, 2013*). The arrangement found in *Archaeopteryx* and other paravians thus most likely represents a post-mortem displacement, which seems to be related to the opisthopubic anatomy, as evidenced from the lack of this displacement in the mesopubic *Gobivenator* (*Tsuihiji et al., 2014*). Although the change in pelvic anatomy probably led to a reconfiguration of pelvic and thoracal musculature (*Carrier & Farmer, 2000*), a post-mortem displacement of the gastral basket after death due to tension is unlikely, as muscle tones are triggered by nerve impulses, occurring only pre- and perimortem (*Leisman & Koch, 2009*; *Reisdorf & Wuttke, 2012*). However, it is possible that the retroversion of pubis in the taxa in question led to an increase or ventrally directed relocation of the abdominal air sac system, so that the reposition of the gastralia may result from a post-mortem collapse of the air sacs, causing a depression in the abdominal region. This might thus indicate that the opisthopubic morphology in paravians was related to a change in function and probably an increase of these abdominal air sacs.

## Variation in *Archaeopteryx*
### Skeletal proportions
Previous studies proposed that all differences found in the skeletal proportions of *Archaeopteryx* represent allometric variation within a growth series (*Houck, Gauthier & Strauss, 1990*; *Senter & Robins, 2003*). However, the sample size of these two studies was only five specimens. Since 2003 five more specimens were discovered or described. Using a single value *t*-test, we compared the 12th specimen against a sample of eight other *Archaeopteryx* specimens (London, Berlin, Maxberg, Eichstätt, Solnhofen, Munich, Thermopolis, and 11th specimen). Because the Eichstätt specimen is a juvenile individual (*Wellnhofer, 1974*; *Erickson et al., 2009*), we excluded this specimen in a second analysis, to

**Table 3 Comparison of skeletal proportions of specimens of *Archaeopteryx*.**

| Specimen | Skull | Scapula | Humerus | Ulna | Metacarpal I | Tibia | Metatarsal II | Metatarsal III | Metatarsal IV | PD II-2 | PD III-2 | PD III-3 | PD IV-1 | Digit IV-2 |
|---|---|---|---|---|---|---|---|---|---|---|---|---|---|---|
| Eichstätt | 1.05 | 0.68 | 1.12 | 0.99 | 0.14 | 1.43 | 0.76 | 0.82 | 0.74 | 0.19 | 0.22 | 0.19 | 0.16 | 0.14 |
| Maxberg | ? | 0.78 | 1.24 | 1.07 | 0.17 | 1.37 | 0.66 | 0.72 | 0.67 | 0.17 | 0.18 | 0.16 | 0.15 | 0.14 |
| Thermopolis | 1.05 | 0.70 | 1.13 | 1.01 | 0.13 | 1.48 | 0.70 | 0.79 | 0.72 | 0.17 | 0.19 | 0.16 | 0.15 | 0.13 |
| Munich | 0.97 | 0.69 | 1.18 | 1.14 | 0.15 | 1.54 | 0.75 | 0.87 | 0.80 | 0.18 | 0.18 | 0.18 | 0.17 | 0.13 |
| Berlin | 1.00 | 0.80 | 1.21 | 1.08 | 0.13 | 1.36 | 0.67 | 0.71 | 0.62 | 0.13 | 0.17 | 0.16 | 0.13 | 0.12 |
| London | ? | 0.75 | 1.23 | 1.10 | ? | 1.34 | 0.66 | 0.72 | ? | 0.18 | 0.18 | 0.16 | ? | ? |
| Solnhofen | 0.97 | 0.76 | 1.24 | 1.10 | ? | 1.37 | 0.67 | 0.71 | 0.67 | 0.19 | 0.18 | 0.16 | 0.15 | 0.13 |
| 11th | ? | 0.80 | 1.19 | 1.13 | 0.18 | 1.38 | 0.68 | 0.74 | 0.69 | 0.17 | 0.18 | 0.16 | ? | ? |
| 12th | 1.06 | 0.81 | 1.15 | 1.04 | 0.13 | 1.25 | 0.60 | 0.64 | 0.62 | 0.16 | 0.17 | 0.15 | 0.14 | 0.12 |
| *t* Value | 2.56 | 3.66 | 2.53 | 2.06 | 2.80 | 6.82 | 6.46 | 5.62 | 3.61 | 2.80 | 2.38 | 2.99 | 2.68 | 3.94 |
| *p* Value | >0.05 | <0.01 | <0.05 | >0.05 | <0.05 | <0.01 | <0.01 | <0.01 | <0.02 | <0.05 | <0.05 | <0.05 | <0.05 | <0.02 |
| *t* Value* | 3.08 | 3.23 | 3.46 | 3.22 | 2.59 | 5.84 | 6.80 | 4.92 | 2.94 | 2.26 | 3.25 | 2.98 | 2.03 | 3.17 |
| *p* Value* | <0.05 | <0.02 | <0.02 | <0.02 | >0.05 | <0.01 | <0.01 | <0.01 | <0.05 | >0.05 | <0.02 | <0.05 | >0.05 | <0.05 |

**Notes:**
All measurements have been divided by femur length as proxy for body size to account for differences in absolute size. *t* Values refer to values of single-value *t* tests against the entire population; *t* values with * refer to values if the juvenile Eichstätt specimen is excluded. Bold numbers indicate values for which the difference is significant at $p < 0.05$.

minimise the effect of potential ontogenetic variation. All data are based on *Mayr et al. (2007)*, *Wellnhofer (2008, 2009)* and own measurements. The data were divided through femur length to minimise the impact of size.

According to the *t*-test, the new specimen shows some significant differences compared to the other specimens, including the presence of a relatively long scapula, a relatively short humerus, ulna (only when the Eichstätt specimen is excluded from the sample), metacarpal I, tibia and metatarsus (Table 3). In addition, some pedal digits were found to be relatively shorter, too. Thus, the test indicates that the new specimen has relatively short legs and arms when compared to size. In addition, when the Eichstätt specimen is excluded the skull of the 12th specimen is significantly longer than in the other *Archaeopteryx*.

Due to lack of data, it cannot currently be evaluated if the differences found are within the range of intraspecific variation (when compared to other non-avialan theropods and recent birds) or if they might justify the establishment of a new species. The latter option might be supported by a few osteological differences, such as the different shape and position of the promaxillary foramen and the short and blunt jugal process of the palatine (if compared to the much longer and more slender process in the Thermopolis specimen; *Mayr et al., 2007*), but little is known about the variability of such characters in paravian theropods. Furthermore, given its slightly older age and different provenance it also cannot be ruled out that the differences seen in the 12th specimen represent an anagenetic or biogeographic variation of the genus *Archaeopteryx* (see discussion below). Moreover, many of the specimens of *Archaeopteryx* deviate from the average seen in this genus in one or more proportions. A relative long scapula, for example, can also be found in the Berlin and 11th specimens, while a shortened humerus is present in the Eichstätt (subadult) and

Thermopolis specimens. Further research into variability in both non-avialan and avialan theropods is necessary to better evaluate the significance of this variation.

### Variation in the dentition

As discussed above, *Archaeopteryx* shows a very characteristic tooth shape in the premaxillary, often the anterior maxillary and anterior dentary teeth, which can be used to diagnose the genus. However, when the entire dentition is compared, the individual specimens show a lot of variation, as previously recognised by *Howgate (1984a)*. Variation is seen in the number of teeth, the spacing of teeth in different regions, the shape of teeth in different positions within the dentition, and the inclination of individual teeth.

While the number of premaxillary teeth is constant at four in all specimens where this number can be established (Berlin, Eichstätt, Solnhofen, Thermopolis, 11th, and, probably, 12th specimen), the number of maxillary teeth varies between eight (Berlin and Eichstätt specimens) and nine (Daiting, Thermopolis and 12th specimens), and the number of dentary teeth between 11 (Eichstätt), 12 (Munich) and 13 (12th specimen). However, as the total number of teeth in the maxilla and the dentary are only determinable in a few specimens, the significance of these differences in tooth count is unclear. Individual variation in tooth number of one or more positions in the maxilla and dentary has been noted in several non-avialan theropods (*Madsen, 1976*; *Colbert, 1989*; *Currie, 2003*), whereas the number of premaxillary teeth generally seems to be more stable. Furthermore, different tooth counts might also be due to ontogenetic variation (see *Rauhut & Fechner, 2005*; *Kundrát et al., 2008*; *Bever & Norell, 2009*).

The spacing of individual teeth shows a high amount of variation between the different specimens of *Archaeopteryx*. As noted above, both the premaxillary and dentary dentitions are offset from the anterior tip of their respective bones in the 12th specimen. A similarly large offset in the premaxilla is present in the Solnhofen (*Wellnhofer, 1992*) and, probably, the London specimens (*Howgate, 1984b*), whereas it is present, but notably smaller in the Berlin, Eichstätt, Thermopolis, and 11th specimens (*Dames, 1884*; *Wellnhofer, 1974*, *1988b*, *1992*, *2008*, *2009*; *Howgate, 1984a*; *Mayr et al., 2007*; *Foth, Tischlinger & Rauhut, 2014*). An offset in the dentition of the dentaries is also found in the Eichstätt and Munich specimens. This small space anterior to the dentition might indicate the presence of an incipient rhamphotheca, although other indications for this (such as increased vascularity) cannot be established. In the premaxilla, tooth spacing in the Berlin, 11th and, probably, the 12th specimen differs from other specimens. In the Eichstätt, Solnhofen and Thermopolis specimens, the first two premaxillary teeth are closely spaced (half a tooth width or less in between the teeth), whereas there is a gap of at least one tooth width between the second and third and the third and fourth premaxillary tooth (Fig. 39B; *Wellnhofer, 1974*, *1988b*, *2008*, *2009*; *Howgate, 1984a*; *Mayr et al., 2007*). In contrast, the Berlin specimen has premaxillary teeth one to three closely spaced, followed by a gap between the third and fourth tooth (Fig. 39A; *Howgate, 1984a*; *Wellnhofer, 2008*, *2009*), and in the 11th specimen, all premaxillary teeth are equally and widely spaced, with approximately one tooth width distance between individual elements (Fig. 40; *Foth, Tischlinger & Rauhut, 2014*). If our interpretation that there is a further

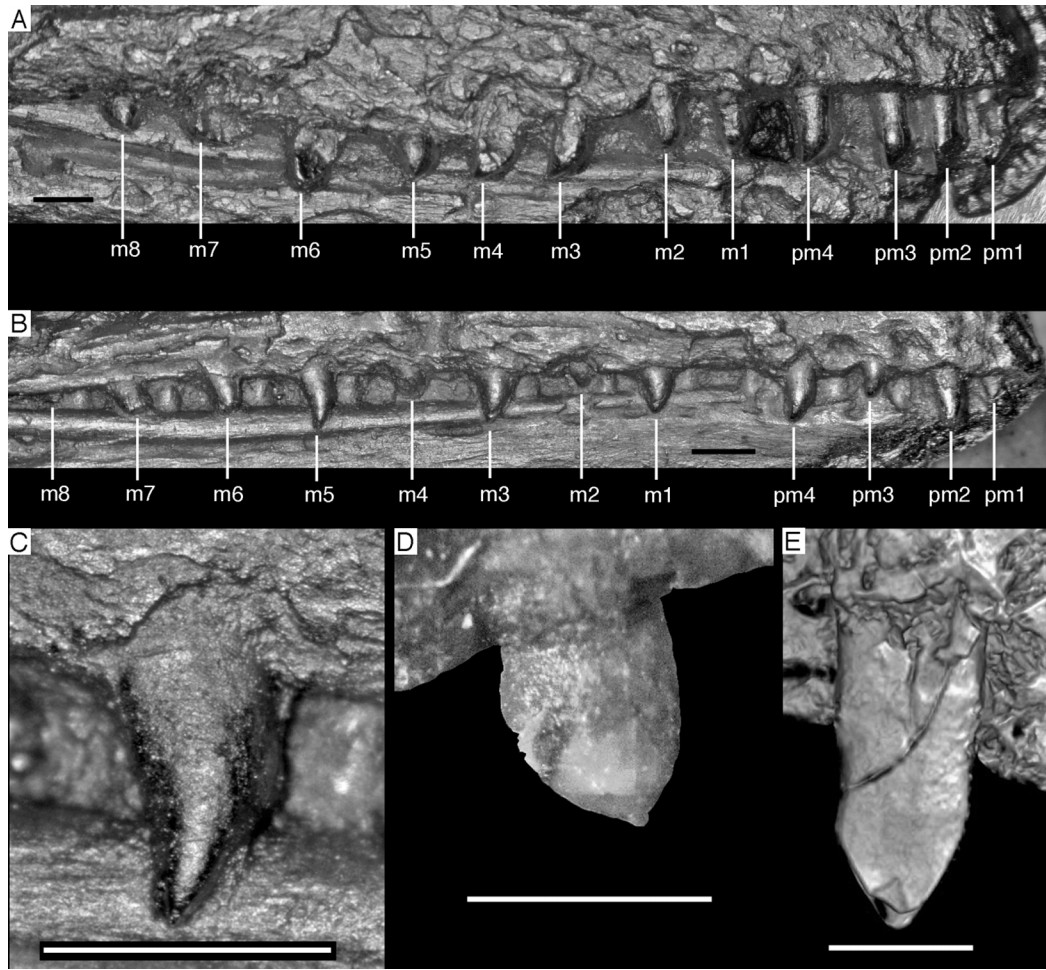

**Figure 39** **Dentition of *Archaeopteryx*.** (A) Premaxillary and maxillary dentition of the Berlin specimen (based on a high-resolution cast held at the BSPG). (B) Premaxillary and maxillary dentition of the Eichstätt specimen (based on a high-resolution cast held at the BSPG). (C) Fifth maxillary tooth of the Eichstätt specimen (based on a high-resolution cast held at the BSPG). (D) Sixth maxillary tooth of the Solnhofen specimen. (E) Sixth maxillary tooth of the Daiting specimen in lingual view (based on synchotron data; courtesy Paul Tafforeau). m, maxillary tooth positions; pm, premaxillary teeth or tooth positions. Scale bars are 1 mm.

premaxillary tooth missing between the second and third preserved tooth in the new specimen (see above), this specimen would again differ from other specimens in that all teeth are equally, but closely spaced, with considerably less than one tooth width in between individual teeth (Fig. 14).

Variation in the spacing of individual teeth is also present in the maxillary and dentary dentitions. One extreme condition is seen in the Eichstätt specimen, in which the teeth are approximately evenly spaced, but individual teeth are considerably more than one tooth width apart (Fig. 39B; *Wellnhofer, 1974*, *2008*, *2009*; *Howgate, 1984a*). Furthermore, a diastema of more than two tooth widths is present between the premaxillary and maxillary teeth in the anterior end of the maxilla (Fig. 39B; *Wellnhofer, 1974*, *1988b*, *1992*, *2008*, *2009*; *Howgate, 1984a*). In the Berlin and Thermopolis specimens, the maxillary

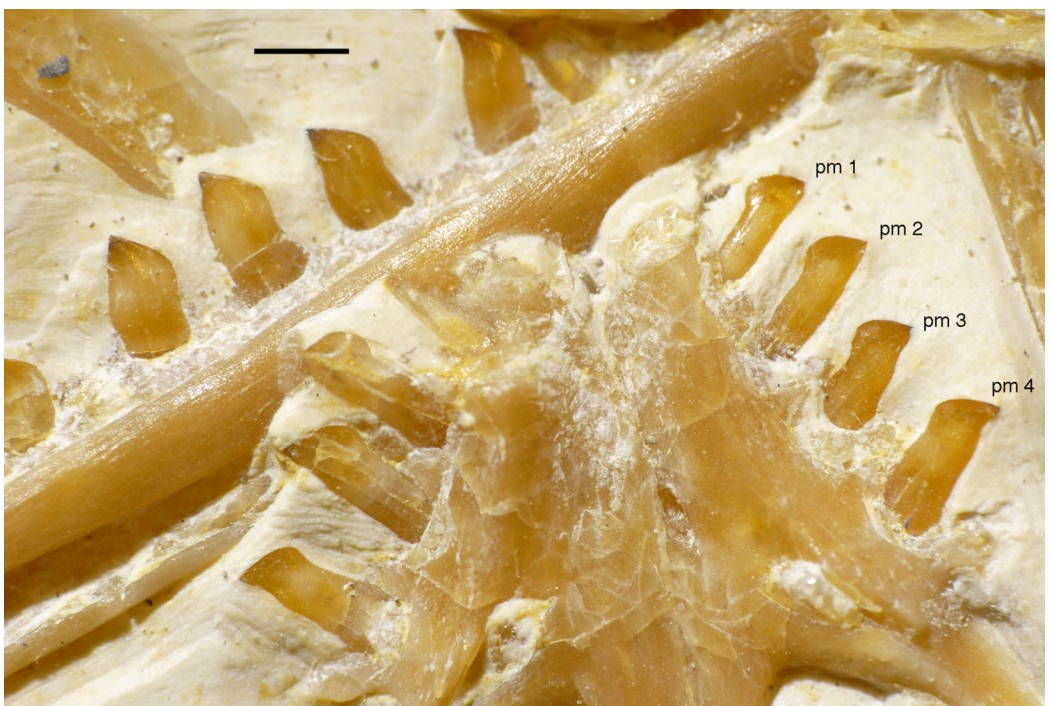

**Figure 40 Premaxillary and mid-dentary teeth of the 11th specimen of *Archaeopteryx.*** pm, premaxillary teeth. Exact position of the dentary teeth cannot be established, as anterior end of the dentary is not exposed. Scale bar is 1 mm. 

teeth are also rather widely spaced, with the distance between individual teeth being approximately one tooth width in the anterior part of the maxilla, but increasing in more posterior teeth (Fig. 39A; *Howgate, 1984a*; *Mayr et al., 2007*; *Wellnhofer, 2008*, *2009*) to a maximum of almost two tooth widths between the seventh and the eighth tooth in the Thermopolis specimen (*Mayr et al., 2007*). The Solnhofen specimen also shows a diastema between the premaxillary and maxillary teeth of more than two tooth widths, but in contrast to the Eichstätt specimen, here the toothless portion is mainly formed by the posterior end of the premaxilla (*Wellnhofer, 1988b*, *1992*, *2008*, *2009*). The first maxillary tooth is offset from both the anterior end of the maxilla and the second maxillary tooth by approximately one tooth width, but maxillary teeth two to four seem to be closely spaced. From the space between the fourth and fifth maxillary tooth onwards, the distance between individual teeth is again more than one tooth width. A similar pattern seems to be present in the Daiting specimen, in which maxillary teeth two to four are closely spaced, followed by more widely spaced posterior teeth (Fig. 41; *Tischlinger, 2009*). However, in this specimen, the premaxillary body and anterior end of the maxilla are missing, and the first maxillary tooth is only represented by its broken root. The 12th specimen described here also has a gap of approximately one tooth width between the first and second maxillary tooth, but here, maxillary teeth two to six are closely spaced, and only from the gap between maxillary tooth six and seven onwards, the spacing becomes wider again (Fig. 14). Furthermore, there is no marked diastema between the premaxillary and maxillary dentition; the distance between the last premaxillary and first maxillary

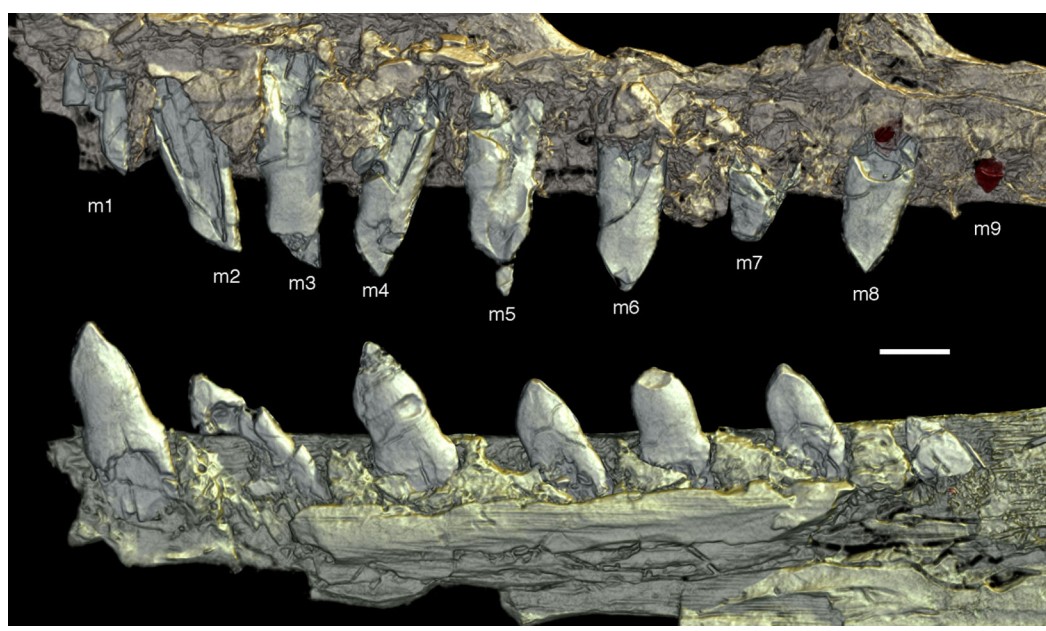

**Figure 41 Dentition of the Daiting specimen of *Archaeopteryx*.** Preserved maxillary (top) and dentary tooth row (bottom) in lingual view, not in their original relative positions. m, maxillary tooth positions. Tooth positions in the maxillary are based on the assumption of nine tooth positions; exact tooth positions in the dentary cannot be determined, as the element is incomplete anteriorly. Based on synchotron data; reconstruction by Paul Tafforeau, used with permission. Scale bar is 1 mm.

tooth is approximately one tooth width and thus considerably less than in the Eichstätt and Solnhofen specimens. In the London specimen, finally, the preserved maxillary teeth are even up to two tooth widths apart (*Wellnhofer, 2008*: fig. 5.27). *Howgate (1984b)* argued that there were empty tooth positions between the preserved teeth in this specimen, mainly based on a UV image published by *de Beer (1954*: pl. 9, fig. 4*)*. If this was the case, the spacing of maxillary teeth would be considerably less than one tooth width; however, close inspection of a high quality cast of the jaw remains of the London specimen kept at the BSPG did not reveal any evidence for the presence of additional alveoli in between the preserved teeth. Detailed observations on the actual specimen would be necessary to clarify this point.

As the Berlin, Eichstätt and Solnhofen specimens are preserved with the lower jaw in occlusion, the spacing of their dentary teeth is difficult to establish. However, the visible dentary teeth in both the Eichstätt and Solnhofen specimens are widely spaced, with the distance between teeth being apparently more than the width of the individual teeth (*Wellnhofer, 1974*, *1988b*, *1992*, *2008*, *2009*). In the Daiting specimen, only the posterior seven dentary teeth (or their roots) are preserved, but these teeth are widely spaced, at approximately one tooth width distance (Fig. 41). In contrast, the anterior dentary teeth in the Munich (SNSB-BSPG 1999 I 50) and 12th specimen are less than one tooth width apart, and the spacing of the teeth increases from the seventh or eighth dentary tooth onwards (Fig. 42). In the 11th specimen, the dentary teeth seem to be less than one tooth width apart throughout the series (Fig. 15).

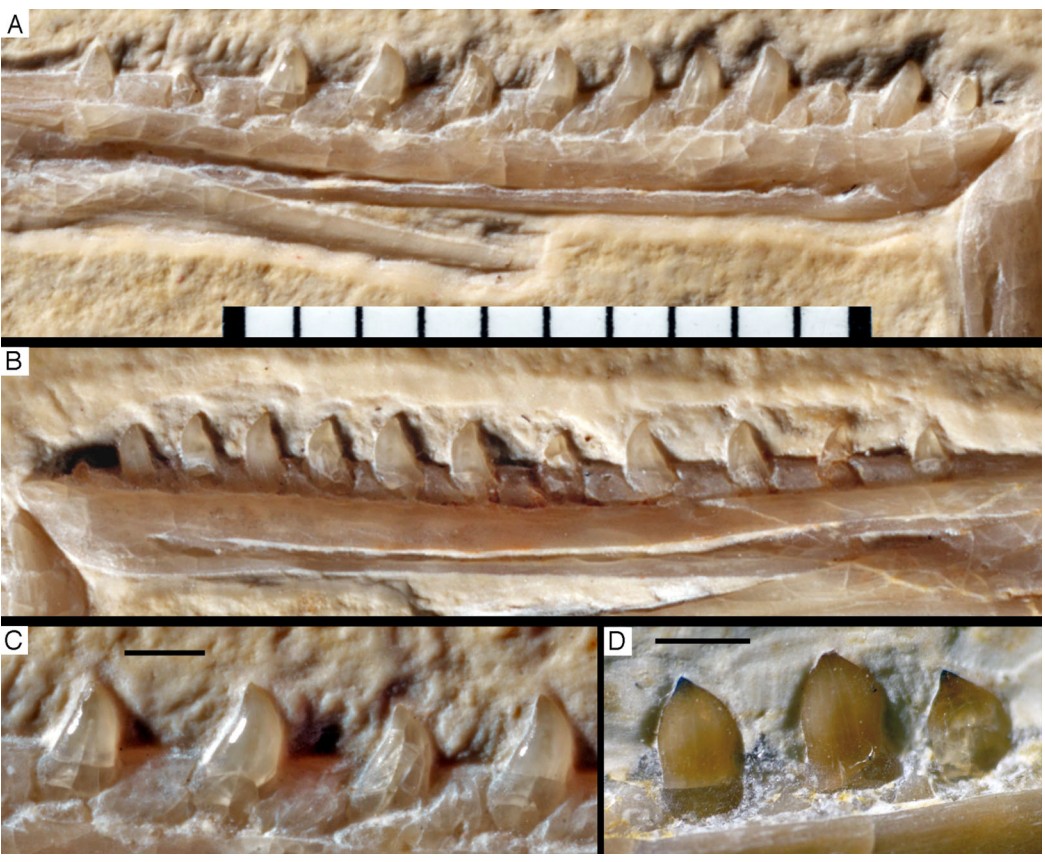

**Figure 42 Dentary dentition of *Archaeopteryx*.** A, B, Left (A) and right (B) dentary of the Munich specimen in lingual view. (C) Posterior mid-dentary teeth of the Munich specimen in lingual view. (D) Posterior mid-dentary teeth of the 11th specimen in lingual view. Scale bar for A and B is 10 mm and 1 mm for C and D.               

Further variation affects the inclination of the teeth, which not only varies between the specimens, but sometimes also between different regions of the dentition within one specimen. While the teeth are usually perpendicular with respect to the long axis of the jaw bone, the premaxillary teeth of the Solnhofen and 11th specimen (Fig. 40), the maxillary teeth of the Thermopolis and 12th specimen (Fig. 14) and the dentary teeth of the Daiting (Fig. 41) and Munich specimen (Fig. 42) are notably procumbent. In the Eichstätt specimen, the last premaxillary tooth and the anterior maxillary teeth are perpendicular, but the anterior premaxillary teeth and posterior maxillary teeth are notably procumbent. Similarly, in the Thermopolis specimen, the first two premaxillary teeth are slightly procumbent, whereas the third and fourth premaxillary tooth are perpendicular (*Mayr et al., 2007*). Although teeth angulation might be influenced by taphonomic factors, as teeth may become disattached from their respective jaw bone and thus change orientation somewhat, it seems unlikely that this can account for all of this variation. Indeed, differences in angulation of the teeth that are probably preservational artefacts are present in a number of specimens, such as the Daiting specimen, where one maxillary tooth seems to be posteriorly inclined, whereas another is apparently procumbent, and yet another tooth is perpendicular (*Tischlinger, 2009*). However, the very regular anterior inclination of the teeth in the specimens

(e.g. in the premaxillary of the 11th specimen and the dentary of the Munich specimen; Figs. 40, 42A and 42B) noted above indicate that this might really be a primary morphology.

Finally, some variation occurs in the tooth shape itself, both within the individual tooth rows, as well as between specimens, as already noted, based on the much smaller sample size then available, by *Howgate (1984a)*. Although premaxillary teeth of all specimens in which the premaxilla is preserved show the common characteristics outlined above, there is some variation in the shape of premaxillary teeth. The anterior two premaxillary teeth of the Berlin, Solnhofen, and 11th specimen have a strongly convex mesial and straight (11th specimen) to notably convex (Solnhofen specimen) distal margin, so that the tooth tip is placed slightly mesial to or above the level of the distal margin of the crown base (see *Wellnhofer, 1988b*, fig. 4; *2008*: fig. 5.50; *Mayr et al., 2007*: fig. 5c). The posterior two premaxillary teeth in these specimens have a slightly concave distal margin, but with the tip still being placed above the distal margin of the crown base. All of these teeth show a notable mesial bulge basally above the root, and a small distal bulge is present in the third and fourth premaxillary tooth. The premaxillary tooth preserved with the London specimen shows a slightly concave distal margin (*Wellnhofer, 2008*: fig. 5.27), indicating that this tooth belongs to premaxillary tooth position three or four, whereas the complete second preserved premaxillary tooth of the 12th specimen has a straight distal margin, supporting its identification as second premaxillary tooth. In contrast to all of these specimens, the more or less complete second to fourth premaxillary teeth of the Eichstätt specimen (the first tooth is missing the tip) are recurved, with only a very slight mesial bulge and no distal bulge, and a notably concave distal margin. Finally, in the Thermopolis specimen, the premaxillary teeth follow the pattern seen in the Berlin, Solnhofen and 11th specimen in having a convex mesial and straight (first two teeth) to concave (premaxillary teeth three and four) distal margin, but they have a reduced mesial bulge, as seen in the Eichstätt specimen.

The lateral teeth of *Archaeopteryx* change morphology in the course of the tooth row, usually from more slender, recurved mesial teeth to more robust, less curved distal teeth, but patterns and morphologies seen differ between different specimens. One extreme is the Eichstätt specimen, in which all maxillary teeth are slender and recurved, being only slightly more robust, but otherwise very similar to the premaxillary teeth in this specimen (*Wellnhofer, 1974*, *2008*, *2009*; *Howgate, 1984a*). A very similar situation is found in the Thermopolis specimen, in which the maxillary teeth are also recurved and pointed (*Mayr et al., 2007*). In the Berlin specimen, the anterior maxillary teeth are similar to the distal premaxillary teeth in having slight mesial and distal bulges above the root and a slightly concave distal margin (*Wellnhofer, 2008*: fig. 5.50). From the sixth maxillary tooth onwards, the teeth are more robust and have convex mesial and distal margins. In the Solnhofen, Daiting and 12th specimen, the anteriormost maxillary teeth have a convex mesial and straight distal margin, but from the second (12th specimen) to fourth (Daiting specimen) tooth position on, the teeth are bulbous, with strongly convex mesial and distal margins, and the tip of the crown becomes more centrally placed in distal teeth. This also seems to be the situation in the London specimen, as far as can be judged from the preserved maxillary fragment (*Wellnhofer, 2008*: fig. 5.27).
Similar differences are also found in the dentary teeth in those specimens where these teeth are exposed. In the Munich specimen, the dentary teeth are similar to the maxillary teeth in the Eichstätt and Thermopolis specimens in being slightly recurved throughout the series, with notably convex mesial and straight to slightly concave distal margins (*Wellnhofer, 1993*, *2008*, *2009*; *Elzanowski & Wellnhofer, 1996*). Such recurved teeth are only present in the anteriormost tooth positions in the dentary of the 12th specimen; more distal teeth are similar to the maxillary teeth in being bulbous and having convex mesial and distal margins. The latter morphology is also found in the middle and distal dentary teeth of the Daiting and 11th specimens.

In summary, the different specimens of *Archaeopteryx* show variation in several different aspects of tooth morphology, number, spacing, and inclination, and no two specimens show the exact same pattern in all of these aspects (Table 4). *Howgate (1984a)* proposed four possible explanations for differences in the dentition of the London, Berlin and Eichstätt specimen, including sexual dimorphism, ontogenetic variation, polymorphism (all different types of intraspecific variation) and interspecific variation. Indeed, based on the unusual, recurved teeth of the Eichstätt specimen, *Howgate (1984a)* proposed a new species, *Archaeopteryx recurva*, for this specimen, which he even referred to a separate genus, *Jurapteryx*, based on additional proposed osteological differences, in a later publication (*Howgate, 1985*). Unfortunately, with the increase of number of specimens the pattern of variation remains confusing, so that none of *Howgate's (1984a)* hypotheses can currently be favoured. Due to the unsolved species taxonomy of *Archaeopteryx* (see above), both intra- and interspecific variation are possible, although some types of intraspecific variation are maybe more relevant than others. As, for instance, the sex of the different *Archaeopteryx* specimens is unknown, sexual dimorphism cannot be tested. Furthermore, although dental polymorphism is widespread throughout tetrapods, most studies are focused on urodeles or mammals (*Pedersen, 1991*; *Hanken, Wake & Freeman, 1999*; *Szuma, 2002*, *2011*), and the phenomenon of dental polymorphism is not well understood in reptiles in general and possibilities to study this phenomenon in dinosaurs are restricted only to some key taxa with high number of specimens. Current information on dental variability mainly concerns variability in the number of teeth between different individuals (*Madsen, 1976*; *Raath, 1977*; *Colbert, 1989*, *1990*) or the variabilty of tooth shape within a single dentition (*Smith, 2005*, *2007*; *Hendrickx, Mateus & Araújo, 2015b*). Furthermore, various studies documented ontogenetic variation in both number and morphology of teeth within theropod dinosaurs (*Colbert, 1989*; *Carr, 1999*; *Rauhut & Fechner, 2005*; *Samman et al., 2005*; *Kundrát et al., 2008*; *Bever & Norell, 2009*; *Buckley et al., 2010*; *Tsuihiji et al., 2011*; *Rauhut et al., 2012*), which seems to be most extremely developed in the basal ceratosaur *Limusaurus* (*Wang et al., 2017b*). These changes probably correlate with ontogenetic shifts in the diet spectrum (*Farlow, 1976*; *Rauhut et al., 2012*) as documented in extant crocodylians (*Cott, 1961*; *Hutton, 1987*; *Webb, Hollis & Manolis, 1991*; *Da Silveira & Magnusson, 1999*; *Horna, Cintra & Ruesta, 2001*).

When compared with body size, some differences in the dentition of *Archaeopteryx* could be explained as ontogenetic differences, including the variation in the tooth shape of

**Table 4 Comparison of dentition of different specimens of *Archaeopteryx*.**

**Premaxillary teeth**

| Specimen | Number | Offset from tip | Tooth spacing | Tooth inclination | Tooth morphology |
|---|---|---|---|---|---|
| London | 4? | Large | ? | Perpendicular? | Mesially convex with large bulge, distally concave (pm3 or pm4) |
| Berlin | 4 | Small | First three teeth closely spaced | All teeth perpendicular | All teeth mesially convex with large bulge; pm1 and pm2 distally straight, pm3 and pm4 distally concave |
| Eichstätt | 4 | Small | First two teeth closely spaced | First three procumbent, fourth perpendicular | All teeth mesially convex with reduced bulge; all teeth recurved, distally concave with no distal bulge |
| Solnhofen | 4 | Large | First two teeth closely spaced | All teeth procumbent | All teeth mesially convex with large bulge; pm1 and pm2 distally straight, pm3 and pm4 distally convex |
| Thermopolis | 4 | Small | First two teeth closely spaced | First two procumbent, three and four perpendicular | All teeth mesially convex with reduced bulge; pm1 and pm2 distally straight, pm3 and pm4 distally concave |
| 11th | 4 | Small | All teeth widely spaced | All teeth procumbent | All teeth mesially convex with large bulge; pm1 and pm2 distally straight, pm3 and pm4 distally concave |
| 12th | 4(?) | Large | All teeth closely spaced (?) | ? | Mesially convex with large bulge, distally convex (pm2) |

**Maxillary teeth**

| Specimen | Number | pm-m diastema | Tooth spacing | Tooth inclination | Tooth morphology |
|---|---|---|---|---|---|
| London | ? | ? | Widely spaced(?) | Slightly procumbent | Preserved crown resembles anterior pm teeth; last preserved tooth robust |
| Berlin | 8 | Absent | Widely and ±evenly spaced | Anterior teeth perpendicular | m1–m5 like pm3–pm4, from m6 onwards robust and mesially and distally convex |
| Eichstätt | 8 | Present (maxilla) | Widely and evenly spaced | Anterior teeth perpendicular, posterior teeth procumbent | All teeth slender and recurved |
| Solnhofen | ? | Present (premax) | m2–m4 closely spaced, more posterior teeth widely spaced | Largely perpendicular | Anterior teeth mesially convex, distally straight, posterior teeth bulbous |
| Daiting | 9 | ? | m2–m4 closely spaced, more posterior teeth widely spaced | Largely perpendicular | Anterior teeth mesially convex, distally straight, posterior teeth bulbous |
| Thermopolis | 9 | Absent | Widely spaced with increase posteriorly | All teeth procumbent | All teeth slender and recurved |
| 12th | 9 | Absent | m2–m6 closely spaced, more posterior teeth widely spaced | All teeth procumbent | Anterior teeth mesially convex, distally straight, posterior teeth bulbous |

(Continued)

**Dentary teeth**

| Specimen | Number | Offset from tip | Tooth spacing | Tooth inclination | Tooth morphology |
|---|---|---|---|---|---|
| Eichstätt | 11 | Present | Widely spaced, more than individual tooth width | ? | ? |
| Solnhofen | ? | ? | Widely spaced, more than individual tooth width | ? | ? |
| Munich | 12 | Present | Anterior teeth closely spaced, from d7 to d8 onwards widely spaced | All teeth procumbent | All teeth slender and recurved |
| Daiting | ? | ? | Widely spaced, about individual tooth width | All teeth procumbent | Middle and posterior dentary teeth bulbous |
| 11th | ? | ? | All teeth closely spaced | Perpendicular | Anterior teeth robust, but recurved, posterior teeth bulbous |
| 12th | 13 | Present | Anterior teeth closely spaced, from d7 to d8 onwards widely spaced | Largely perpendicular | Anterior teeth slender and recurved, middle and posterior teeth bulbous |

premaxillary and anterior dentary teeth, and the number of dentary teeth. Thus, assuming an ontogenetic trend, smaller specimens possess recurved premaxillary and maxillary (Eichstätt and Thermopolis specimen) and dentary teeth (Eichstätt and Munich specimen), changing to a more bulbous morphology with growth, while the number of dentary teeth increases from 11 to 13. However, the presence of bulbous maxillary and dentary teeth in the Daiting specimen, which is smaller than both the Thermopolis and Munich specimens in respect to its humeral length (OR, own data), casts doubt on this interpretation, at least in respect to tooth shape. Furthermore, the size differences between the Thermopolis and several other specimens with bulbous teeth are minimal. Thus, the 11th and 12th specimens are 10% and 5% larger than the Thermopolis specimen (based on femoral length), respectively, but both show well-developed bulbous teeth in both maxilla and dentary.

Current research has shown increasing evidence that the different basins of the Late Jurassic limestones of Bavaria are characterised by different ages and fossil assemblages (*Röper, 2005*; *Schweigert, 2007*, *2015*; *Viohl & Zapp, 2007*; *López-Arbarello & Schröder, 2014*; *Ebert, Kölbl-Ebert & Lane, 2015*; *Rauhut et al., 2017*; *Foth & Rauhut, 2017*). Based on stratigraphic data, *Archaeopteryx* spans of the entire *Hybonoticeras hybonotum* zone of the Tithonian (from the lowermost Painten Formation to the Mörnsheim Formation), including five ammonite horizons (*Schweigert, 2015*). Given that *Schweigert (2005)* calculated an average duration of 165,000 years per ammonite horizon in the Late Jurassic, this thus represents approximately 700,000 to one million years (see also *Viohl, 2015b*). Furthermore, the finds of the *Archaeopteryx* spread from the Schamhaupten Basin in the east to the Langenaltheim–Solnhofen Basin and Rennertshofen Basin in the west, with the eastern basins being generally older (*Schweigert, 2015*) and showing a higher terrestrial influence, indicated by abundant plant remains (including tree trunks), lepidosaurs, freshwater turtles and the non-avialan theropod dinosaurs (*Reisdorf & Wuttke, 2012*; *Joyce, 2015*; *Jung, 2015*; *Tischlinger & Rauhut, 2015*; *Tischlinger, Göhlich & Rauhut, 2015*; *Viohl,*

*2015b*). Thus, both anagenesis and biogeography could be further factors that influence variation in *Archaeopteryx*. However, the order of specimens from east to west does not reveal any pattern in the dentition. As noted by *Viohl (1998)*, the limited flight capabilities of *Archaeopteryx* and the significant distance to the larger landmasses of the Rhenian and Bohemian massifs strongly indicate that the habitat of *Archaeopteryx* were numerous small islands formed by emerged parts of the reef complexes surrounding the plattenkalk basins, which resulted from a notable transgression at around the Kimmeridgian–Tithonian boundary (*Keupp et al., 2007*; *Viohl, 2015b*). One possibility might thus be that, following the immigration of the common ancestor of all *Archaeopteryx* specimens into the Solnhofen Archipelago in the late Kimmerdigian, probably from the east (*Foth & Rauhut, 2017*), populations of the urvogel became isolated on different islands and evolved in divergent directions, possibly resulting in a 'species flock' of *Archaeopteryx* due to island speciation (*Glor, Losos & Larson, 2005*; *Losos & Ricklefs, 2009*). If so, the variation in the dentition of *Archaeopteryx* might represent to some degree a Jurassic equivalent to the famous Darwin Finches, in which species show a great variety in the shape of their beaks, correlating with respective diet spectra (*Sato et al., 2001*; *Abzhanov et al., 2006*; *Lamichhaney et al., 2015*). However, in the absence of more rigorous approaches to the taxonomy of *Archaeopteryx* (e.g. by summarising all morphological and morphometric variation in a phylogenetic analysis on specimen level; see e.g. *Yates, 2003*; *Tschopp, Mateus & Benson, 2015*), this hypothesis must remain speculative at the moment.

## CONCLUSIONS

Although *Archaeopteryx* is by now a rather well-known taxon of avialan theropod, new specimens, such as the 12th skeletal specimen described here, still add valuable information on the anatomy of this genus. Apart from finally confirming the presence of a closed postorbital bar, the new specimen also showed a separate prefrontal and coronoid ossification for the first time, although at least the presence of a prefrontal could subsequently be confirmed in two other specimens (Eichstätt and Thermopolis specimens).

The discovery of numerous small-sized paravian theropods in the Late Jurassic and Early Cretaceous of China in the past decades have greatly enhanced our understanding of basal paravian anatomy and evolution. However, they also provided sometimes confusing evidence of widespread convergence and parallel evolution in this clade, highlighting that the assumption that all paravian specimens from the Solnhofen Archipelago can by default be assigned to *Archaeopteryx* might be problematic (see also *Foth & Rauhut, 2017*). Thus, a more rigorous diagnosis of this genus was necessary to better evaluate if all of the specimens traditionally referred to this taxon can really be identified as *Archaeopteryx*. Although few of the characters found can currently be regarded as true autpomorphies (as many characters have a mosaic-like distribution in other basal paravians), there is strong evidence for a truly unique combination of characters that helps to diagnose the genus *Archaeopteryx* (or, at least, a monophyletic Archaeopterygidae if one considers differences between specimens to be sufficient for separation of different genera; see *Howgate, 1985*; *Elzanowski, 2001b*). Although differences in the preservation of different specimens makes

the evaluation of this character combination sometimes difficult, there is good reason to assume that most of the specimens traditionally identified as *Archaeopteryx* really represent this taxon, with the exception of the Haarlem specimen, which also represents the geographically most divergent specimen (*Foth & Rauhut, 2017*).

Given the apparent diversity of paravian theropods in slightly older rocks in China (see *Xu & Zhang, 2005*; *Xu et al., 2009a*, *2011*, *2015b*; *Godefroit et al., 2013a*, *2013b*; *Lefèvre et al., 2017*; *Pei et al., 2017*), this low generic diversity of paravian theropods in the Solnhofen Archipelago might be surprising. However, it might reflect a rather recent immigration of these derived maniraptorans from eastern Asia, possibly aided by limited flight abilities of *Archaeopteryx* or its immediate ancestor (*Foth & Rauhut, 2017*). As indicated by the high variation seen in the different specimens of *Archaeopteryx*, this immigration might have been followed by a rapid radiation of species within this genus, but more data is needed to investigate this hypothesis further.

## ACKNOWLEDGEMENTS

First and foremost, we have to thank the finder of the 12th specimen of *Archaeopteryx*, who wants to remain anonymous. Without his professional excavation and prudent handling of the situation this specimen might never have become available for scientific study! Further sincere thanks go to Raimund Albersdörfer, who supervised the preparation and, together with the finder, made the specimen available for study. We furthermore thank the preparators who did an extraordinary job in the preparation of this difficult specimen. Many colleagues have helped with discussions, access to specimens and providing photographs and/or data, including Martina Kölbl-Ebert, Christoph Keilmann, Adriana López-Arbarello, Gerald Mayr, Mark Norell, Jingmai O'Connor, Rui Pei, Burkhard Pohl, Martin Röper, Anne Schulp, Paul Tafforeau, Peter Wellnhofer, and Xu Xing. Critical reviews by Michael Pittman and Thomas Holtz greatly helped to improve the paper. This is a contribution to Volkswagen Foundation Project I/84 640.

### Funding

This work was supported by the Volkswagen Foundation (No. I/84 640). The funders had no role in study design, data collection and analysis, decision to publish, or preparation of the manuscript.

### Grant Disclosures

The following grant information was disclosed by the authors:
Volkswagen Foundation: No. I/84 640.

### Competing Interests

The authors declare that they have no competing interests.

## Author Contributions

- Oliver W.M. Rauhut conceived and designed the experiments, performed the experiments, analyzed the data, wrote the paper, prepared figures and/or tables.
- Christian Foth performed the experiments, analyzed the data, wrote the paper, prepared figures and/or tables.
- Helmut Tischlinger contributed reagents/materials/analysis tools, wrote the paper, prepared figures and/or tables, reviewed drafts of the paper.

## Data Availability

The specimen described is on permanent exhibition at the Dinosaurier-Park Altmühltal in Denkendorf, Bavaria, where it is also available for additional scientific study.

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
