# Peer review of "The oldest Archaeopteryx (Theropoda: Avialiae): a new specimen from the Kimmeridgian/Tithonian boundary of Schamhaupten, Bavaria"

_PeerJ, doi:10.7717/peerj.4191_

## Round 0.1 · original submission · Minor Revisions

Dear authors,

I am sorry for the delay in this decision. I have accepted the decision of 'minor revision' from reviewer two, although the comments from both reviewers should be quick and easy to make.

Please note the comments made during the pause in the review process, and if it is possible to rectify the issue, then that would be very helpful.

Once again, thank you for submitting your manuscript to PeerJ.

·

Basic reporting

The description of the new specimen is thorough, both in text form and as documented photographically.

I was somewhat concerned about the legality of and potential access in posterity for the specimen. However, the authors seem to have found an effective and appropriate means of making the specimen permanently available to scientific study.

Experimental design

The paper is essentially all descriptive, not analytical. Thus there is no significant part of this particular study which involves experimental design.

The relevant specimen data, however, all all provided. Thus, future workers would be able to independently evaluate the morphological observations of the author, and could potentially study the original locality in search of additional specimens or new stratigraphic/sedimentological data.

Validity of the findings

The morphological observations presented are sound, and measurements of the skeletal elements are provided.

Additional comments

Additional comments and corrections:
p. 9, line 62: Origin of Species should be in italics, not quotation marks.

p. 9, lines 69: “1970ies” should be “1970s”.

·

Basic reporting

Clear and unambiguous, professional English used throughout.

Sufficient field background/context provided. Additional references could further enhance the paper but not essential (see reviewer PDF).

Professional article structure, figs, tables. Some minor suggestions to the figures. Suggestions for two non-essential additional supplementary tables (see reviewer PDF).

Self-contained with relevant results to hypotheses.

Experimental design

Original primary research within Aims and Scope of the journal

Research question well defined, relevant & meaningful. It is stated how research fills an identified knowledge gap.

Rigorous investigation performed to a high technical & ethical standard.

A paragraph of the UV imaging methodology should be added as their are a range of UV imaging techniques currently available. The methodology is otherwise sufficient in detail and information to replicate the results.

Validity of the findings

Valid, well-supported and impactful findings. Data are robust. Conclusions are clear and speculation is stated as such.

Additional comments

This is an impressive amount of work, well done! I am actually supportive of publication as is (with a few typo and grammar corrections and a short paragraph on your UV methodology), but it would be great if you can take my minor corrections on board as I think they will improve the quality and value of the paper.

Importantly, I think the manuscript should not be shortened. In fact my suggested corrections would involve a very slight lengthening of the paper.

---

## Round 0.2 · Minor Revisions

Dear authors,

There are a few small revisions that still need to be made (see the reviewers' comments).

I look forward to receiving your revised manuscript.

·

Basic reporting

Meets PeerJ's standards. The authors addressed the comments from my first review.

- Clear and unambiguous, professional English used throughout.
- Literature references, sufficient field background/context provided.
- Professional article structure, figs, tables. Raw data shared.
- Self-contained with relevant results to hypotheses.

Experimental design

Meets PeerJ's standards. The authors addressed the comments from my first review.

- Original primary research within aims and scope of the journal.
- Research question well defined, relevant & meaningful. It is clear how the research fills
identified knowledge gap.
- Rigorous investigation performed to a high technical & ethical standard.
- Methods described with sufficient detail & information to replicate.

Validity of the findings

Meets PeerJ's standards. The authors addressed the comments from my first review.

- Impact and novelty not assessed. Negative/inconclusive results accepted. Meaningful details for replication provided (rationale and benefit to literature clearly stated)
- Data is robust.
- Conclusions are well stated, linked to original research question & limited to supporting results.
- Speculation identified as such.

Additional comments

Oliver, Christian and Helmut,

Thank you for taking my comments on board. Here are some final items to address. I do not need to see the manuscript again and fully support its publication.

- line 139: 'artificial' not needed
- lines 165 and 166: replace '.' with ',' in '50.000' and '8.000' so it is clear you mean fifty-thousand microwatts and eight-thousand microwatts.
- typos in the marked changes on lines 1166, 1327, 1381, 1419 and 1564.
- line 1921: the potential cartilaginous pad shown in the LSF image in Fig. 4b of Wang, Pittman et al. 2017 is visible as a orange-coloured feature under white light. You could perhaps change the text to support the fact that soft tissues can preserve differently in the same specimen e.g. the orange-coloured pad vs. the white-coloured areas that reveal patagia. This might mean that some 'absent' soft tissues could be possible to reveal in the future using a wider suite of imaging techniques.

I look forward to seeing the published paper.

Best regards,

Michael Pittman

---

## Round 0.3 · accepted · Accept

Dear authors,

Many thanks for your revised manuscript. After reading it, I have accepted it for publication in PeerJ.

Once again, thank you for submitting your manuscript to PeerJ and I hope you will use us again as your publication venue.

If we need to clarify any details required to move the manuscript forward, then our production staff will get in touch with you. Otherwise, a proof will be forthcoming shortly for your review.

Congratulations and thank you for your submission.